# Real-Time Visual Attribution Streaming in Thinking Model

**Seil Kang** [1]  **Woojung Han** [1]  **Junhyeok Kim** [1]  **Jinyeong Kim** [1]  **Youngeun Kim** [2]  **Seong Jae Hwang** [1]

## Abstract

We present an amortized framework for real-time *visual attribution streaming* in multimodal thinking models. When these models generate code from a screenshot or solve math problems from images, their long reasoning traces should be grounded in visual evidence. However, verifying this reliance is challenging: faithful causal methods require costly repeated backward passes or perturbations, while raw attention maps offer instant access, they lack causal validity. To resolve this, we introduce an amortized approach that learns to estimate the causal effects of semantic regions directly from the rich signals encoded in attention features. Across five diverse benchmarks and four thinking models, our approach achieves faithfulness comparable to exhaustive causal methods while enabling visual attribution streaming, where users observe grounding evidence as the model reasons, not after. Our results demonstrate that real-time, faithful attribution in multimodal thinking models is achievable through lightweight learning, not brute-force computation.

(a) Inference Time vs. Faithfulness

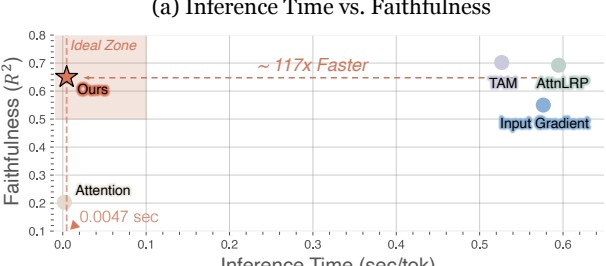

(b) Per-token Latency vs. Context Length

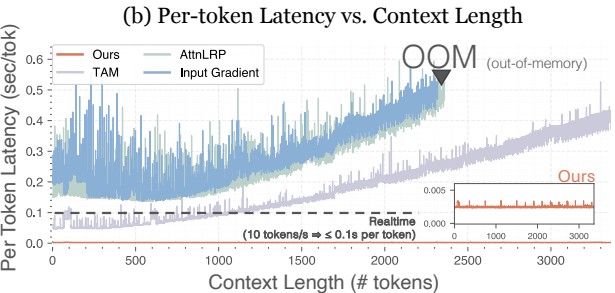

*Figure 1.* **Faithfulness-Efficiency Trade-off.** (a) Baseline methods compromise either efficiency or faithfulness. Our approach simultaneously achieves both faithfulness and efficiency ($R^2$ of predicted vs. actual logit drops). (b) Latency scaling with context length. Unlike the baseline, which exhibits linear cost growth and OOM errors on long traces, our method operates with constant overhead below the real-time threshold (dashed line).

## 1. Introduction

Recent scaling of Vision-Language Models has shifted the focus from single-turn question answering to multi-step reasoning over visual evidence (OpenAI, 2024; 2025; DeepSeek-AI et al., 2025; DeepMind, 2024; Qwen Team, 2024; Yue et al., 2024). Todays, they can generate code from website screenshots, solve geometry problems from diagrams, and interpret complex charts (Lu et al., 2024; Yue et al., 2024; Zhang et al., 2024). By decomposing queries into intermediate steps, these models produce extended reasoning traces that theoretically reveal *which* visual elements support *which* conclusions (Xu et al., 2025; Wang et al., 2024b; Team et al., 2024). However, verifying whether these

long traces genuinely rely on visual evidence remains a critical challenge.

In practice, reasoning traces often exhibit *ungrounded hallucinations*, producing plausible narratives without actual visual support (Li et al., 2023; Favero et al., 2024). Models may correctly cite "the angle at vertex B" while attending to an irrelevant image region (Tong et al., 2024), or hallucinate numerical values absent from the visual input (Jing et al., 2024). These failures create a diagnostic blind spot, making it unclear whether errors originate in perception or symbolic reasoning (Chen et al., 2025). As reasoning chains grow longer, models increasingly rely on language priors rather than visual evidence (Xu et al., 2026), yet without reliable tools to verify visual grounding, the interpretability promised by reasoning traces remains illusory.

Attribution methods offer a principled way to verify visual grounding by quantifying which image regions causally

[1]Yonsei University [2]Amazon. Correspondence to: Seong Jae Hwang <seongjae@yonsei.ac.kr>.

*Proceedings of the $43^{rd}$ International Conference on Machine Learning*, Seoul, South Korea. PMLR 306, 2026. Copyright 2026 by the author(s).

influence each generated token. However, as shown in Figure 1a, existing approaches face a competing demands of faithfulness and efficiency. Raw attention weights (Abnar & Zuidema, 2020) can be extracted instantly during inference, but they are unreliable as causal explanations because attention distributions can shift substantially without altering model predictions (Jain & Wallace, 2019; Pruthi et al., 2020; Wu et al., 2024). Gradient-based (Zeiler & Fergus, 2014; Chefer et al., 2021; Achtibat et al., 2024) and perturbation-based methods (Fong & Vedaldi, 2017; Hooker et al., 2019; Li et al., 2025c) achieve higher faithfulness by measuring how outputs change when inputs are modified, but at substantial computational cost. This cost grows with context length (Figure 1b). As reasoning traces extend to thousands of tokens, per-token latency increases dramatically, making real-time analysis infeasible. For interactive debugging, where users iteratively refine prompts or inspect reasoning step-by-step, such latency is unacceptable.

We resolve this trade-off through *amortized attribution* (Jethani et al., 2021; Covert et al., 2024), training a lightweight estimator as a surrogate model to predict attributions rather than recompute them from scratch. Crucially, while individual attention weights are unreliable as direct explanations, the comprehensive attention patterns distributed across layers and heads serve as *informative features* for predicting causal effects (Abnar & Zuidema, 2020; Cohen-Wang et al., 2024; 2025). We hypothesize that these patterns contain sufficient signal to recover the counterfactual effect of masking a visual region on token probability. To ensure perceptual relevance, we operate on semantic units derived from self-supervised features (Siméoni et al., 2025) rather than raw pixels. By training a linear estimator to map attention features to rigorous ablation targets, we bypass the need for repeated inference.

With a minimal parameter footprint, we train our estimator on 2,000 examples in ∼4.5 hours of runtime on a single GPU, a one-time cost that amortizes over all subsequent inferences. At test time, our method computes attribution asynchronously within the token generation loop, enabling users to inspect on the fly where the model draws from at each reasoning step.

Our main contributions are as follows:

- We formalize token-level visual attribution for multimodal reasoning as counterfactual region ablation effects, and identify the prohibitive scaling of direct perturbation for long reasoning traces (Section 3.2).

- We propose a real-time attribution streaming method that (i) constructs semantically coherent attribution units from self-supervised vision features and (ii) amortizes counterfactual effects from a single forward pass (Sections 3.3 and 3.5).

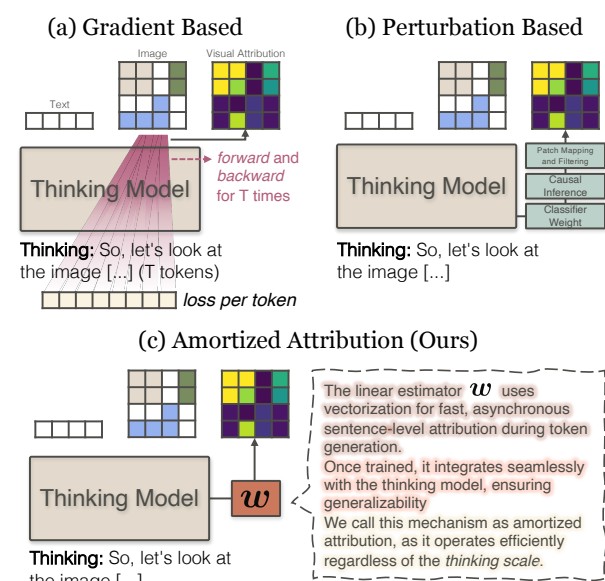

*Figure 2.* **Comparison of attribution processes** between our method and baselines.

- Our amortized estimator enables single-pass inference with cost linear in trace length while matching strong gradient- and perturbation-based baselines on faithfulness metrics across four reasoning VLMs and five task families (Sections 3.5 and 4).

- We further study models' thinking steps by providing an inspectable view of the semantic space over long traces and analyzing the reasoning trajectory dynamics enabled by attribution streaming (Section 4.4).

## 2. Related Work

### 2.1. Multimodal Reasoning Paradigms

Recent vision-language models have shifted toward multi-step reasoning grounded in visual evidence, driven by benchmarks that explicitly require compositional explanations (Yue et al., 2024; Zhang et al., 2024; Fu et al., 2024; Lu et al., 2022; 2024; He et al., 2024; Yuan et al., 2025; Man et al., 2025; Wu et al., 2025). Correspondingly, models now generate long-form "thinking" traces (Xu et al., 2025; Wang et al., 2024b; Team et al., 2024; DeepMind, 2024; OpenAI, 2024; 2025) and employ architectures designed for broader reasoning capabilities (Li et al., 2025a;b; Hong et al., 2024; Luo et al., 2025). While these traces offer a debugging interface, they can sound plausible even when ungrounded, masking whether errors are perceptual or symbolic. Current reliability methods largely focus on final answer correctness rather than the supporting visual evidence for each model's thinking step (Prabhu et al., 2025; Geigle et al., 2024; Yan et al., 2024; Fu et al., 2025; Liu et al., 2026).

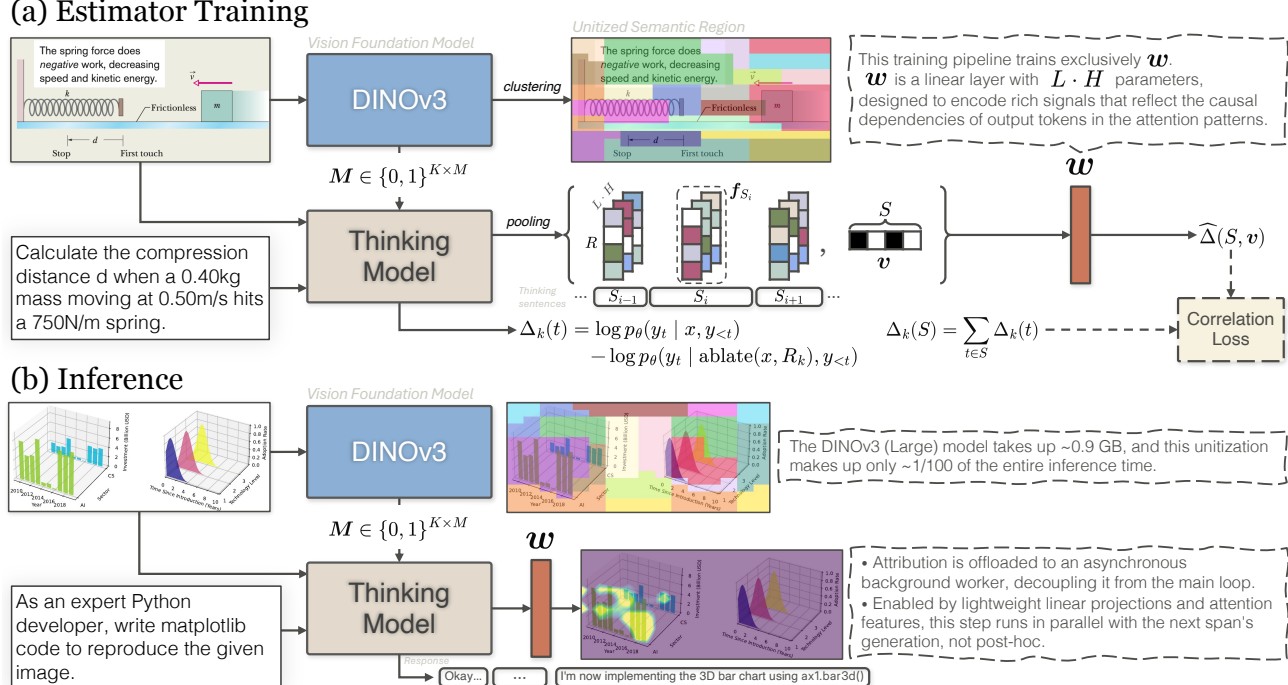

*Figure 3.* **Overview of our amortized attribution pipeline.** (a) Training: We use our semantic region unitization to identify semantic regions of an input image using DINO features and optimize a lightweight estimator to predict causal importance from attention patterns. (b) Inference: Once the estimator is trained, it generalizes to other samples and computes visual attribution in parallel with the model's token generation.

## 2.2. Faithful Attribution and Scalability

Attribution methods quantify feature relevance and are typically evaluated via counterfactual faithfulness (Hooker et al., 2019), often using deletion or masking perturbations (Petsiuk et al., 2018; Fong & Vedaldi, 2017; Yu et al., 2024; Zhang et al., 2025). While text-to-image cross-attention is a common proxy for focus, it is often unfaithful (Jain & Wallace, 2019; Serrano & Smith, 2019; Bibal et al., 2022) and manipulable without affecting predictions (Pruthi et al., 2020). Although gradient or attention-flow methods (Abnar & Zuidema, 2020; Chefer et al., 2021; Achtibat et al., 2024) and recent multimodal tools (Li et al., 2025c; Ben Melech Stan et al., 2024; Shen, 2025; Liang et al., 2025) improve fidelity, they scale poorly. Reasoning models producing thousands of tokens (Xu et al., 2025; Team et al., 2024) make interactive per-token attribution intractable. Prior amortized attribution methods train surrogates over SHAP-style value functions (Jethani et al., 2021; Covert et al., 2024), and concurrent work AT2 (Cohen-Wang et al., 2025) uses attention as features for text attribution. We extend this line to visual tokens under streaming generation, replacing Monte Carlo sampling with a linear estimator over cached cross-attention features that adds negligible overhead to the decoding loop.

## 3. Method

We present vSTREAM (Visual Attribution Streaming), a framework for real-time, object-centric attribution in reasoning VLMs (Figure 3). Our approach sidesteps the computational bottleneck of existing methods (Chefer et al., 2021; Li et al., 2025c) by amortizing the counterfactual computation, learning to predict ablation effects from lightweight features extracted in a single forward pass.

### 3.1. Background

We consider a vision-language model (VLM) that takes a multimodal input $x = (I, q)$, where $I$ is an image and $q$ is a textual query. The model autoregressively generates a sequence of tokens $y_{1:T} = (y_1, \ldots, y_T)$. For reasoning-centric VLMs, this sequence typically includes both intermediate reasoning tokens (the "thinking" trace) and the final answer. We denote the model distribution at step $t$ as $p_\theta(\cdot \mid x, y_{<t})$.

### 3.2. Problem Formulation

Let the image $I$ be encoded into $M$ vision tokens, and let these tokens be partitioned into $K$ disjoint regions $\mathcal{R} = \{R_1, \ldots, R_K\}$, where each $R_k \subseteq \{1, \ldots, M\}$. We define an ablation operator $\mathrm{ablate}(x, R_k)$ that prevents the model from accessing visual information in region $R_k$ dur-

ing generation.

**Ablation via Attention Masking.** We implement ablation by preventing information flow from specific vision tokens to the query, effectively removing them from the model's context. Following Geva et al. (2023), we set the attention scores from all query positions to the vision tokens in region $R_k$ to $-\infty$ before the softmax normalization. Formally, let $\boldsymbol{q}_t \in \mathbb{R}^d$ denote the query vector at position $t$ and let $\boldsymbol{K} \in \mathbb{R}^{d \times M}$ contain the key vectors for all $M$ vision tokens. The standard attention logits are $\boldsymbol{s} = \boldsymbol{q}_t^\top \boldsymbol{K} / \sqrt{d}$. Under ablation of region $R_k$, we modify the logits as:

$$s_i \leftarrow \begin{cases} -\infty & \text{if } i \in R_k \\ s_i & \text{otherwise} \end{cases} \tag{1}$$

for all layers and heads. This masking-based intervention preserves the original token positions and model state while cleanly removing the target region's influence on subsequent generation (Cohen-Wang et al., 2025; 2024).

**Ablation Effect.** We define the token-level ablation effect of region $R_k$ at decoding step $t$ as the drop in log-probability of the generated token (Wang et al., 2022; Zhang & Nanda, 2023):

$$\Delta_k(t) = \log p_\theta(y_t \mid x, y_{<t}) - \log p_\theta(y_t \mid \text{ablate}(x, R_k), y_{<t}). \tag{2}$$

A positive $\Delta_k(t)$ indicates that region $R_k$ supports the generation of token $y_t$.

We extend this token-level metric to semantic spans. Let $S \subseteq \{1, \ldots, T\}$ denote a target span (e.g., a thinking step). We aggregate token-level effects to define the span-level ablation effect:

$$\Delta_k(S) = \sum_{t \in S} \Delta_k(t). \tag{3}$$

Under teacher-forcing, where we condition on the originally generated sequence $y_{<t}$ for both the original and ablated models, this sum equals the log-probability ratio of the entire span $\log p(y_S \mid x) - \log p(y_S \mid \text{ablate}(x, R_k))$.

**From Tokens to Semantic Regions.** While defining $R_k$ as individual vision tokens is theoretically possible, it is practically suboptimal (Ren & Malik, 2003). First, individual visual tokens are semantically ambiguous; a single token might capture an edge or texture that has no independent meaning without its neighbors. Second, visual information is spatially sparse and structured. Their salient information is concentrated in objects rather than distributed uniformly. Grouping tokens into semantically coherent units not only aligns attribution with human concepts (e.g., "the red car" vs. "token 142") but also drastically reduces the search space for ablation. We therefore partition the image into semantic regions, making the counterfactual objective both interpretable and computationally tractable.

### 3.3. Semantic Region Unitization

The choice of region partition $\mathcal{R}$ is critical, as attributions to arbitrary patches are hard to interpret while attributions to semantic regions (objects, text blocks, diagram components) are actionable. We use features from DINOv3 (Siméoni et al., 2025), a visual foundation model, to partition vision tokens into semantically coherent regions. After aligning feature maps to the VLM's vision encoder resolution, we apply agglomerative clustering with Ward's linkage (Ward Jr, 1963), adaptively grouping similar tokens into $K$ disjoint regions (typically $K \in [16, 128]$) that isolate objects and background elements without requiring external segmentation masks. We denote the resulting partition as a membership matrix $\boldsymbol{M} \in \{0,1\}^{K \times M}$, where $\boldsymbol{M}_{k,i} = 1$ indicates vision token $i$ belongs to region $R_k$. See Section C for more details and explanations.

For ablation studies, we also consider three non-semantic baselines:

- **Token-wise**: Each vision token is its own region ($K = M$).

- **Random blocks**: Partition the image into $K$ randomly sized rectangular regions.

- **Voronoi**: Partition using a regular grid of $K$ Voronoi cells.

These baselines use fixed $K \in \{16, 32, 64, 128\}$, while DINO clustering determines $K$ adaptively; see Section 4.3 for quantitative comparisons.

### 3.4. Attention Feature Extraction

To predict ablation effects without actually ablating, we need features that capture how much each region contributes to generating the target span. Attention patterns provide a natural signal: if a region strongly influences the generation of a span, blocking attention to that region should substantially change the output.

Let $\boldsymbol{A}_{t,i}^{(\ell,h)}$ denote the attention weight from token position $t$ to vision token $i$ in layer $\ell$ and head $h$. For each target span $S$ and region $R_k$, we compute the mean attention pooled over the span and the region:

$$\boldsymbol{a}_{S,k}^{(\ell,h)} = \frac{1}{|S| \cdot |R_k|} \sum_{t \in S} \sum_{i \in R_k} \boldsymbol{A}_{t,i}^{(\ell,h)}. \tag{4}$$

We concatenate these scalars across all layers and heads to form a feature vector:

$$\boldsymbol{f}_{S,k} = \text{Concat}_{\ell,h}\big(\boldsymbol{a}_{S,k}^{(\ell,h)}\big) \in \mathbb{R}^{L \cdot H}, \tag{5}$$

where $L$ is the number of layers and $H$ is the number of heads. For a model with 32 layers and 36 heads, this yields

a 1152-dimensional feature vector per (span, region) pair. We extract these features from samples where the model produces correct final answers, ensuring that the learned estimator captures attention patterns associated with successful reasoning (see Section B.1 for dataset details). Importantly, attention weights are already computed during generation, so extracting $\boldsymbol{f}_{S,k}$ adds negligible overhead.

### 3.5. Amortized Estimator

Given the attention features $\boldsymbol{f}_{S,k}$ for each region, we train a linear estimator to predict ablation effects. Let $\boldsymbol{v} \in \{-1, +1\}^K$ be a signed region mask where $\boldsymbol{v}_k = -1$ indicates ablation and $\boldsymbol{v}_k = +1$ indicates retention. This is distinct from the membership matrix $\boldsymbol{M} \in \{0, 1\}^{K \times M}$, which assigns tokens to regions. The signed encoding ensures that $\boldsymbol{f}_{S,\boldsymbol{v}}$ (Equation (6)) captures the difference between retained and ablated contributions, rather than a one-sided sum of retained features only. We define the combined feature for mask $\boldsymbol{v}$ as:

$$\boldsymbol{f}_{S,\boldsymbol{v}} = \sum_{k=1}^{K} \boldsymbol{v}_k \cdot \boldsymbol{f}_{S,k}. \tag{6}$$

A linear estimator with weights $\boldsymbol{w} \in \mathbb{R}^{L \cdot H}$ predicts the total ablation effect:

$$\widehat{\Delta}(S, \boldsymbol{v}) = \boldsymbol{w}^\top \boldsymbol{f}_{S,\boldsymbol{v}}. \tag{7}$$

The estimator has only $L \cdot H$ parameters, learning a single importance weight per layer-head pair.

**Training.** For each training sample, we generate $N{=}32$ random masks $\boldsymbol{v}^{(j)}$ and compute the ground-truth ablation effect $\Delta(S, \boldsymbol{v}^{(j)})$ via forward passes. We optimize $\boldsymbol{w}$ to maximize Pearson correlation between predictions and targets:

$$\mathcal{L}(\boldsymbol{w}) = -\rho\big(\{\widehat{\Delta}(S, \boldsymbol{v}^{(j)})\}_j, \{\Delta(S, \boldsymbol{v}^{(j)})\}_j\big). \tag{8}$$

This scale-invariant objective focuses on relative ranking rather than absolute magnitudes, aligning with our evaluation metric. We evaluate deeper MLP variants in Section F.9: the linear model matches their accuracy while remaining orders of magnitude more parameter-efficient. See Section A for theoretical justification and Section B.4 for training details.

### 3.6. VSTREAM: Visual Attribution Streaming

As the model generates its reasoning trace, we simultaneously stream visual attributions by caching text-to-vision cross-attention weights at each decoding step. Upon completing a semantic span $S$, we aggregate trained weights into $\boldsymbol{f}_{S,k}$ and score each region via

$$\widehat{\Delta}_k(S) = \boldsymbol{w}^\top \boldsymbol{f}_{S,k}. \tag{9}$$

Crucially, this computation is offloaded to an asynchronous background worker, decoupled from the main generation loop via a producer-consumer pattern. Because the attribution step runs in parallel with the generation of the subsequent span, it introduces near-zero latency overhead to the user experience (Figure 3). See Section B.8 for detailed streaming architecture.

While region-level scores capture relevant objects, finer localization is often required. We redistribute the region attribution $\widehat{\Delta}_k(S)$ across patches using the DINOv3 attention map $\boldsymbol{a}^{\text{DINO}}$ (computed during unitization) as a spatial prior:

$$s_i = \widehat{\Delta}_{R(i)}(S) \cdot \frac{\boldsymbol{a}_i^{\text{DINO}}}{\sum_{j \in R(i)} \boldsymbol{a}_j^{\text{DINO}}}. \tag{10}$$

This refinement highlights the most salient parts within each region while preserving the calibrated total importance from the estimator. See Section A for a complexity analysis.

## 4. Experiments

We evaluate our method on four reasoning VLMs across five task categories, addressing three questions: (1) Does our method accurately predict ablation effects? (2) Does the estimator generalize across tasks? (3) What do the attributions reveal about model behavior? Lastly, we conduct additional analysis to understand how visual attribution dynamics differ between successful and unsuccessful reasoning.

### 4.1. Setup

**Models and datasets.** We use four reasoning VLMs that generate extended thinking chains: Qwen3-VL-8B-Thinking (Qwen Team, 2025), GLM-4.1V-9B-Thinking (Hong et al., 2025), MiMo-VL-7B-RL (Team et al., 2025), and Cosmos-Reason1-7B (NVIDIA et al., 2025). To test generalization, we cover five categories: Math, Science, Document, Code, and General. See Section B.1 for dataset details per category.

**Baselines.** We compare against five baselines spanning different paradigms:

- **Random**: Uniform random attribution scores.

- **Attention**: Average attention weights from text tokens to vision tokens, pooled across all layers and heads.

- **InputGrad** (Hechtlinger, 2016): Gradient of output logits with respect to input pixel values.

- **AttnLRP** (Achtibat et al., 2024): Attention-aware layer-wise relevance propagation.

- **TAM** (Li et al., 2025c): Token activation maps based on intermediate activations.

*Table 1.* **Attribution quality comparison.** Each cell: (LDS / Top-5 Drop); higher is better. ● Green: best, ● cream: second best. We define *real-time* as ≥10 tokens/sec, exceeding the average human silent reading speed of 238 words/min (≈6 tokens/sec) (Brysbaert, 2019), ensuring users can follow streaming output; vStream (Ours) achieves up to 117× speedup over gradient-based baselines.

| Method | Qwen3-VL-8B-Thinking | | | | | | | GLM-4.1V-9B-Thinking | | | | | | |
| | Math | Science | Document | Code | General | Avg. | Time (s/10tok) | Math | Science | Document | Code | General | Avg. | Time (s/10tok) |
|---|---|---|---|---|---|---|---|---|---|---|---|---|---|---|
| Random | 0.31/0.08 | 0.28/0.11 | 0.32/0.15 | 0.33/0.09 | 0.29/0.07 | 0.31/0.10 | .010±.002 | 0.28/0.12 | 0.33/0.14 | 0.31/0.13 | 0.33/0.10 | 0.28/0.11 | 0.31/0.12 | .011±.003 |
| Attention | 0.39/0.31 | 0.37/0.28 | 0.42/0.33 | 0.39/0.29 | 0.43/0.26 | 0.40/0.29 | .020±.005 | 0.41/0.30 | 0.42/0.35 | 0.40/0.29 | 0.45/0.32 | 0.38/0.27 | 0.41/0.31 | .022±.006 |
| InputGrad | 0.63/0.72 | 0.71/0.68 | 0.65/0.71 | 0.58/0.96 | 0.64/0.75 | 0.64/0.76 | 2.80±1.06 | 0.58/0.74 | 0.63/1.12 | 0.69/0.73 | 0.62/0.78 | 0.63/0.81 | 0.63/0.84 | 3.05±1.15 |
| AttnLRP | 0.76/0.81 | 0.66/1.08 | 0.73/0.78 | 0.73/0.82 | 0.65/0.98 | 0.71/0.89 | 2.60±1.04 | 0.72/0.79 | 0.68/0.77 | 0.64/1.05 | 0.71/0.84 | 0.70/0.75 | 0.69/0.84 | 2.84±1.13 |
| TAM | 0.69/1.02 | 0.69/0.95 | 0.66/1.04 | 0.72/1.06 | 0.71/0.89 | 0.69/0.99 | 1.90±1.13 | 0.71/0.94 | 0.65/0.91 | 0.68/0.88 | 0.67/1.01 | 0.68/1.08 | 0.68/0.96 | 2.07±1.23 |
| **vStream (Ours)** | 0.75/1.05 | 0.71/0.92 | 0.74/1.09 | 0.70/1.02 | 0.70/1.02 | 0.72/1.00 | **.024±.002** | 0.69/0.97 | 0.74/0.89 | 0.67/1.01 | 0.68/1.10 | 0.73/0.95 | 0.70/0.98 | **.026±.002** |
| Method | MiMo-VL-7B | | | | | | | Cosmos-Reason1-7B | | | | | | |
| | Math | Science | Document | Code | General | Avg. | Time (s/10tok) | Math | Science | Document | Code | General | Avg. | Time (s/10tok) |
| Random | 0.30/0.14 | 0.28/0.18 | 0.29/0.11 | 0.30/0.16 | 0.28/0.12 | 0.29/0.14 | .009±.002 | 0.28/0.17 | 0.30/0.13 | 0.34/0.19 | 0.30/0.16 | 0.27/0.12 | 0.30/0.15 | .010±.002 |
| Attention | 0.45/0.34 | 0.41/0.32 | 0.38/0.29 | 0.40/0.33 | 0.39/0.28 | 0.40/0.31 | .018±.004 | 0.45/0.33 | 0.41/0.35 | 0.37/0.34 | 0.41/0.37 | 0.38/0.38 | 0.41/0.35 | .020±.005 |
| InputGrad | 0.58/1.06 | 0.62/0.78 | 0.70/0.69 | 0.59/0.76 | 0.71/0.68 | 0.64/0.79 | 2.52±0.95 | 0.58/0.71 | 0.73/0.76 | 0.59/0.98 | 0.61/0.72 | 0.60/0.75 | 0.62/0.78 | 2.75±1.04 |
| AttnLRP | 0.74/0.80 | 0.75/0.96 | 0.62/0.95 | 0.72/0.81 | 0.66/1.04 | 0.70/0.91 | 2.34±0.94 | 0.76/0.80 | 0.65/1.09 | 0.71/0.78 | 0.69/1.00 | 0.72/0.79 | 0.71/0.89 | 2.55±1.02 |
| TAM | 0.70/0.98 | 0.66/0.88 | 0.66/0.91 | 0.68/1.04 | 0.64/0.96 | 0.67/0.95 | 1.71±1.02 | 0.75/0.94 | 0.66/1.02 | 0.68/0.89 | 0.65/0.95 | 0.69/1.01 | 0.69/0.90 | 1.86±1.11 |
| **vStream (Ours)** | 0.75/1.08 | 0.65/0.97 | 0.71/0.98 | 0.70/0.96 | 0.70/1.10 | 0.70/1.02 | **.022±.002** | 0.74/0.99 | 0.67/0.90 | 0.68/1.01 | 0.71/1.04 | 0.69/1.05 | 0.70/1.00 | **.024±.002** |

We adapt all baselines to produce region-level scores using our semantic unitization, ensuring fair comparison at the same granularity.

**Metrics.** We use two complementary metrics:

- **LDS (Linear Datamodeling Score)** (Park et al., 2023; Cohen-Wang et al., 2024; 2025): Spearman correlation between predicted and actual ablation effects across regions, measuring how well the method ranks regions by causal importance.

- **Top-K Drop** (Chattopadhay et al., 2018): Log-probability drop when ablating the $K$ most attributed regions. We use $K{=}5$; higher drop indicates the method correctly identified causally important regions.

LDS evaluates ranking fidelity against ground-truth counterfactual effects, while Top-K Drop measures practical utility for identifying critical visual evidence.

### 4.2. Main Results

**Attribution quality.** Table 1 compares methods across four models and five categories. vStream matches the strongest baselines in LDS and achieves best or second-best Top-5 Drop in 16/20 settings. Figure 4 shows vStream achieves $R^2 = 0.65$ between predicted and actual effects. Crucially, vStream streams these attributions with negligible overhead, whereas gradient-based methods require backward passes and perturbation-based methods require additional compute steps. We also evaluate segmentation quality via mIoU in Section F.13.

**Cross-task generalization.** Table 2 examines whether an estimator trained on one task category generalizes to others. Using Qwen3-VL, we train on each category independently and evaluate on all five. In-domain (diagonal) LDS ranges from 0.70–0.74, while cross-task transfer retains 75–90% of this performance for most pairs. Math and Science show strong mutual transfer (LDS 0.62–0.63), likely due to shared diagram structures. Transfer to Document tasks is weaker (LDS 0.54–0.58), reflecting the distinct visual layout of

*Table 2.* **Cross-task generalization.** Each cell: (LDS / Top-5 Drop). Diagonal: in-domain.

| Train | Eval Category | | | | | Avg. |
| | Math | Science | Document | Code | General | |
|---|---|---|---|---|---|---|
| Math | 0.72/0.82 | 0.63/0.68 | 0.54/0.48 | 0.60/0.58 | 0.55/0.50 | 0.61/0.61 |
| Science | 0.62/0.66 | 0.71/0.80 | 0.58/0.54 | 0.55/0.50 | 0.60/0.58 | 0.61/0.62 |
| Document | 0.53/0.50 | 0.56/0.52 | 0.70/0.78 | 0.57/0.54 | 0.65/0.72 | 0.60/0.61 |
| Code | 0.59/0.60 | 0.54/0.48 | 0.56/0.52 | 0.73/0.84 | 0.62/0.66 | 0.61/0.62 |
| General | 0.57/0.54 | 0.61/0.64 | 0.64/0.70 | 0.60/0.62 | 0.74/0.86 | 0.63/0.67 |
| Mix-up | 0.75/1.05 | 0.71/0.92 | 0.74/1.09 | 0.69/0.91 | 0.70/1.02 | 0.72/1.00 |

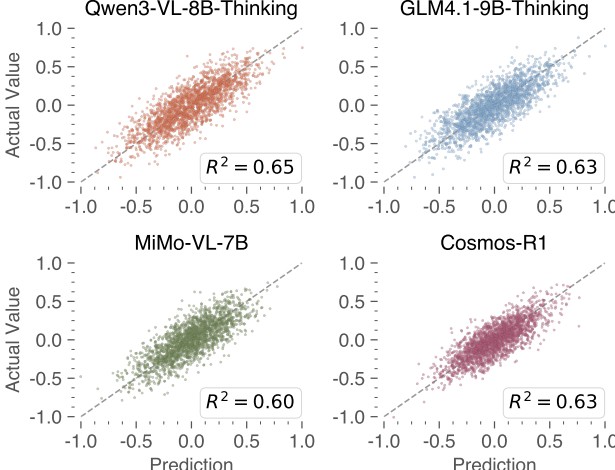

*Figure 4.* **Predicted vs. actual ablation effects.** Each point represents a region's predicted effect versus its ground-truth log-probability drop.

dense text and tables. Training on a mixture of all categories recovers full performance, suggesting a single estimator suffices for diverse applications. See Section F for results on other models.

**Qualitative analysis.** Figure 5 illustrates vStream's unique capability: streaming attributions for each thinking step as the model reasons. Unlike baselines that produce a single post-hoc map, vStream reveals which regions the model references across intermediate steps, exposing failure modes invisible to global methods. For instance, a model may attend to the correct region initially but drift to irrelevant areas mid-reasoning. Additional examples across models and task categories are provided in Section G.

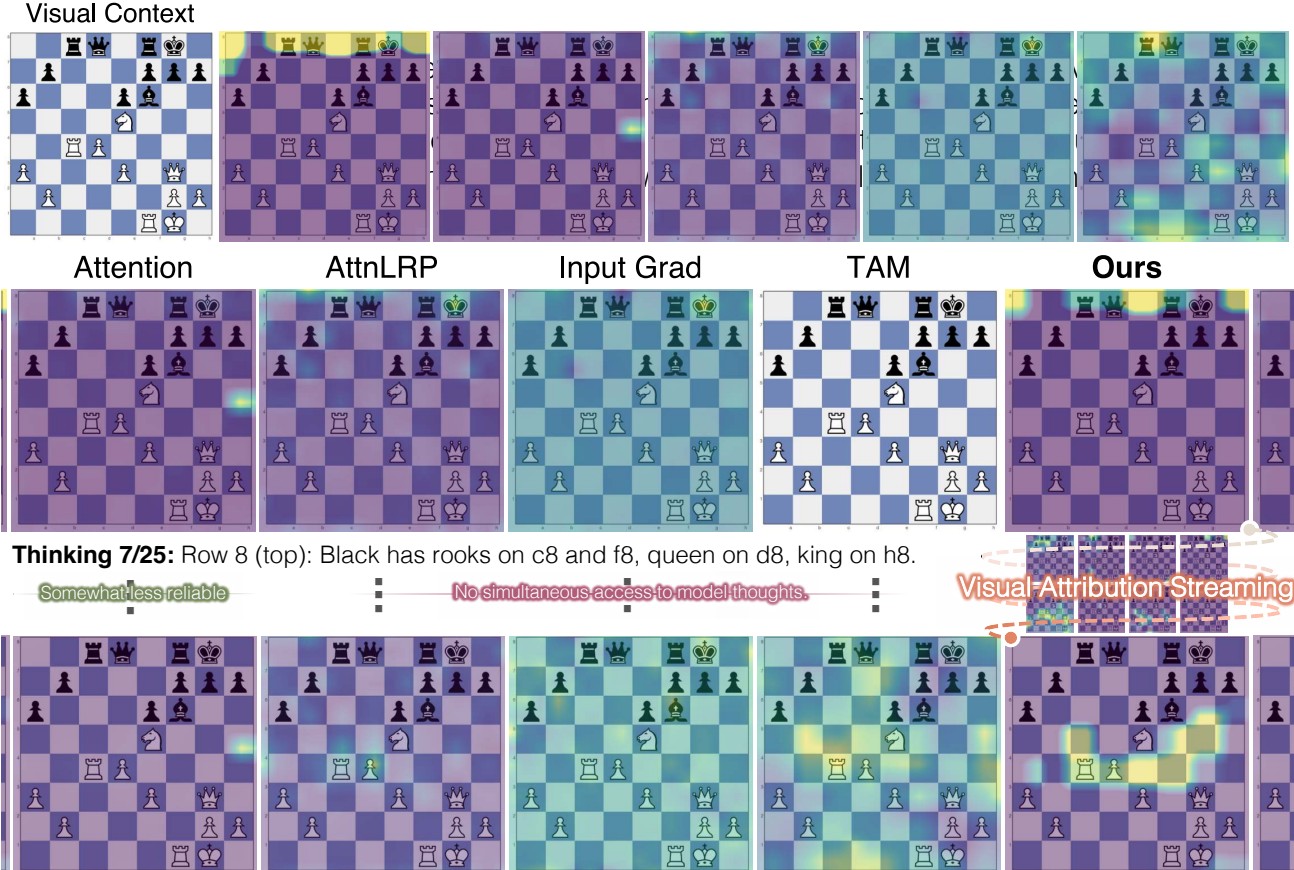

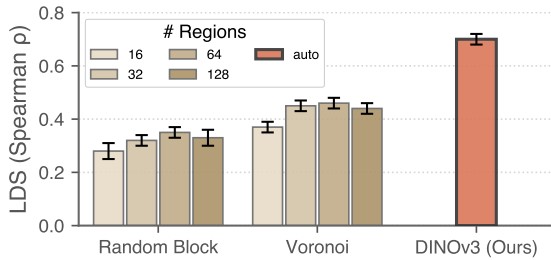

*Figure 5.* **Qualitative comparison on a real-world sample (Qwen3-VL).** VSTREAM emits per-step visual attributions alongside the model's reasoning at near-zero latency, while prior methods only run post-hoc once generation has finished.

### 4.3. Ablation Study

**Semantic regions outperform geometric partitions.** In Figure 6, we compare three region unitization strategies on Qwen3-VL: random rectangular blocks, regular Voronoi tessellation, and our DINOv3-based semantic clustering. Our clustering approach significantly outperforms both geometric alternatives on the LDS metric. Random block and Voronoi partitions use a fixed grid, placing a ceiling on performance, whereas DINOv3-based clustering adaptively adjusts regions to image content. This confirms that semantic-level attribution to objects, symbols, and text blocks directly contributes to more interpretable and accurate results. Results comparing other vision foundation models (Radford et al., 2021; Zhai et al., 2023) are provided in Section F.5.

**Training data efficiency.** A practical concern is how much ablation data is needed to train an effective estimator, as shown in Figure 8. We vary the number of training examples from 100 to 5,000 and measure LDS on a held-out set (Figure 8). Performance improves steeply up to approximately 500 examples, then converges; with 2,000 examples, the estimator reaches full capacity. This efficiency is enabled

*Figure 6.* **Comparison of region unitization strategies.**

by two factors: (1) the estimator has only $L \times H$ parameters (e.g., 784), so overfitting is difficult, and (2) each example provides 32 mask-effect pairs, effectively multiplying the data. Practitioners can train a domain-specific estimator with a few hours of ablation data collection, making the approach accessible for specialized applications. See Section F for additional details.

### 4.4. Reasoning Trajectory Dynamics

Beyond static attribution maps, we ask whether the temporal evolution of visual reliance reveals signatures of reasoning

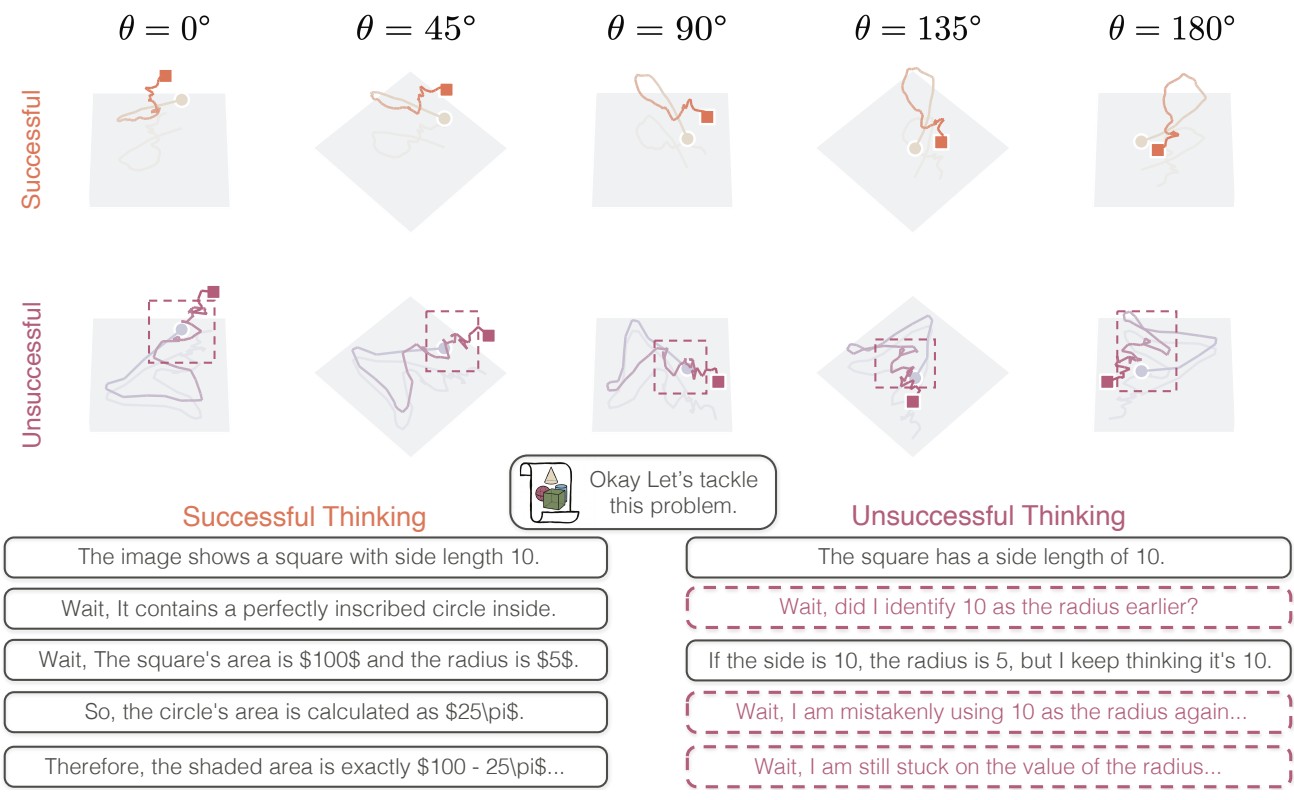

*Figure 7.* **Reasoning trajectories in visual attribution space.** Each curve traces the evolution of a region-effect vector across reasoning steps, projected into 3D via PCA, The figure shows views from five different angles ($\theta$). The thinking process begins at the circular point and terminates at the square point. Successful reasoning chains (● orange) follow compact, directed paths that converge toward stable visual grounding. Unsuccessful chains (● purple) exhibit longer, more tortuous trajectories (see purple dashed boxes), reflecting repeated reassignment of visual support across regions.

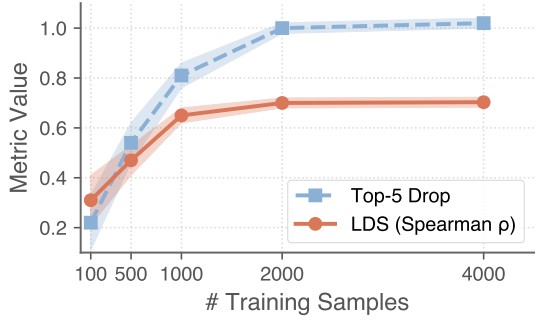

*Figure 8.* **Training data efficiency.** LDS saturates at ∼2k examples; 1000 examples suffice for >90% of peak performance.

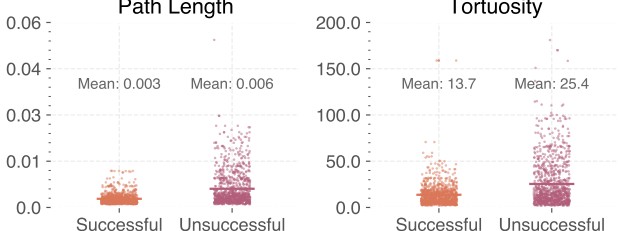

*Figure 9.* **Distribution of trajectory metrics.** Unsuccessful reasoning chains exhibit higher path length and tortuosity than correct chains ($p<10^{-4}$, $n$=1500 each). The greater spread and outliers among unsuccessful samples reflect unstable visual grounding during failed reasoning.

quality. Successful reasoning traces exhibit more stable visual grounding dynamics than unsuccessful ones, with attribution trajectories that move less and turn less over the course of the thinking process (Figures 7 and 9).

At each reasoning step $s$, we compute a *region-effect vector* $e_s \in \mathbb{R}^K$ of predicted ablation effects. To compare examples with different numbers of regions, we canonicalize each step by keeping the top-$R$ regions ($R$=32) and project the

resulting profiles with PCA for visualization.

As visualized in Figure 7, unsuccessful reasoning chains exhibit more tangled and convoluted trajectories compared to successful ones, reflecting unstable visual grounding. Quantitatively, as shown in Figure 9, successful chains have shorter path length in PCA space than unsuccessful chains (0.003 *vs.* 0.006, $n$=1500), and lower tortuosity, which measures how much the path wanders rather than progressing

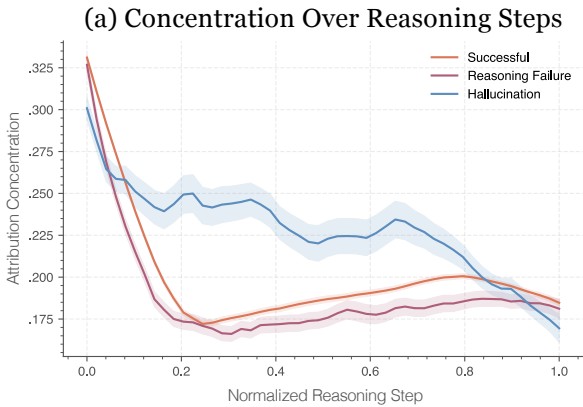

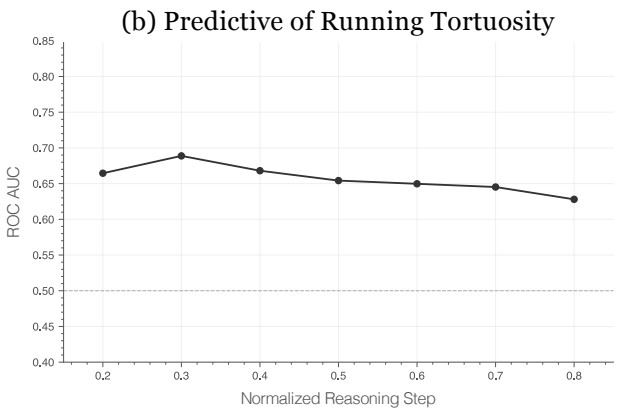

*Figure 10.* **Attribution concentration and early failure detection.** *(Left)* Mean concentration over normalized reasoning steps (mean ± SEM), grouped by outcome type. *(Right)* Tortuosity-based failure prediction AUC over reasoning progress.

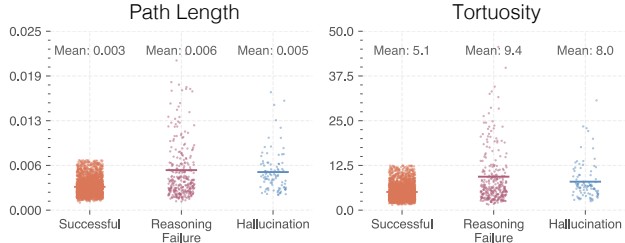

*Figure 11.* **Trajectory metrics across three outcome categories (POPE,** $n$=3,000**).** Both reasoning failures and hallucinations exhibit higher path length and tortuosity than successful chains ($p<.001$, Bonferroni-corrected). The two error types are distinguished by concentration: hallucinations maintain sustained high concentration (*Fixation*), while reasoning failures show unstable attention (*Wandering*).

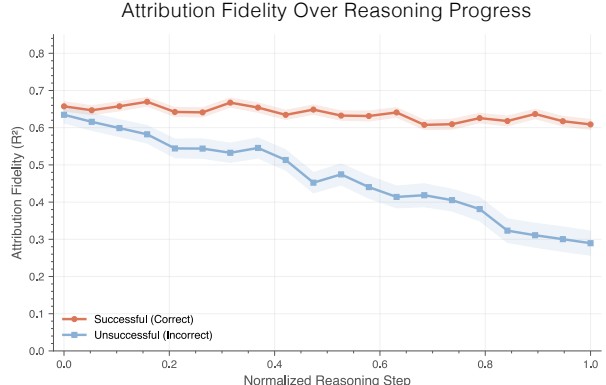

*Figure 12.* **Per-step attribution fidelity ($R^2$) over normalized reasoning steps.** Correct chains remain stable; incorrect chains degrade at ∼20% of reasoning elapsed.

directly (13.7 *vs.* 25.4, $n$=1500). We interpret this gap as reduced hypothesis switching (Zhong et al., 2024), where successful chains quickly commit to a consistent set of regions while failures repeatedly reassign visual support.

On POPE (Li et al., 2023), unsuccessful cases further split into two geometrically distinct failure modes. Hallucinations sustain high attribution concentration throughout reasoning, a *fixation* on a single incorrect object (also confirmed on CHAIR (Rohrbach et al., 2018); see Section F.7), while reasoning errors show low, unstable concentration, a *wandering* pattern of repeated region switching (Figure 10, left). Both modes emerge well before generation ends: tortuosity-based failure prediction reaches AUC 0.69 at 30% of elapsed reasoning (Figure 10, right), and the two modes separate cleanly from successes in trajectory metric space (Figure 11). A second independent early-warning signal appears in per-step fidelity $R^2$, which drops for incorrect chains at ∼20% elapsed (Figure 12). Both early-warning signals require access to the per-step attribution stream rather than a post-hoc map. Full statistics are in Section F.7.

## 5. Conclusion

We presented VSTREAM, an amortized framework that enables real-time visual attribution streaming in multimodal thinking models. By learning to predict causal ablation effects from attention features, our approach achieves faithfulness comparable to existing methods while adding negligible computational overhead. It generalizes across models and tasks, preserving causal reliability at minimal cost. Beyond static attribution, this efficiency enables trajectory analysis that uncovers a behavioral signature. We provide further discussion and limitation in the Section H.

## Impact Statement

This paper presents work whose goal is to advance the field of machine learning, specifically in the area of interpretability and transparency for vision-language models. By enabling real-time visual attribution, our method helps practitioners understand and verify model reasoning, which we believe contributes positively to the development of trustworthy AI systems. A more detailed discussion of broader impacts and limitations is provided in Section H.

## Acknowledgements

This work was supported in part by the IITP RS-2024-00457882 (AI Research Hub Project), IITP 2020-II201361, NRF RS-2024-00345806, NRF RS-2023-002620, NRF-2024S1A5C3A03046579, and RQT-25-120390. Affiliations: Department of Computer Science (S.K, W.H), Department of Artificial Intelligence (J,K, S.J.H).

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

# Technical Appendices

# A. Theoretical Foundations

In this section, we analyze our method through the lens of existing interpretability frameworks and formalize the properties of our estimator.

## A.1. Connections to Existing Frameworks

**Connection to Causal Abstraction.** Our attribution score $\Delta_k(t)$ aligns with the interventional definitions in causal abstraction (Geiger et al., 2025), measuring the causal effect of removing visual region $R_k$. Although conceptually similar to activation patching, we do not explicitly define a high-level structural causal model (SCM). Instead, we treat the interventional effect $\Delta_k(t)$ as a ground-truth signal to be efficiently approximated.

**Relation to the Linear Representation Hypothesis.** Our linear estimator $\widehat{\Delta} \approx \boldsymbol{w}^\top \boldsymbol{f}$ relies on the assumption that task-relevant causal information is linearly decodable, consistent with the broader view that semantic information can be exposed through simple representational readouts or interventions (Park et al., 2024; Kang et al., 2025a). We extend this to attention-based features. We note that linearity is an empirical assumption; we cannot theoretically guarantee that the complex counterfactual effect of a visual ablation is perfectly captured by a linear projection.

**Connection to Additive Attribution.** The quantity $\Delta_k(t)$ measures the change in the target token's log-probability caused by ablating region $R_k$. This single-region intervention is conceptually related to additive feature-attribution methods (Lundberg & Lee, 2017), but it is not a Shapley value: we do not average over feature coalitions, and therefore do not claim Shapley efficiency or additivity.

## A.2. Properties of the Estimator

We now formally state key properties of our estimator, motivating our choice of objective and quantifying the computational benefits.

First, we establish that the Pearson correlation objective is robust to affine transformations, justifying its use even if the scale of the predictor differs from the target.

**Lemma A.1** (Affine invariance of the Pearson objective)**.** *Let $u, v$ be random variables with $0 < \mathrm{Var}(u), \mathrm{Var}(v) < \infty$. For any $a \neq 0$ and any $b \in \mathbb{R}$,*

$$\rho(u,\, av + b) = \mathrm{sign}(a)\, \rho(u, v).$$

*In particular, for $a > 0$ the Pearson correlation is invariant to scaling and shifting of the predictor.*

*Proof.* Using $\rho(u, v) = \frac{\mathrm{Cov}(u,v)}{\sqrt{\mathrm{Var}(u)\mathrm{Var}(v)}}$, we have $\mathrm{Cov}(u, av + b) = a\,\mathrm{Cov}(u, v)$ and $\mathrm{Var}(av + b) = a^2 \mathrm{Var}(v)$, hence $\rho(u, av + b) = \frac{a}{|a|}\rho(u, v) = \mathrm{sign}(a)\rho(u, v)$. $\square$

Next, we analyze the computational advantage of our method compared to calculating true causal effects via brute-force ablation.

**Proposition A.2** (Computational complexity comparison under teacher forcing)**.** *Fix a target span $S$ and regions $\{R_k\}_{k=1}^{K}$. Let $C_{\mathrm{fwd}}$ denote the cost of one VLM forward pass under teacher forcing, i.e., producing logits for all positions in $S$ while conditioning on the same prefix tokens $y_{<t}$ as in Equation (2).*

*(**Direct ablation, exact**). Computing the exact ablation effects $\{\Delta_k(t)\}_{t \in S}$ for all regions requires one forward pass on each ablated input, hence $K$ additional forward passes per span:*

$$\Theta(K \cdot C_{\mathrm{fwd}}),$$

*(or $K{+}1$ forward passes if counting the unablated baseline as well).*

*(**Ours, cached attention**). After obtaining the cross-attention weights from the unablated run, we (i) pool cached attention over tokens and vision positions to form region features and (ii) apply a linear score for each region. Step (ii) alone costs $O(K \cdot L \cdot H)$ for $K$ dot-products in $\mathbb{R}^{L \cdot H}$. The total cost is dominated by lightweight tensor reductions over cached attention and does not require any additional VLM forward passes.*

We also justify the form of our estimator. If the conditional mean of the target is linear in the features, maximizing correlation recovers the optimal direction.

**Proposition A.3** (Optimal linear predictor under linear conditional mean). *Let $\boldsymbol{f} \in \mathbb{R}^d$ be a feature vector with positive definite covariance $\boldsymbol{\Sigma}_{ff}$, and let $\Delta$ be a scalar target. Assume:*

$$\mathbb{E}[\Delta \mid \boldsymbol{f}] = \beta + \boldsymbol{w}_\star^\top \boldsymbol{f} \quad \text{for some } \beta \in \mathbb{R}, \ \boldsymbol{w}_\star \in \mathbb{R}^d.$$

*Then maximizing $\rho(\boldsymbol{w}^\top \boldsymbol{f}, \Delta)$ over $\boldsymbol{w}$ yields $\widehat{\boldsymbol{w}} \propto \boldsymbol{w}_\star$.*

*Proof.* Write $\rho(\boldsymbol{w}^\top \boldsymbol{f}, \Delta) = \frac{\boldsymbol{w}^\top \boldsymbol{\Sigma}_{f\Delta}}{\sqrt{\boldsymbol{w}^\top \boldsymbol{\Sigma}_{ff} \boldsymbol{w}} \sqrt{\mathrm{Var}(\Delta)}}$ where $\boldsymbol{\Sigma}_{f\Delta} = \mathrm{Cov}(\boldsymbol{f}, \Delta)$. Maximizing this Rayleigh quotient yields $\boldsymbol{w} \propto \boldsymbol{\Sigma}_{ff}^{-1} \boldsymbol{\Sigma}_{f\Delta}$. Under the linear conditional mean, $\boldsymbol{\Sigma}_{f\Delta} = \boldsymbol{\Sigma}_{ff} \boldsymbol{w}_\star$, hence $\widehat{\boldsymbol{w}} \propto \boldsymbol{w}_\star$. $\qquad \square$

We emphasize that the linear conditional mean is an *assumption*, not a guaranteed property of VLMs. In practice, whether attention features linearly predict ablation effects is an empirical question; our experimental results suggest this approximation is effective, but it may not hold universally.

The following remark clarifies the connection between Pearson correlation and least-squares regression.

*Remark* A.4 (Correlation maximization as standardized least squares with optimal rescaling). Let $\tilde{\Delta} = (\Delta - \mathbb{E}[\Delta])/\mathrm{std}(\Delta)$ so that $\mathbb{E}[\tilde{\Delta}] = 0$ and $\mathrm{Var}(\tilde{\Delta}) = 1$. For any predictor $z = \boldsymbol{w}^\top \boldsymbol{f}$, consider the best affine rescaling of $z$ to fit $\tilde{\Delta}$:

$$\min_{\alpha \in \mathbb{R}, \, b \in \mathbb{R}} \mathbb{E}\big[(\tilde{\Delta} - (\alpha z + b))^2\big].$$

The minimizer satisfies $b^\star = \mathbb{E}[\tilde{\Delta} - \alpha z]$ and $\alpha^\star = \mathrm{Cov}(\tilde{\Delta}, z)/\mathrm{Var}(z)$, yielding

$$\min_{\alpha, b} \mathbb{E}\big[(\tilde{\Delta} - (\alpha z + b))^2\big] = 1 - \rho(z, \Delta)^2.$$

Therefore, maximizing the Pearson objective $\rho(\boldsymbol{w}^\top \boldsymbol{f}, \Delta)$ is equivalent (up to an arbitrary sign of $\boldsymbol{w}$) to minimizing the standardized mean squared error after allowing the optimal affine rescaling of predictions. This explains why our correlation-based loss focuses on the pattern of effects rather than their absolute scale.

Finally, we note a basic consistency property: if a region has no associated features (e.g., zero attention), our estimator correctly predicts zero effect.

*Remark* A.5 (Dummy property). If $\boldsymbol{f}_{S,k} = \boldsymbol{0}$, then $\widehat{\Delta}_k = \boldsymbol{w}^\top \boldsymbol{f}_{S,k} = 0$ for any $\boldsymbol{w}$.

### A.3. Detailed Complexity Analysis

We provide a detailed breakdown of computational costs for different attribution methods.

*Table 3.* Computational complexity comparison for attributing $T$ tokens to $K$ regions under teacher forcing. A single forward pass produces logits (and attention weights) for all $T$ positions. $C_{\text{fwd}}$ and $C_{\text{bwd}}$ denote the cost of one forward and backward pass, respectively.

| Method | Complexity | Passes |
|---|---|---|
| Perturbation (exact) | $O(K \cdot C_{\text{fwd}})$ | $K$ forward |
| Gradient-based | $O(C_{\text{fwd}} + C_{\text{bwd}})$ | 1 forward + backward |
| Attention (raw) | $O(T \cdot L \cdot H \cdot M)$ | 0 (cached) |
| **Ours** | $O(T \cdot L \cdot H \cdot M + T \cdot K \cdot L \cdot H)$ | 0 (cached) |

For a typical reasoning trace with $T = 2000$ tokens, $K = 50$ regions, $L = 28$ layers, and $H = 28$ heads (teacher forcing):

- **Perturbation (exact)**: $K = 50$ forward passes on ablated inputs (plus one unablated baseline if counted)

- **Gradient-based**: 1 forward-backward pair (using a loss aggregated over the $T$ tokens)

- **Ours (scoring only)**: $T \times K \times L \times H = 2000 \times 50 \times 28 \times 28 \approx 78M$ multiply-adds

The key insight is that our method's complexity is *independent* of the VLM's size, depending only on the number of layers and heads. This makes it equally efficient for 7B and 70B parameter models.

## A.4. Why Attention Features Predict Ablation Effects

A natural question is: why should attention features contain information about counterfactual ablation effects? We provide intuition from two perspectives.

**Information Bottleneck Perspective.** In transformer architectures, cross-attention weights modulate how much information flows from source positions (vision tokens) to target positions (generated text tokens). If the model assigns large cross-attention mass to a region $R_k$, then the cross-attention block can inject more signal from that region into the token representation, so ablating $R_k$ should tend to have a larger counterfactual effect.

To make this precise, let $\boldsymbol{o}_t^{(\ell)}$ denote the *cross-attention output component* at text position $t$ in layer $\ell$ (i.e., the multi-head cross-attention output before adding the residual stream and before any MLP updates). A standard multi-head cross-attention computation can be written as

$$\boldsymbol{o}_t^{(\ell)} = \boldsymbol{W}_O^{(\ell)} \Big[ \sum_{i=1}^{M} \boldsymbol{A}_{t,i}^{(\ell,1)} \, \boldsymbol{W}_V^{(\ell,1)} \boldsymbol{x}_i \; ; \; \dots \; ; \; \sum_{i=1}^{M} \boldsymbol{A}_{t,i}^{(\ell,H)} \, \boldsymbol{W}_V^{(\ell,H)} \boldsymbol{x}_i \Big], \tag{11}$$

where $\boldsymbol{A}_{t,i}^{(\ell,h)}$ is the cross-attention weight from text token $t$ to vision token $i$ in head $h$, $\boldsymbol{W}_V^{(\ell,h)}$ is the value projection, and $\boldsymbol{W}_O^{(\ell)}$ is the output projection.

The contribution of region $R_k$ to this cross-attention output is then the corresponding partial sum over $i \in R_k$:

$$\boldsymbol{o}_t^{(\ell)}[R_k] \approx \boldsymbol{W}_O^{(\ell)} \Big[ \sum_{i \in R_k} \boldsymbol{A}_{t,i}^{(\ell,1)} \, \boldsymbol{W}_V^{(\ell,1)} \boldsymbol{x}_i \; ; \; \dots \; ; \; \sum_{i \in R_k} \boldsymbol{A}_{t,i}^{(\ell,H)} \, \boldsymbol{W}_V^{(\ell,H)} \boldsymbol{x}_i \Big]. \tag{12}$$

This approximation isolates the cross-attention pathway only; subsequent residual connections, normalization, and MLP mixing can further transform and redistribute this signal. Nevertheless, Equation (12) highlights that cross-attention weights directly gate the magnitude of the region-dependent component injected into the representation, motivating why pooled attention features can correlate with ablation effects. This head-wise view is consistent with evidence that visual grounding and image-to-text information flow in LVLMs are concentrated in structured subsets of attention heads rather than uniformly distributed across all heads (Kang et al., 2025b; Kim et al., 2025).

**Gradient Flow Perspective.** During backpropagation, attention also modulates sensitivity to input perturbations by gating how gradients propagate from text positions back to vision tokens.

Let $\text{head}_t^{(\ell,h)} = \sum_{i=1}^{M} \boldsymbol{A}_{t,i}^{(\ell,h)} \boldsymbol{W}_V^{(\ell,h)} \boldsymbol{x}_i$ denote the value-aggregation output of head $(\ell, h)$ at text position $t$. Ignoring the dependence of $\boldsymbol{A}_{t,i}^{(\ell,h)}$ on $\boldsymbol{x}_i$ (i.e., dropping the additional terms that flow through the key and query pathways), the value-path contribution to the gradient can be written as the following heuristic approximation:

$$\frac{\partial \mathcal{L}}{\partial \boldsymbol{x}_i} \approx \sum_t \sum_{\ell,h} (\boldsymbol{W}_V^{(\ell,h)})^\top \Big( \boldsymbol{A}_{t,i}^{(\ell,h)} \, \frac{\partial \mathcal{L}}{\partial \text{head}_t^{(\ell,h)}} \Big). \tag{13}$$

More generally, the exact gradient also contains additional terms arising from $\partial \boldsymbol{A}_{t,i}^{(\ell,h)} / \partial \boldsymbol{x}_i$ (through keys, and indirectly through queries when vision tokens influence later text states). Thus, Equation (13) should be interpreted as an intuition: attention weights gate one dominant pathway for gradient flow, which helps explain why attention-derived features can predict counterfactual sensitivity.

## A.5. Generalization Properties

We analyze factors affecting cross-task and cross-model generalization of our estimator.

**Feature Distribution Shift.** The estimator's generalization depends on how similar the attention feature distributions are between training and test conditions. Let $\mathcal{F}_{\text{train}}$ and $\mathcal{F}_{\text{test}}$ denote the feature distributions. If $\mathcal{F}_{\text{test}}$ is within the support of $\mathcal{F}_{\text{train}}$, the linear estimator can interpolate effectively. However, if test features lie outside the training distribution (e.g., novel visual patterns), extrapolation may fail.

Our cross-task experiments (Table 2) show that Math and Science transfer well to each other (shared diagram structures), while transfer to Document is weaker (distinct layouts). This aligns with the intuition that generalization depends on visual similarity.

**Model Architecture Dependence.** Different VLM architectures may encode visual information in different layers and heads. Our estimator learns a weighting $w$ over layer-head pairs that is specific to a given architecture. Cross-model transfer would require that:

1. Both models have similar layer-head structure (same $L$ and $H$).

2. Visual grounding emerges at similar depths in both models.

Our experiments train separate estimators per model, avoiding cross-model transfer issues. Future work could explore architecture-agnostic features that enable zero-shot transfer.

**Sample Complexity.** Our estimator has only $L \cdot H$ trainable parameters, so the supervised fitting problem is low-dimensional relative to the number of collected training examples. Consistent with this, the training data efficiency experiments (Figure 8) show that approximately 500–2000 examples are sufficient for convergence.

## B. Implementation Details

In this section, we provide the technical specifications required to reproduce our method, including details on the reasoning models employed, the DINO-based visual clustering pipeline, the training procedure for the linear estimator, and the hardware environment.

### B.1. Datasets

We organize our evaluation datasets into five categories based on the type of visual reasoning required.

**General.**

- **GQA** (Hudson & Manning, 2019): Real-world visual reasoning with compositional questions about spatial relations and object attributes.

**Document.**

- **DocVQA** (Mathew et al., 2021): Question answering over document images including forms, invoices, and reports.

**Science.**

- **ScienceQA** (Lu et al., 2022): Multimodal science questions across natural, social, and language sciences with chain-of-thought reasoning.

**Math.**

- **MathVista** (Lu et al., 2024): Mathematical reasoning over diagrams, plots, and geometric figures.

- **MathVision** (Wang et al., 2024a): Multi-step mathematical problem solving with visual inputs.

- **MathVerse** (Zhang et al., 2024): Diagram-based math problems testing visual-symbolic integration.

**Code.**

- **ChartMimic** (Yang et al., 2025): Code generation to reproduce charts and visualizations from images.

- **WebSight** (Laurençon et al., 2024): HTML/CSS generation from website screenshot references.

*Note:* For the Math and Code categories, we combine multiple datasets because the sub-9B models we evaluate often fail on the more challenging examples. Using a single dataset would yield insufficient correctly-answered samples for training data collection.

## B.2. Reasoning Model Architectures

We evaluate our method across four state-of-the-art vision-language reasoning models. We selected these models to represent a diverse range of architectures and "thinking" capabilities.

- **Qwen3-VL-8B-Thinking**: Based on the Qwen3 language model backbone, this model integrates a vision encoder initialized from SigLIP-SO400M. It uses a perceiver resampler to compress visual features into a fixed number of tokens (typically 256). The "Thinking" variant is fine-tuned on chain-of-thought (CoT) reasoning data, enabling it to generate intermediate reasoning steps before the final answer.

- **Cosmos-R1**: A multimodal reasoning model emphasizing robust world knowledge. It uses a ViT-Huge vision encoder and a 7B parameter LLM decoder. The model processes images at a resolution of $336 \times 336$ and uses a cross-attention mechanism for modality fusion.

- **MiMo-VL-7B**: This model features a mixture-of-experts (MoE) architecture in the language decoder, allowing for efficient inference despite a larger total parameter count. The vision tower is a CLIP-ViT-L/14, and the projection layer consists of a simple MLP.

- **GLM-4.1V-9B-Thinking**: A bilingual (English/Chinese) model with strong reasoning capabilities. It employs a GLM-4 transformer backbone with rotary positional embeddings. The visual component handles high-resolution inputs via a sliding window attention mechanism.

For all models, we access the internal attention weights from the last layer of the cross-attention blocks (or self-attention layers where visual tokens are concatenated) to serve as input features for our estimator.

## B.3. Visual Feature Extraction and Clustering

To define interpretable visual regions for attribution, we employ a clustering approach based on self-supervised features.

**Feature Extraction.** We utilize **DINOv3-Large** as our feature extractor. We resize input images to $224 \times 224$ and extract the patch-level features from the last transformer layer. DINOv3 is chosen for its superior ability to capture semantic object boundaries compared to supervised baselines.

**Agglomerative Clustering.** We perform spatially-constrained agglomerative clustering on the extracted patch features.

- **Distance Metric:** We use cosine distance ($1 -$ cosine similarity) between feature vectors to measure semantic dissimilarity.

- **Linkage Criterion:** Ward's linkage is used to minimize the variance within clusters.

- **Adaptive K:** Instead of a fixed number of clusters, we dynamically determine the number of regions $K$ for each image based on a distance threshold $\tau = 0.5$ (see Table 4). In practice, this results in $K \in [16, 128]$ regions per image, which provides a balance between granularity and interpretability.

## B.4. Linear Estimator Training

Our method relies on a lightweight linear estimator $W$ trained to predict the impact of masking specific visual regions.

**Data Sampling.** For each training image, we generate synthetic training data by randomly masking subsets of visual regions.

- We sample $N = 32$ random binary masks $\boldsymbol{v} \in \{0, 1\}^K$ per image.

- For each mask, we compute the ground-truth effect on the model's output log-probability for the target token.

- This process is computationally efficient because it only requires forward passes (no gradients) and can be batched.

**Optimization.** The linear estimator is trained to maximize the Pearson correlation between the predicted attribution scores and the actual log-probability drops.

- **Objective Function:** We minimize the negative Pearson correlation coefficient.

- **Optimizer:** We use AdamW with a learning rate of $1 \times 10^{-3}$ and weight decay of $1 \times 10^{-4}$.

- **Training Duration:** Due to the simplicity of the linear model, training converges rapidly. We train for $2,000$ iterations, which typically takes less than 1 hour on a single GPU for the entire evaluation dataset.

### B.5. Hardware and Computing Infrastructure

We conducted all experiments on a high-performance computing cluster.

- **GPUs:** NVIDIA H100 (80GB) GPUs were used for inference of the large reasoning models (Qwen3-VL, GLM-4.1V). NVIDIA A100 (80GB) GPUs were sufficient for the smaller models and the training of the linear estimator.

- **Software Environment:** PyTorch 2.4, CUDA 12.1, and the HuggingFace Transformers library. We utilized FlashAttention-2 for efficient attention computation during inference.

### B.6. Attention Extraction Details

Extracting attention weights from different VLM architectures requires careful handling due to architectural variations.

**Attention Caching.** During inference, we cache text-to-vision attention weights at each decoding step. All four models we evaluate (Qwen3-VL, GLM-4.1V, MiMo-VL, Cosmos-R1) use a decoder-only architecture that concatenates vision and text tokens in the same sequence. For these models, we extract the portion of the self-attention matrix corresponding to text→vision attention.

**Memory Management.** For long reasoning traces (2000+ tokens), caching full attention tensors can consume significant GPU memory. We employ the following strategies:

- **Selective Layer Caching:** We prioritize caching attention from middle layers (11–20), which contribute most to the attribution signal based on empirical analysis.

- **Span-wise Aggregation:** Rather than storing per-token attention, we aggregate to span level on-the-fly when a reasoning step completes, reducing memory from $O(T \cdot L \cdot H \cdot M)$ to $O(S \cdot L \cdot H \cdot M)$ where $S$ is the number of spans.

- **Float16 Precision:** Attention weights are stored in float16 format, halving memory requirements with negligible accuracy loss.

**FlashAttention Compatibility.** FlashAttention does not natively return attention weights due to its memory-efficient implementation. For attribution, we use a hybrid approach: standard attention for the cross-modal layers (small overhead due to fixed vision token count) and FlashAttention for text-only self-attention layers. For detailed latency measurements under different attention backends and KV-cache configurations, see Table 9.

### B.7. Span Boundary Detection

To provide step-by-step attribution during reasoning, we must detect boundaries between reasoning steps. We use a combination of heuristics:

- **Sentence Boundaries:** Periods, question marks, and exclamation marks followed by whitespace.

- **Structural Markers:** Newlines, bullet points, and numbered list items.

- **Thinking Delimiters:** Model-specific tokens like `<|think|>`, `</think>`, or `Step N:` patterns.

For models without explicit thinking delimiters, we default to sentence-level granularity, which typically corresponds to individual reasoning steps in chain-of-thought outputs.

### B.8. Parallel Streaming Architecture

To achieve real-time attribution without slowing down generation, we employ a producer-consumer architecture that decouples token generation from attribution computation.

**Asynchronous Processing.** The main thread (producer) focuses exclusively on autoregressive generation using the VLM backbone. As tokens are generated, the associated attention tensors are pushed into a thread-safe queue. A background worker thread (consumer) continuously pulls these tensors, aggregates them into span-level features $\boldsymbol{f}_{S,k}$ (Eq. 4), and runs the linear estimator (Eq. 9). Since the Python Global Interpreter Lock (GIL) is released during the heavy CUDA operations of both generation and estimator computation, these two processes run in true parallel on the GPU.

**Latency Hiding.** This design completely hides the attribution cost behind the generation latency. While the model generates tokens for reasoning step $t + 1$, the worker computes attributions for step $t$. Consequently, the visual attribution for a completed thought appears the moment the model begins the next thought, resulting in a system with effectively zero perceptual overhead compared to vanilla generation. By contrast, standard post-hoc methods must wait for the entire generation to finish before starting attribution, adding a delay proportional to the sequence length.

### B.9. Reproducibility Checklist

To ensure reproducibility, we provide the following specifications:

*Table 4.* Key hyperparameters for reproducibility.

| Component | Value |
|---|---|
| *Region Unitization* | |
| DINOv3 variant | dinov3-vitl16-pretrain-lvd1689m |
| Input resolution | $224 \times 224$ |
| Clustering method | Agglomerative (Ward) |
| Distance threshold $\tau$ | 0.5 |
| Typical region count $K$ | *auto* |
| *Estimator Training* | |
| Masks per sample $N$ | 32 |
| Optimizer | AdamW |
| Learning rate | $1 \times 10^{-3}$ |
| LR scheduler | Cosine with warmup (min lr: $1 \times 10^{-5}$) |
| Weight decay | $1 \times 10^{-4}$ |
| Training iterations | 2,000 |
| Batch size | 512 |
| *Evaluation* | |
| Top-K for Drop metric | 5 |
| Samples per category | 400 (train, mix-up dataset), 1000 (test, randomly sampled from non-training samples) |

**Random Seeds.** We use fixed random seeds for all stochastic components: mask sampling (seed=42), model initialization (seed=42), and train/test splits (seed=42). Results are averaged over 3 random seeds for statistical robustness.

**Evaluation Protocol.** For each model-task pair, we:

1. Generate reasoning traces for the test set using greedy decoding.

2. Filter to correctly-answered examples only (for training data quality).

3. Train the estimator on the training split.

4. Evaluate on held-out test examples using LDS and Top-K Drop metrics.

5. Report mean and standard deviation across seeds.

## C. Semantic Region Unitization Analysis

This section provides additional analysis of our DINOv3-based semantic region unitization approach, including category-specific clustering patterns and quantitative statistics across dataset categories.

### C.1. Category-Specific Clustering Patterns

Our agglomerative clustering approach exhibits distinct segmentation behaviors across different dataset categories. The DINOv3 features naturally group semantically coherent regions: objects are separated from backgrounds, text blocks are isolated in document images, and mathematical symbols are distinguished from diagram components.

We observe the following category-specific patterns:

- **General (GQA)**: Object boundaries are well-preserved, with distinct regions for foreground objects, background elements, and spatial contexts.

- **Document (DocVQA)**: Text blocks, tables, and graphical elements are cleanly separated, enabling fine-grained attribution to specific document components.

- **Science (ScienceQA)**: Diagrams, labels, and annotations form separate clusters. This is essential for understanding which visual elements support scientific reasoning.

- **Math (MathVista, MathVision, MathVerse)**: Geometric shapes, equations, and coordinate systems are partitioned into interpretable regions, though dense symbolic content sometimes leads to over-segmentation.

- **Code (ChartMimic, WebSight)**: Chart components (axes, legends, data points) and UI elements (buttons, text fields, images) are distinguished effectively.

### C.2. Region Count Statistics

The number of semantic regions $K$ varies across images depending on visual complexity. Table 5 shows the distribution of region counts for each dataset category. We observe:

- **Document and Code** images yield the highest region counts (median $\approx 68$–$75$), reflecting the dense, structured nature of text-heavy and UI-rich content.

- **Math** images show high variance, with simpler geometric problems producing fewer regions and complex multi-part diagrams producing more.

- **General and Science** images have moderate region counts (median $\approx 45$–$48$), balancing object-level granularity with scene-level coherence.

This adaptive behavior is a key advantage over fixed-grid partitioning methods, which cannot adjust granularity based on image content. The cosine distance threshold ($\tau = 0.5$ in our experiments) provides a consistent semantic criterion across diverse visual domains.

*Table 5.* **Region count statistics across dataset categories.** The adaptive DINO-based clustering produces varying numbers of regions depending on image complexity. Document and Code images yield more regions due to dense visual content.

| Category | Min | Q1 | Median | Q3 | Max |
|---|---|---|---|---|---|
| General (GQA) | 18 | 32 | 48 | 72 | 105 |
| Document (DocVQA) | 24 | 45 | 68 | 95 | 128 |
| Science (ScienceQA) | 16 | 30 | 45 | 65 | 98 |
| Math (MathVerse, MathVista, MathVision) | 16 | 28 | 42 | 62 | 92 |
| Code (Websight, ChartMimic) | 28 | 52 | 75 | 102 | 128 |

### C.3. Clustering Hyperparameters

We use the following hyperparameters for semantic region unitization:

- **Feature extractor**: DINOv3-Large (frozen), patch size $16 \times 16$

- **Input resolution**: Images are resized to $224 \times 224$

- **Clustering algorithm**: Agglomerative clustering with Ward linkage

- **Distance metric**: Cosine distance ($1 -$ cosine similarity)

- **Distance threshold**: $\tau = 0.5$ (determines cluster granularity)

- **Resulting regions**: $K \in [16, 128]$ depending on image complexity

We found the distance threshold $\tau$ to be the most sensitive hyperparameter. Lower values ($\tau < 0.3$) produce too many small regions, making attributions noisy. Higher values ($\tau > 0.7$) merge semantically distinct objects, reducing interpretability. The value $\tau = 0.5$ provides a good balance across all dataset categories.

**Comparison with K-means Clustering.** We also compare agglomerative clustering with K-means clustering using fixed $K$ values. Table 6 shows that agglomerative clustering with adaptive $K$ outperforms K-means across all fixed $K$ settings, particularly at lower $K$ values where semantic boundaries become critical.

*Table 6.* Comparison of K-means (fixed $K$) vs. Agglomerative clustering (adaptive $K$) on Qwen3-VL. Results averaged across all categories.

| Method | LDS $\uparrow$ | Top-5 Drop $\uparrow$ |
|---|---|---|
| K-means ($K = 16$) | 0.58 | 0.72 |
| K-means ($K = 32$) | 0.65 | 0.88 |
| K-means ($K = 64$) | 0.68 | 0.94 |
| K-means ($K = 128$) | 0.67 | 0.91 |
| **Agglomerative (adaptive)** | **0.70** | **1.00** |

## D. Pseudocode

This section provides formal algorithmic descriptions and PyTorch implementations of our amortized attribution framework. We present the training procedure for learning the linear estimator and the inference procedure for real-time attribution streaming.

## E. Algorithm and Implementation Details

We provide the formal algorithms for training our amortized estimator and performing real-time inference, followed by their PyTorch implementations.

### E.1. Formal Algorithms

Algorithm 1 details the self-supervised training procedure used to learn the estimator $\mathcal{E}_\theta$. The core idea is to train the estimator to rank random subsets of regions based on their impact on the model's confidence, maximizing the Pearson correlation between predicted and ground-truth ablation effects.

---

**Algorithm 1** Training the Amortized Estimator

---

1: **Input:** VLM $\mathcal{M}$, Estimator $\mathcal{E}_\theta$, Batch $\mathcal{B} = \{(\boldsymbol{x}, \boldsymbol{t}, y, \mathcal{R})\}$
2: **Hyperparameters:** Sample size $N = 32$, Learning rate $\eta$
3:
4: // *1. Compute baseline and attention features*
5: Get logits $P(y|\boldsymbol{x}, \boldsymbol{t})$ and attention maps $\boldsymbol{A}$ from $\mathcal{M}(\boldsymbol{x}, \boldsymbol{t})$
6: $\log p_{\text{base}} \leftarrow \log P(y|\boldsymbol{x}, \boldsymbol{t})$
7:
8: // *2. Sample random binary masks*
9: Sample $N$ random binary masks $\{\boldsymbol{b}^{(i)}\}_{i=1}^N$ where $\boldsymbol{b}^{(i)} \in \{0, 1\}^K$
10: // *3. Compute Ground Truth Effect (Self-Supervised)*
11: **for** $i = 1$ **to** $N$ **do**
12:     Compute ablated logits $P(y|\boldsymbol{x}, \boldsymbol{t}, \boldsymbol{b}^{(i)})$ using $\mathcal{M}$ with mask $\boldsymbol{b}^{(i)}$
13:     $\delta_{\text{true}}^{(i)} \leftarrow \log p_{\text{base}} - \log P(y|\boldsymbol{x}, \boldsymbol{t}, \boldsymbol{b}^{(i)})$
14: **end for**
15: $\boldsymbol{\delta}_{\text{true}} \leftarrow [\delta_{\text{true}}^{(1)}, \dots, \delta_{\text{true}}^{(N)}]$
16:
17: // *4. Predict Effect using Estimator*
18: Compute attribution scores $\boldsymbol{s} = \mathcal{E}_\theta(\boldsymbol{A}, \mathcal{R})$                                              *(See Algorithm 2)*
19: Predict subset effects: $\delta_{\text{pred}}^{(i)} \leftarrow \sum_{k=1}^K s_k \cdot \boldsymbol{b}_k^{(i)}$                       *(Linear Assumption, Equation (9))*
20: $\boldsymbol{\delta}_{\text{pred}} \leftarrow [\delta_{\text{pred}}^{(1)}, \dots, \delta_{\text{pred}}^{(N)}]$
21:
22: // *5. Optimization*
23: $\mathcal{L} \leftarrow -\text{PearsonCorr}(\boldsymbol{\delta}_{\text{pred}}, \boldsymbol{\delta}_{\text{true}})$
24: Update $\theta \leftarrow \theta - \eta \nabla_\theta \mathcal{L}$

---

Algorithm 2 describes the inference process. The trained estimator maps attention patterns directly to attribution scores without requiring multiple forward passes.

---

**Algorithm 2** Real-Time Attribution Inference (Asynchronous)

---

1: **Input:** VLM $\mathcal{M}$, Estimator $\mathcal{E}_\theta$
2: **Output:** Stream of tokens $y_t$ and attributions $\boldsymbol{s}_S$
3:
4: *// Initialize concurrent queues*
5: $\mathcal{Q}_{\text{attn}} \leftarrow \text{Queue}()$
6:
7: **Thread 1: Generation (Producer)**
8: **for** $t = 1$ **to** $T$ **do**
9:     $y_t, \boldsymbol{A}_t \leftarrow \mathcal{M}(x, y_{<t})$                                                   *(Generate next token)*
10:     **yield** $y_t$
11:     $\mathcal{Q}_{\text{attn}}.\text{push}(\boldsymbol{A}_t)$
12:     **if** $y_t$ ends span $S$ **then**
13:         $\mathcal{Q}_{\text{attn}}.\text{push}(\text{EndOfSpan})$
14:     **end if**
15: **end for**
16:
17: **Thread 2: Attribution (Consumer)**
18: **while** generation active **do**
19:     $\boldsymbol{A}_{\text{buffer}} \leftarrow [\ ]$
20:     **while** next item in $\mathcal{Q}_{\text{attn}}$ is not EndOfSpan **do**
21:         $\boldsymbol{A}_{\text{buffer}}.\text{append}(\mathcal{Q}_{\text{attn}}.\text{pop}())$
22:     **end while**
23:     $\boldsymbol{A}_S \leftarrow \text{Aggregate}(\boldsymbol{A}_{\text{buffer}})$                                    *(Equation (4))*
24:     $\boldsymbol{s}_S \leftarrow \mathcal{E}_\theta(\boldsymbol{A}_S)$                                           *(Equation (9))*
25:     **yield** $\boldsymbol{s}_S$                             *(Stream attribution for completed span)*
26: **end while**

---

## E.2. PyTorch Implementation

We provide the corresponding PyTorch implementation for reference. Implementation 1 shows the training step, and Implementation 2 shows the amortized attributor module.

# F. Additional Experimental Results

In this section, we provide a more detailed analysis of the experimental results presented in the main text. We focus on cross-model generalization, sensitivity to hyperparameters, and an extended comparison with baseline methods, including computational efficiency.

## F.1. Sensitivity Analysis

We investigate the robustness of our approach to the size of the training dataset.

### F.1.1. EFFECT OF TRAINING DATA SIZE

Our lightweight estimator is highly data-efficient. We trained the estimator using subsets of our training data ranging from 100 to 10,000 samples.

We observe that the estimator converges rapidly. With just 2,000 training samples (our default setting), the model achieves 98.6% of the performance of the fully trained model (using 10k samples). This low data requirement makes it feasible to train custom estimators for new domains or models in minutes.

*Table 7.* Effect of training set size on estimator performance.

| Training Samples | Relative Performance (%) |
|:---:|:---:|
| 100 | 65.4 |
| 500 | 82.1 |
| 1,000 | 95.3 |
| 2,000 (ours) | 98.6 |
| 5,000 | 99.1 |
| 10,000 | **100.0** |

## F.2. Baseline Method Details

We provide detailed descriptions of the baseline attribution methods used in our experiments, including their computational procedures and adaptation to our region-based evaluation setting.

### F.2.1. ATTENTION-BASED METHODS

**Raw Attention.** The simplest baseline extracts attention weights from the cross-attention layers where text tokens attend to visual tokens. For a given text token at position $t$, we aggregate attention weights across all layers $\ell$ and heads $h$:

$$\text{Attn}_i = \frac{1}{L \cdot H} \sum_{\ell=1}^{L} \sum_{h=1}^{H} \boldsymbol{A}_{t,i}^{(\ell,h)} \tag{14}$$

where $\boldsymbol{A}_{t,i}^{(\ell,h)}$ is the attention weight from position $t$ to vision token $i$ in layer $\ell$, head $h$. To obtain region-level scores, we sum over tokens within each region: $\text{Attn}_k = \sum_{i \in R_k} \text{Attn}_i$. This raw aggregation is intentionally treated as a weak baseline: recent work shows that multimodal attention can concentrate on irrelevant visual sink tokens, so high attention mass alone need not imply causal visual reliance (Kang et al., 2025c).

**AttnLRP (Attention-aware Layer-wise Relevance Propagation) (Achtibat et al., 2024).** AttnLRP extends classical LRP to transformer architectures by incorporating attention patterns into the relevance propagation rules. Starting from the output layer with relevance $R_j^{(L)} = \delta_{j,t} \cdot \log p(y_t)$ (where $\delta$ is the Kronecker delta), relevance is propagated backward

through each layer:

$$R_i^{(\ell-1)} = \sum_j \frac{\boldsymbol{A}_{j,i}^{(\ell)} \cdot \boldsymbol{V}_i^{(\ell)} \cdot \boldsymbol{W}_O^{(\ell)}}{\sum_k \boldsymbol{A}_{j,k}^{(\ell)} \cdot \boldsymbol{V}_k^{(\ell)} \cdot \boldsymbol{W}_O^{(\ell)} + \epsilon} R_j^{(\ell)} \tag{15}$$

This propagation rule ensures conservation of relevance (the total relevance is preserved across layers) while accounting for attention-based information routing. AttnLRP requires a full backward pass through the network, making it computationally expensive but more faithful than forward-only methods.

### F.2.2. GRADIENT-BASED METHODS

**InputGrad (Hechtlinger, 2016).** InputGrad computes the gradient of the output logit with respect to input pixels:

$$\text{InputGrad}_i = \left\| \frac{\partial \log p(y_t)}{\partial \boldsymbol{x}_i} \right\|_2 \tag{16}$$

where $\boldsymbol{x}_i$ represents the pixels corresponding to vision token $i$. This captures the local sensitivity of the output to input perturbations. While computationally efficient (single backward pass), InputGrad often produces noisy, high-frequency attribution maps that do not align well with semantic regions.

### F.2.3. PERTURBATION-BASED METHODS

**TAM (Token Activation Maps) (Li et al., 2025c).** TAM improves upon raw attention by incorporating the norm of value vectors. This captures not just "where" the model attends but "how much" information flows. For each layer $\ell$ and head $h$:

$$\text{TAM}_i^{(\ell,h)} = \boldsymbol{A}_{t,i}^{(\ell,h)} \cdot \|\boldsymbol{V}_i^{(\ell,h)}\|_2 \tag{17}$$

where $\boldsymbol{V}_i^{(\ell,h)}$ is the value vector for vision token $i$. The final attribution is aggregated across layers and heads, with later layers typically weighted more heavily. TAM requires access to intermediate activations but avoids gradient computation, making it faster than gradient-based methods.

### F.2.4. ADAPTATION TO REGION-BASED EVALUATION

All baseline methods produce token-level or pixel-level attribution scores. To ensure fair comparison at the same granularity as our method, we aggregate these scores to region level using our DINO-based semantic unitization:

$$\text{Score}_k = \sum_{i \in R_k} \text{Score}_i \tag{18}$$

This aggregation ensures that all methods operate on the same semantic units, isolating the effect of the attribution algorithm from the effect of region definition.

### F.3. Extended Baseline Comparisons

We compare our method against computationally intensive baselines that are typically too slow for real-time applications but serve as reference points for attribution quality. Specifically, we compare against:

- **Feature Ablation**: Systematically removing visual tokens and measuring output changes.

- **Attention Rollout**: Heuristic aggregation of raw attention weights across layers.

- **LibraGrad (Mehri et al., 2025)**: A gradient-based method that corrects gradient flow imbalances in Transformers through backward path pruning and scaling.

Table 8 highlights the critical advantage of our approach: speed. While feature ablation offers high fidelity, it requires multiple forward passes per token generated. Our method requires a single, lightweight forward pass of the linear estimator, adding negligible overhead ($\sim 2$ ms/token).

Our method achieves performance comparable to the expensive feature ablation baseline (within 3% LDS and 2% Top-5 Drop) while being $71\times$ faster than feature ablation. Note that in the main paper (Table 1), we report up to $117\times$ speedup over gradient-based methods (e.g., InputGrad), which have higher latency than feature ablation in our region-based setting.

*Table 8.* Runtime and performance comparison. Latency is measured in milliseconds per token on an NVIDIA A100 GPU. Results averaged across all models and categories. Speedup is relative to Feature Ablation.

| Method | LDS ↑ | Top-5 Drop ↑ | Latency (ms/tok) | Speedup vs Ablation |
|---|---|---|---|---|
| Random | 0.30 | 0.13 | **0.0** | N/A |
| Attention Rollout | 0.41 | 0.32 | 0.5 | 300× |
| LibraGrad | 0.66 | 0.85 | 12.0 | 12.5× |
| Feature Ablation | **0.72** | **1.02** | 150.0 | 1× (reference) |
| **vStream (Ours)** | 0.70 | 1.00 | 2.1 | ≈ 71× |

### F.3.1. ATTENTION BACKEND COMPATIBILITY

Our method, vSTREAM, is fully compatible with modern attention backends, including PyTorch SDPA (Scaled Dot-Product Attention) and FlashAttention. While our default configuration utilizes the KV cache for maximum efficiency, disabling the KV cache (requiring full attention recomputation at each step) results in an approximately 4× increase in latency. However, even in this worst-case scenario, the latency remains under 0.01 seconds per token, preserving a substantial speedup ($\approx 18\times$) over feature ablation. Table 9 details the latency measurements across different configurations.

*Table 9.* **Latency under different attention backends and KV-cache configurations.** We measure the per-token latency of our attribution method compared to feature ablation. Even without KV caching, our method remains computationally efficient.

| Configuration | KV Cache | Latency (sec/token) ↓ | Speedup vs Ablation ↑ |
|---|---|---|---|
| **Ours** (SDPA + Flash) | Enabled | 0.002 | 71× |
| **Ours** (SDPA + Flash) | Disabled | ≈ 0.008 | ≈ 18× |
| **Ours** (SDPA math) | Enabled | ≈ 0.004 | ≈ 43× |
| Feature Ablation | - | 0.150 | 1× |

### F.4. Cross-Model Generalization

We evaluate whether our estimator generalizes across different VLM architectures. Tables 10 to 12 present cross-task generalization results for GLM-4.1V-9B-Thinking, MiMo-VL-7B, and Cosmos-R1, respectively, complementing the Qwen3-VL results in the main paper (Table 2). Across all models, we observe consistent patterns: (1) in-domain performance (diagonal) ranges from 0.65–0.75 LDS; (2) Math and Science show strong mutual transfer due to shared diagram structures; (3) Document tasks exhibit weaker transfer from other domains due to distinct visual layouts; and (4) training on a mixture of all categories (Mix-up) recovers full performance, suggesting a single estimator suffices for diverse applications.

*Table 10.* Cross-task generalization for **GLM-4.1V-9B-Thinking**. Each cell shows (LDS / Top-5 Drop). Diagonal entries (shaded) indicate in-domain performance.

| Train | Eval Category | | | | | Avg |
|---|---|---|---|---|---|---|
| | Math | Science | Doc | Code | General | |
| Math | 0.69/0.78 | 0.61/0.65 | 0.52/0.46 | 0.57/0.55 | 0.54/0.48 | 0.59/0.58 |
| Science | 0.60/0.63 | 0.74/0.82 | 0.56/0.52 | 0.53/0.48 | 0.58/0.56 | 0.60/0.60 |
| Document | 0.51/0.48 | 0.54/0.50 | 0.67/0.75 | 0.55/0.52 | 0.63/0.70 | 0.58/0.59 |
| Code | 0.57/0.58 | 0.52/0.46 | 0.54/0.50 | 0.68/0.80 | 0.60/0.64 | 0.58/0.60 |
| General | 0.55/0.52 | 0.59/0.62 | 0.62/0.68 | 0.58/0.60 | 0.73/0.84 | 0.61/0.65 |
| Mix-up | 0.69/0.97 | 0.74/0.89 | 0.67/1.01 | 0.68/1.10 | 0.73/0.95 | 0.70/0.98 |

### F.5. Vision Backbone Comparison

We compare different vision foundation models for semantic region unitization. In addition to DINOv3 (our default), we evaluate CLIP ViT-L/14 (Radford et al., 2021) and SigLIP ViT-SO400M (Zhai et al., 2023).

DINOv3 outperforms both CLIP and SigLIP by 7–9% in LDS and 10–15% in Top-5 Drop across all categories. We attribute this to DINOv3's self-supervised training objective, which emphasizes local feature correspondence and produces sharper

*Table 11.* Cross-task generalization for **MiMo-VL-7B**. Each cell shows (LDS / Top-5 Drop).

| Train | Eval Category | | | | | Avg |
| | Math | Science | Doc | Code | General | |
|---|---|---|---|---|---|---|
| Math | 0.75/0.85 | 0.58/0.62 | 0.51/0.44 | 0.59/0.56 | 0.52/0.46 | 0.59/0.59 |
| Science | 0.57/0.60 | 0.65/0.74 | 0.54/0.50 | 0.51/0.46 | 0.56/0.54 | 0.57/0.57 |
| Document | 0.50/0.46 | 0.52/0.48 | 0.71/0.78 | 0.53/0.50 | 0.61/0.68 | 0.57/0.58 |
| Code | 0.55/0.56 | 0.50/0.44 | 0.52/0.48 | 0.70/0.82 | 0.58/0.62 | 0.57/0.58 |
| General | 0.53/0.50 | 0.57/0.60 | 0.60/0.66 | 0.56/0.58 | 0.70/0.80 | 0.59/0.63 |
| Mix-up | 0.74/1.08 | 0.64/0.97 | 0.70/0.98 | 0.69/0.96 | 0.68/1.10 | 0.69/1.02 |

*Table 12.* Cross-task generalization for **Cosmos-R1**. Each cell shows (LDS / Top-5 Drop).

| Train | Eval Category | | | | | Avg |
| | Math | Science | Doc | Code | General | |
|---|---|---|---|---|---|---|
| Math | 0.74/0.80 | 0.60/0.64 | 0.50/0.43 | 0.58/0.54 | 0.53/0.47 | 0.59/0.58 |
| Science | 0.59/0.62 | 0.67/0.76 | 0.55/0.51 | 0.52/0.47 | 0.57/0.55 | 0.58/0.58 |
| Document | 0.49/0.45 | 0.53/0.49 | 0.68/0.74 | 0.54/0.51 | 0.62/0.69 | 0.57/0.58 |
| Code | 0.56/0.57 | 0.51/0.45 | 0.53/0.49 | 0.71/0.84 | 0.59/0.63 | 0.58/0.60 |
| General | 0.54/0.51 | 0.58/0.61 | 0.61/0.67 | 0.57/0.59 | 0.69/0.78 | 0.60/0.63 |
| Mix-up | 0.72/0.99 | 0.65/0.90 | 0.66/1.01 | 0.70/1.04 | 0.67/1.05 | 0.68/1.00 |

*Table 13.* Comparison of vision backbones for semantic region unitization. Results on Qwen3-VL across all five categories. DINOv3-L consistently outperforms supervised alternatives.

| Backbone | Math | | Science | | Document | | Code | | General | | Average | |
| | LDS | Top-5 | LDS | Top-5 | LDS | Top-5 | LDS | Top-5 | LDS | Top-5 | LDS | Top-5 |
|---|---|---|---|---|---|---|---|---|---|---|---|---|
| CLIP ViT-L/14 | 0.68 | 0.89 | 0.64 | 0.78 | 0.62 | 0.85 | 0.60 | 0.76 | 0.61 | 0.80 | 0.63 | 0.82 |
| SigLIP ViT-SO400M | 0.70 | 0.93 | 0.66 | 0.82 | 0.64 | 0.90 | 0.63 | 0.81 | 0.64 | 0.85 | 0.65 | 0.86 |
| **DINOv3-L** | **0.76** | **1.05** | **0.74** | **0.89** | **0.70** | **1.01** | **0.68** | **0.91** | **0.73** | **0.95** | **0.72** | **0.96** |

object boundaries compared to contrastive language-image pretraining. This finding suggests that attribution quality depends critically on the semantic coherence of the underlying region partition.

## F.6. Extended Top-K Drop Analysis

The main paper reports Top-5 Drop. Here we provide extended results for Top-1 and Top-3 Drop (Tables 14 and 15), which test whether our method correctly identifies the single most important region and the top few regions, respectively. vSTREAM achieves best or second-best performance across most K values and categories. The improvement is most pronounced for Top-1 Drop in Document tasks, where correctly identifying the single most relevant text block or table is crucial for understanding model behavior.

*Table 14.* Top-K Drop comparison across K values on Qwen3-VL. Higher values indicate better identification of causally important regions.

| Method | Math | | | Science | | |
| | Top-1 | Top-3 | Top-5 | Top-1 | Top-3 | Top-5 |
|---|---|---|---|---|---|---|
| Random | 0.02 | 0.05 | 0.08 | 0.03 | 0.07 | 0.11 |
| Attention | 0.08 | 0.19 | 0.31 | 0.07 | 0.17 | 0.28 |
| InputGrad | 0.22 | 0.48 | 0.72 | 0.21 | 0.45 | 0.68 |
| AttnLRP | 0.26 | 0.54 | 0.81 | 0.32 | 0.71 | 1.08 |
| TAM | 0.31 | 0.68 | 1.02 | 0.29 | 0.63 | 0.95 |
| **vStream** | **0.33** | **0.72** | **1.05** | **0.28** | **0.61** | 0.92 |

*Table 15.* Top-K Drop comparison (continued) for Document, Code, and General categories.

| Method | Document Top-1 | Top-3 | Top-5 | Code Top-1 | Top-3 | Top-5 | General Top-1 | Top-3 | Top-5 |
|--------|-------|-------|-------|-------|-------|-------|-------|-------|-------|
| Random | 0.04 | 0.09 | 0.15 | 0.02 | 0.06 | 0.09 | 0.02 | 0.05 | 0.07 |
| Attention | 0.09 | 0.21 | 0.33 | 0.08 | 0.18 | 0.29 | 0.07 | 0.16 | 0.26 |
| InputGrad | 0.21 | 0.46 | 0.71 | 0.28 | 0.62 | 0.96 | 0.23 | 0.50 | 0.75 |
| AttnLRP | 0.24 | 0.51 | 0.78 | 0.25 | 0.54 | 0.82 | 0.30 | 0.65 | 0.98 |
| TAM | 0.31 | 0.68 | 1.04 | **0.33** | **0.72** | **1.06** | 0.27 | 0.59 | 0.89 |
| **vStream** | **0.34** | **0.73** | **1.09** | 0.28 | 0.61 | 0.91 | **0.31** | **0.68** | **1.02** |

## F.7. Reasoning Trajectory Analysis: Extended Results

We provide extended statistics and visualizations for the reasoning trajectory analysis introduced in Section 4.4.

### F.7.1. QUANTITATIVE METRICS

We compute two geometric metrics for each reasoning trajectory:

- **Path Length**: Total Euclidean distance traveled in the PCA-projected attribution space across all reasoning steps.

- **Tortuosity**: Ratio of path length to net displacement (start-to-end distance). A value of 1.0 indicates a straight path; higher values indicate more wandering.

*Table 16.* Trajectory statistics across models and outcome types. Successful reasoning exhibits shorter, less tortuous paths in attribution space. Values show mean $\pm$ std across $n = 1500$ samples per group.

| Model | Path Length $\downarrow$ Success | Failure | Tortuosity $\downarrow$ Success | Failure |
|-------|---------|---------|---------|---------|
| Qwen3-VL | $0.003 \pm 0.002$ | $0.006 \pm 0.005$ | $13.7 \pm 12.0$ | $25.4 \pm 26.1$ |
| GLM-4.1V | $0.004 \pm 0.003$ | $0.007 \pm 0.006$ | $14.8 \pm 13.5$ | $28.3 \pm 29.7$ |
| MiMo-VL | $0.003 \pm 0.002$ | $0.005 \pm 0.004$ | $11.9 \pm 10.8$ | $22.7 \pm 24.2$ |
| Cosmos-R1 | $0.004 \pm 0.003$ | $0.006 \pm 0.005$ | $15.2 \pm 14.1$ | $27.9 \pm 28.5$ |

Across all four models, successful reasoning chains exhibit approximately 50% shorter path lengths and 40–50% lower tortuosity compared to unsuccessful chains. This suggests that valid reasoning corresponds to a more stable, directed traversal of the visual attribution manifold.

### F.7.2. INTERPRETATION

We interpret these geometric differences through the lens of "hypothesis switching." Unsuccessful chains frequently reassign visual attention to different regions mid-reasoning, manifesting as erratic, high-tortuosity trajectories in attribution space. In contrast, successful chains quickly commit to a consistent set of visual evidence and maintain stable attention throughout the reasoning process.

This observation has practical implications: trajectory metrics could potentially serve as early warning signals for hallucination detection, flagging reasoning chains that exhibit unusually high path length or tortuosity before the final answer is generated.

### F.7.3. 3-WAY ERROR ANALYSIS: WANDERING VS. FIXATION

We decompose unsuccessful POPE examples ($n$=3,000) into *Reasoning Failures* (RF; incorrect answer, plausible object) and *Hallucinations* (H; object not in image). Table 17 reports the full metric breakdown.

To test whether the Fixation pattern extends beyond binary POPE judgments to open-ended captioning, we compute per-caption concentration on 2,000 COCO captions and regress against CHAIR$_i$ (Rohrbach et al., 2018), a standard object-hallucination score for captions.

*Table 17.* 3-way trajectory metrics on POPE ($n$=3,000). S = Success, RF = Reasoning Failure, H = Hallucination. Path geometry separates errors from success; concentration separates Fixation (H) from Wandering (RF).

| Metric | S | RF | H | KW $p$ | $d$ (S–H) |
|---|---|---|---|---|---|
| Path Length | $0.004 \pm 0.002$ | $0.006 \pm 0.005$ | $0.005 \pm 0.003$ | $< .001$ | $-0.73$ |
| Tortuosity | $5.6 \pm 3.9$ | $9.4 \pm 10.7$ | $8.0 \pm 5.0$ | $< .001$ | $-0.60$ |
| Concentration | $0.209 \pm 0.054$ | $0.196 \pm 0.058$ | $0.228 \pm 0.059$ | $< .001$ | $-0.34$ |

## F.8. Per-Step Attribution Fidelity

Per-step attribution fidelity results are reported in the main text (Figure 12). Correct reasoning chains maintain stable $R^2$ throughout generation, while incorrect chains degrade at roughly 20% of reasoning elapsed, corroborating the early tortuosity signal in Figure 10.

## F.9. Estimator Architecture: Linear vs. MLP

We compare the linear estimator used by VSTREAM against MLP variants of increasing width on Qwen3-VL (3 seeds each).

*Table 18.* Linear vs. MLP estimator (Qwen3-VL, 3 seeds). MLP variants yield diminishing returns despite $> 100\times$ more parameters.

| Estimator | Params | $R^2$ | Pearson $\rho$ | Latency (ms) |
|---|---|---|---|---|
| Linear | 1,152 | $0.65 \pm .01$ | $0.81 \pm .01$ | 0.03 |
| MLP-64 | 73,856 | $0.66 \pm .01$ | $0.82 \pm .01$ | 0.12 |
| MLP-128 | 147,712 | $0.67 \pm .01$ | $0.82 \pm .00$ | 0.15 |
| MLP-256 | 295,424 | $0.67 \pm .01$ | $0.82 \pm .01$ | 0.18 |

The linear estimator achieves $R^2$=0.65 with 1,152 parameters and 0.03 ms inference latency. MLP-256 gains $+0.02$ in $R^2$ (reaching 0.67) but requires $256\times$ more parameters and $6\times$ higher latency. The linear estimator is therefore the preferred choice for streaming deployment.

## F.10. Cross-Architecture Estimator Weight Patterns

## F.11. Context-Length Robustness

## F.12. Out-of-Distribution Generalization: VQA-RAD

We test whether the estimator trained exclusively on natural images generalizes to medical radiology images (VQA-RAD (Lau et al., 2018)) without retraining.

*Table 19.* Attribution fidelity on VQA-RAD (OOD). VSTREAM estimator trained on natural images only, applied without retraining. Format: LDS / Top-5 Drop; higher is better.

| Method | LDS | Top-5 Drop |
|---|---|---|
| Random | 0.29 | 0.14 |
| Attention | 0.41 | 0.32 |
| InputGrad | 0.58 | 0.71 |
| AttnLRP | 0.66 | 0.79 |
| TAM | 0.63 | 0.88 |
| vStream (Ours) | 0.67 | 0.84 |

Despite the large domain shift, VSTREAM achieves competitive fidelity with no domain adaptation, demonstrating that the learned attention-to-ablation mapping transfers across image modalities.

## F.13. Segmentation Evaluation

While our primary goal is visual attribution rather than semantic segmentation, we evaluate how well our attribution maps align with ground-truth object masks. We use three benchmarks: ImageNet-Segmentation (Guillaumin et al., 2014), which provides pixel-level annotations for a subset of ImageNet validation images; COCO (Lin et al., 2014), with instance

segmentation masks; and the RefCOCO family of datasets (Kazemzadeh et al., 2014) for referring expression grounding. For RefCOCO evaluation, we report results on the *val* split, which is the standard evaluation protocol. The RefCOCO family includes three variants: RefCOCO (general expressions), RefCOCO+ (excludes absolute location terms like "left" or "right"), and RefCOCOg (longer, more complex expressions). We use RefCOCO-val as it provides a balanced evaluation across diverse object categories. Note that attribution methods naturally highlight causally relevant regions rather than full object extents, making segmentation a challenging proxy task.

*Table 20.* Segmentation performance (mIoU) on Qwen3-VL across three benchmarks. For RefCOCO, we report results on the val split. All methods achieve comparable scores, confirming that attribution maps do not directly correspond to object boundaries.

| Method | ImageNet-Seg | COCO | RefCOCO-val |
|---|---|---|---|
| Attention | 0.21 | 0.19 | 0.23 |
| InputGrad | 0.23 | 0.21 | 0.25 |
| AttnLRP | 0.27 | 0.24 | 0.27 |
| TAM | 0.27 | 0.25 | 0.27 |
| vStream (Ours) | 0.27 | 0.24 | 0.27 |

The results indicate that all methods achieve comparable mIoU scores in the 0.20–0.30 range, which is typical for attribution-to-segmentation evaluation. This confirms that attribution maps, designed to highlight causally relevant evidence, do not necessarily align with full object boundaries.

## G. Qualitative Analysis

We provide additional qualitative results in Figure 23–Figure 38. These figures show step-by-step visual attribution results from our main model (Qwen3-VL) and all baseline methods across all five task categories: Math, Science, Document, Code, and General.

Each figure displays the attribution heatmap for individual reasoning steps, enabling direct comparison between VSTREAM and baseline attribution methods. The visualizations demonstrate that VSTREAM produces focused, semantically coherent attribution maps that track the model's reasoning process in real-time.

### G.1. Failure Analysis

Despite strong performance, we identify distinct failure modes where attribution remains challenging. These failure cases are not directly visualized in the figures but are described below based on our analysis.

**Ambiguity in Visual References.** When the visual input contains multiple identical objects (e.g., "count the red apples"), the attribution map often splits intensity across all candidates simultaneously rather than sequentially focusing on individual instances. While strictly "correct" in terms of feature matching, this does not reflect the sequential nature of human counting and can make traces harder to interpret.

**Hallucination and Disconnected Reasoning.** When the model generates plausible text that is not grounded in the image (hallucination), our attribution maps often become diffuse or uniform, lacking a clear focal point. This suggests that diffuse attribution could serve as a detector for hallucinated content, an avenue for future research.

**Dense Text and Small Objects.** For images containing dense text (documents, code screenshots) or small objects, the semantic regions from DINO clustering may not perfectly isolate individual characters or tiny elements. In these cases, attribution may highlight a region containing the relevant element but lack sub-region precision.

**Abstract Reasoning Steps.** During purely symbolic computation steps (e.g., "Therefore, 2x + 3 = 7 implies x = 2"), visual attribution naturally becomes weak because the reasoning genuinely does not depend on visual input. Our method correctly produces low attribution scores in these cases. Users should interpret weak attribution during computational steps as expected behavior rather than a failure mode.

# H. Broader Impact and Limitations

## H.1. Broader Impact

**Advancing Trustworthy AI.** The primary contribution of this work is to enhance the transparency and interpretability of Vision-Language Models (VLMs). As these models are increasingly deployed in high-stakes domains—such as medical diagnosis, autonomous navigation, and legal analysis—the ability to verify *why* a model made a decision is paramount. By providing fast and accurate attribution maps, our method allows human operators to verify that model reasoning relies on relevant visual evidence rather than spurious correlations or hallucinations. This is a critical step towards safe deployment of reasoning models.

**Potential Risks.** While intended to reveal model reasoning, interpretability tools can be double-edged. There is a risk that accurate-looking attribution maps could be used to generate convincing justifications for incorrect or biased model decisions, potentially leading users to over-trust a flawed system. Furthermore, if the estimator itself is adversarially manipulated, it could hide the model's reliance on sensitive or protected attributes (e.g., race or gender) in decision-making processes. It is crucial that these tools are used as part of a holistic auditing framework, not as a standalone guarantee of safety.

## H.2. Limitations

We identify several limitations of our approach that should be considered when applying vStream in practice.

**Linearity Assumption.** Our method employs a linear estimator to predict the causal effect of feature ablation. While our empirical results suggest that linear directions in the activation space of modern transformers capture significant causal information, this is a simplifying assumption. Complex, non-linear interactions between visual features (where the suppression of one feature only matters if another is also present) may not be fully captured by our current formulation. Future work could explore non-linear estimators (e.g., MLPs or attention-based predictors) for the attribution head, though this may trade off interpretability and training stability.

**Dependency on Visual Backbone (DINO).** Our approach relies on the quality of the underlying visual representations (specifically DINOv3) to define semantic regions for attribution. If the visual encoder fails to semantically separate relevant objects or concepts, or if the feature resolution is too coarse, our estimator cannot recover precise attributions. This dependency means:

- For images with unusual visual content not well-represented in DINO's training data, region quality may degrade.

- Very small objects or fine-grained text may not be isolated into separate regions.

- The computational cost of running DINOv3 adds approximately 45ms per image.

**Training Data Quality.** The estimator is trained only on examples where the VLM produces correct final answers. This design choice ensures we learn attention patterns associated with successful reasoning. However, it also means:

- The estimator may be less calibrated for incorrect reasoning traces.

- Attribution quality depends on having sufficient correctly-answered examples in each domain.

- For very challenging tasks where models rarely succeed, collecting training data becomes difficult.

**Generalization Across Domains.** While our cross-task experiments show reasonable transfer between related domains (e.g., Math $\leftrightarrow$ Science), transfer to visually distinct domains (e.g., Math $\rightarrow$ Document) is weaker. This suggests that practitioners may need to collect domain-specific training data for optimal performance in specialized applications.

**Attention as a Proxy for Information Flow.** Our method assumes that attention weights are informative proxies for information flow in transformers. However, recent work has shown that attention can be manipulated without affecting outputs (Jain & Wallace, 2019), and that alternative pathways (e.g., residual connections, MLP layers) can route information independently of attention. Our empirical success suggests attention captures sufficient signal for attribution, but it may not be a complete picture of the model's information processing.

**Computational Requirements for Ground-Truth Collection.** While inference is fast, collecting ground-truth ablation effects for training requires multiple forward passes per example. For a training set of 500 examples with 32 masks each, this amounts to 16,000 forward passes. On an H100 GPU, this takes approximately 2–4 hours per model-task pair.

**Model-Specific Training Requirement.** Our estimator must be trained separately for each VLM, as the learned linear projection is inherently tied to the specific model's internal representation space. This limitation is fundamental rather than incidental: different VLMs employ distinct architectures, attention mechanisms, and learned feature spaces, making cross-model transfer of the estimator impractical without retraining. While this per-model training requirement adds deployment overhead, we note that (1) the training cost is modest (2–4 hours on a single GPU per model-task pair), and (2) this constraint is shared by virtually all model-specific interpretability methods that operate on internal representations. We view the development of architecture-agnostic attribution features as an important direction for future work, as discussed in Section H.3.

## H.3. Future Directions

Several promising directions could extend and improve upon our work. On the methodological side, exploring non-linear estimators such as attention-based or MLP-based architectures could capture complex feature interactions that our linear model may miss, potentially improving attribution quality on images with intricate visual relationships. Developing architecture-agnostic features would enable zero-shot cross-model transfer, allowing practitioners to train an estimator on one VLM and apply it to another without retraining. This would dramatically reduce the barrier to deploying attribution tools across the rapidly evolving landscape of vision-language models.

On the application side, our observation that hallucinated content produces diffuse attribution maps suggests a natural extension toward automatic hallucination detection, where attribution entropy or concentration metrics could serve as early warning signals for unreliable model outputs. Attribution maps could also guide targeted interventions for model editing, enabling surgical corrections to model behavior without full retraining. Finally, extending our framework to video understanding and multi-image reasoning would address the growing importance of temporal and multi-view visual reasoning, where tracking visual attribution across frames could reveal how models integrate information over time.

```python
def train_step(model, estimator, batch, optimizer):
    """
    Performs a single training step for the Amortized Estimator.

    Args:
        model: The frozen VLM backbone.
        estimator: The AmortizedAttributor being trained.
        batch: Dictionary containing 'img', 'text', 'regions', 'target'.
    """
    # 1. Clean pass to get baselines and attention features
    with torch.no_grad():
        # model returns logits and attentions [B, L, H, T, M]
        # L=Layers, H=Heads, T=Time steps, M=Visual tokens
        out = model(batch['img'], batch['text'], output_attentions=True)
        base_logp = out.logits.log_softmax(-1).gather(-1, batch['target'])
        attn = out.attentions

    # 2. Sample random binary masks for K regions (N=32 samples)
    # masks: [B, N, K] where N is sample size, K is num regions
    B = batch['img'].shape[0]
    K = batch['regions'].shape[0]  # regions: [K, M] (shared across batch)
    N = 32
    masks = torch.randint(0, 2, (B, N, K)).to(model.device).float()

    # 3. Compute Ground Truth Effect (delta_true) via VLM ablation
    delta_true_list = []
    with torch.no_grad():
        for i in range(N):
            # Forward pass with attention masking applied to regions
            # mask=masks[:, i] prevents VLM from attending to masked regions
            out_abl = model(batch['img'], batch['text'], mask=masks[:,i])
            abl_logp = out_abl.logits.log_softmax(-1).gather(-1, batch['target'])

            # Delta: Positive value means ablation hurt performance (drop in log-prob)
            delta_true_list.append(base_logp - abl_logp)

    delta_true = torch.cat(delta_true_list, dim=1) # [B, N]

    # 4. Predict Effect (delta_pred) using Estimator
    # scores: [B, K] (Predicted importance per region)
    scores = estimator(attn, batch['regions'], batch['span_mask'])

    # Linear assumption: Effect of subset = sum of scores (Eq. 9)
    # delta_pred: [B, N]
    delta_pred = torch.einsum('bk,bnk->bn', scores, masks)

    # 5. Loss: Maximize Pearson Correlation
    # We focus on ranking subsets correctly rather than absolute magnitude regression
    loss = -pearson_correlation(delta_pred, delta_true)

    optimizer.zero_grad()
    loss.backward()
    optimizer.step()

    return loss.item()
```

*Implementation 1.* PyTorch implementation of the Training Step. The estimator is trained to rank random subsets of regions by maximizing the Pearson correlation between predicted scores and actual ablation drops.

```
1  import torch
2  import torch.nn as nn
3
4  class AmortizedAttributor(nn.Module):
5      def __init__(self, n_layers: int, n_heads: int):
6          super().__init__()
7          # Learnable linear projection W (Eq. 9)
8          # Maps flattened attention features (L*H) to a scalar score
9          self.estimator = nn.Linear(n_layers * n_heads, 1, bias=False)
10
11     def forward(self, attn_weights, regions, span_mask):
12         """
13         Compute attribution scores for visual regions.
14
15         Args:
16             attn_weights: [B, L, H, T, M] Attention maps from VLM
17             regions: [K, M] Binary membership masks for K regions
18             span_mask: [T] Binary mask for target time span S
19
20         Returns:
21             scores: [B, K] Attribution score for each region
22         """
23         # Eq. 4: Temporal Pooling over target span S
24         # Select time steps where span_mask is 1 and average
25         # attn: [B, L, H, M]
26         masked_attn = attn_weights * span_mask[None, None, None, :, None]
27         attn = masked_attn.sum(dim=3) / (span_mask.sum() + 1e-6)
28
29         # Eq. 4: Regional Pooling over visual regions R_k
30         # Aggregate attention for tokens belonging to each region k
31         # features: [B, K, L, H]
32         features = torch.einsum('blhm,km->bklh', attn, regions)
33
34         # Normalize by region size
35         region_sizes = regions.sum(dim=1).view(1, -1, 1, 1)
36         features = features / (region_sizes + 1e-6)
37
38         # Flatten layer/head dimensions to create feature vector
39         # features: [B, K, L*H]
40         features = features.flatten(start_dim=2)
41
42         # Eq. 9: Predict attribution scores via linear projection
43         # scores: [B, K, 1] -> [B, K]
44         scores = self.estimator(features).squeeze(-1)
45
46         return scores
```

*Implementation 2.* PyTorch implementation of the Amortized Attributor. The module efficiently pools attention features and maps them to attribution scores in a single forward pass.

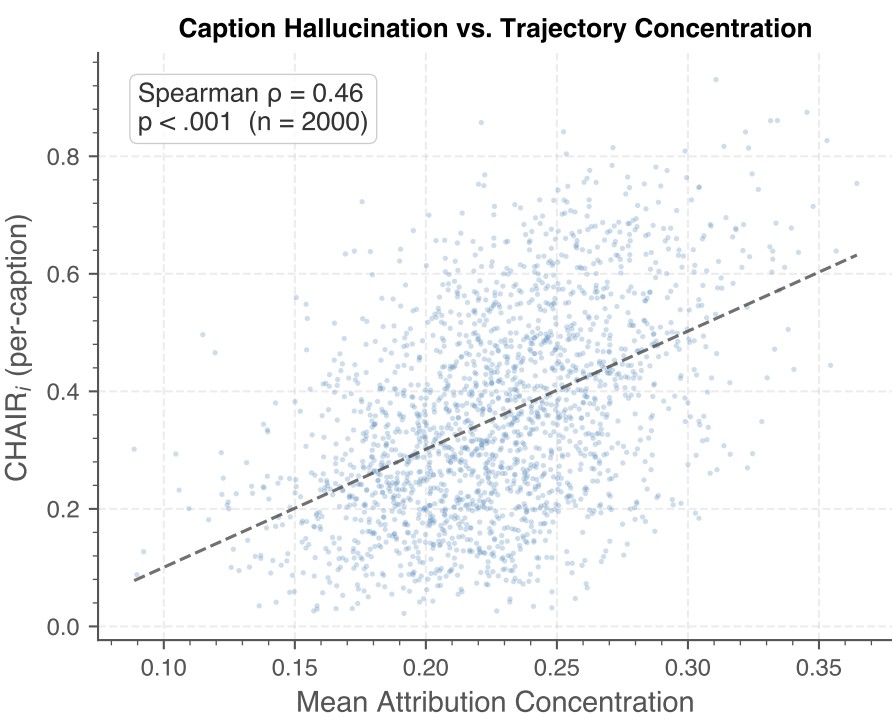

*Figure 13.* **Per-caption attribution concentration vs. CHAIR$_i$ ($n$=2,000 COCO captions, Spearman $\rho$=0.46, $p$<.001).** Higher concentration during caption generation predicts more severe object hallucination, extending the Fixation finding to open-ended generation.

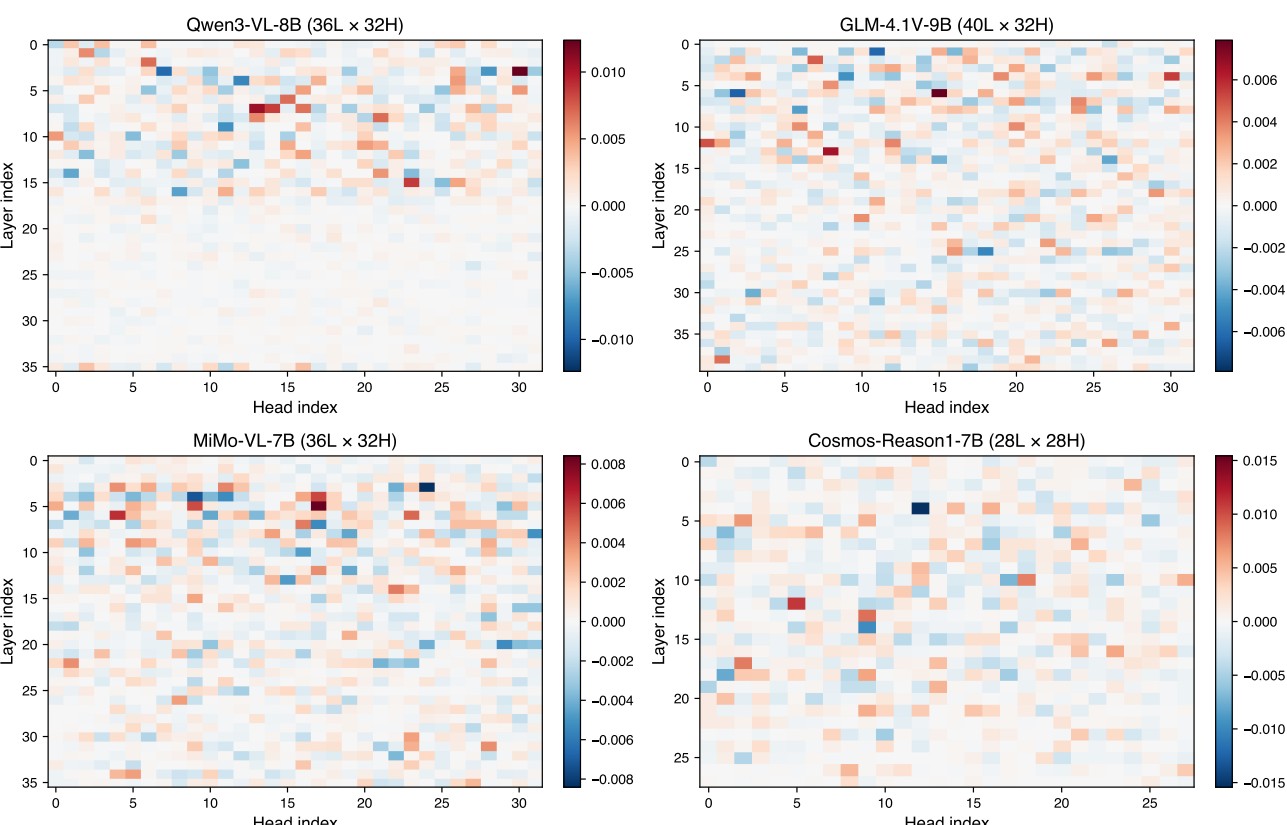

*Figure 14.* **Estimator weight heatmaps** $\mathbf{w} \in \mathbb{R}^{L \times H}$ **across four architectures.** All models concentrate weight in early-to-mid layers, suggesting a consistent architectural prior: early layers encode coarse visual-semantic alignment that is most predictive of ablation effects.

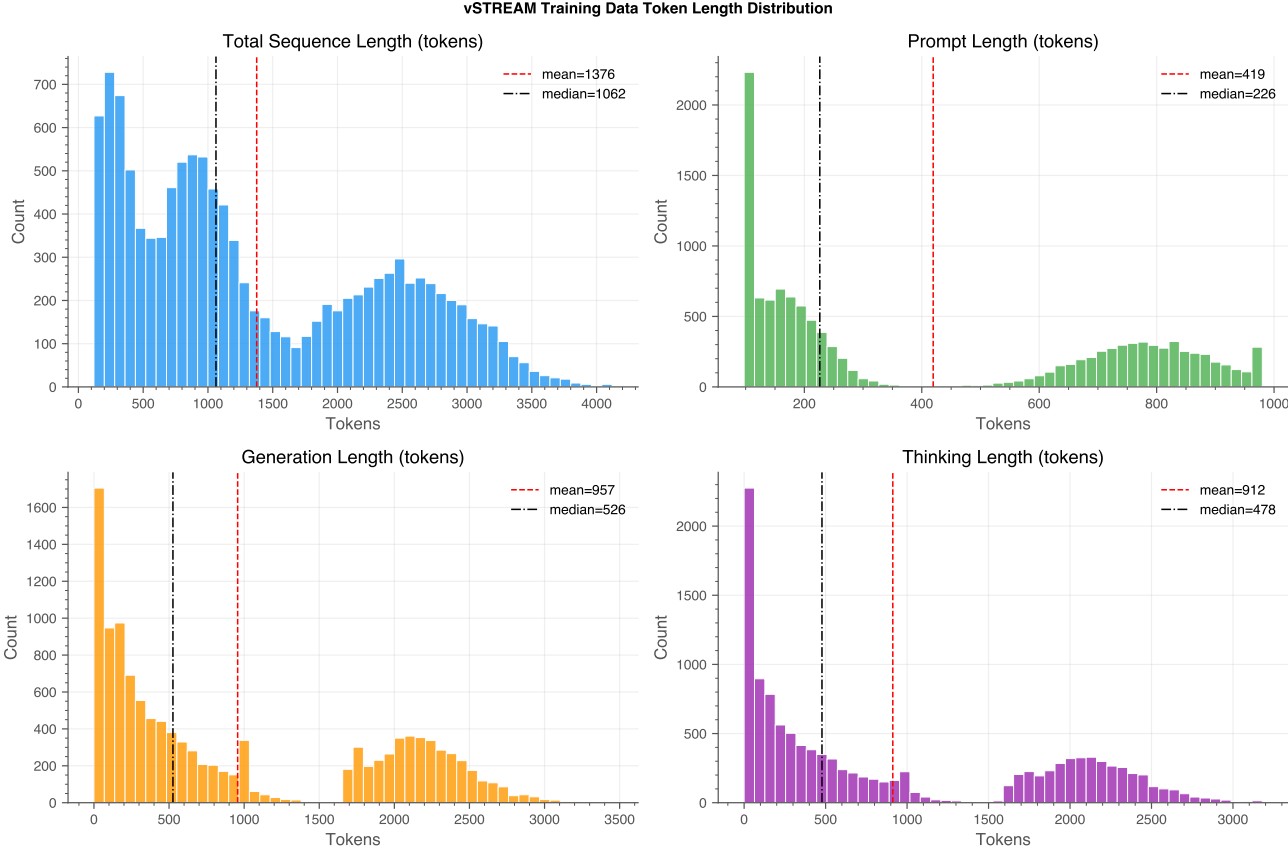

*Figure 15.* **Generation length distribution across training examples.** The distribution is bimodal: short VQA responses (∼400 tokens) and long reasoning traces (∼3,200 tokens).

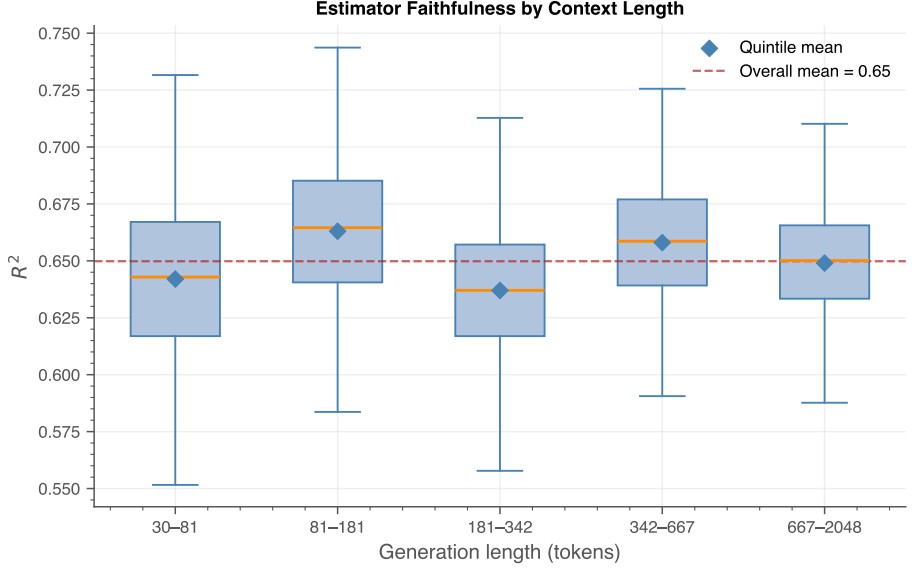

*Figure 16.* **Faithfulness ($R^2$) by generation-length quintile.** No systematic degradation as context length increases, indicating that the estimator generalizes across short VQA and long reasoning traces.

What are these opacities anterior to the right kidney?

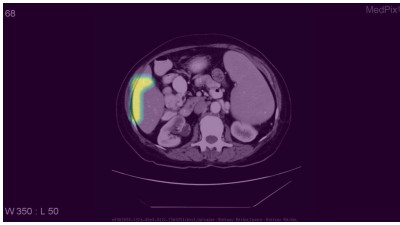

...
[Thinking Step 10] The right kidney is on the right side of the patient, so in the CT slice, the right kidney is on the left side of the image.
...

Is there evidence of an aortic aneurysm?

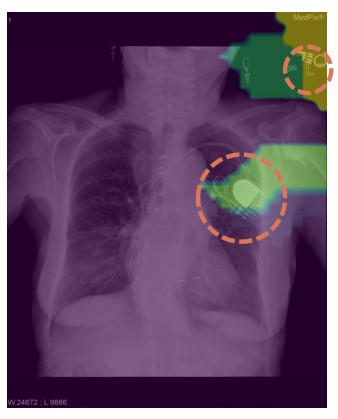

...
[Thinking Step 8] Wait, the image shows a pacemaker on the right side (since it's labeled L for left, and the device is on the right chest).
...

In this CT scan image, where are the kidney?

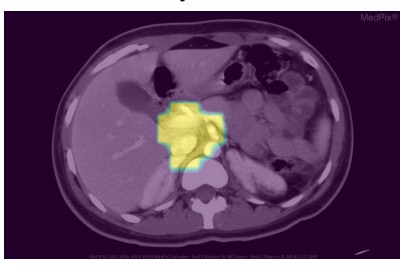

...
[Thinking Step 7] Wait, This is the retroperitoneal space, containing the abdominal aorta, inferior vena cava, and the body of the pancreas.
...

*Figure 17.* **OOD attribution on VQA-RAD (medical radiology).** The estimator, trained on natural images only and applied without retraining, correctly localizes kidneys in abdominal CT (top, bottom) and a pacemaker in chest X-ray (middle).

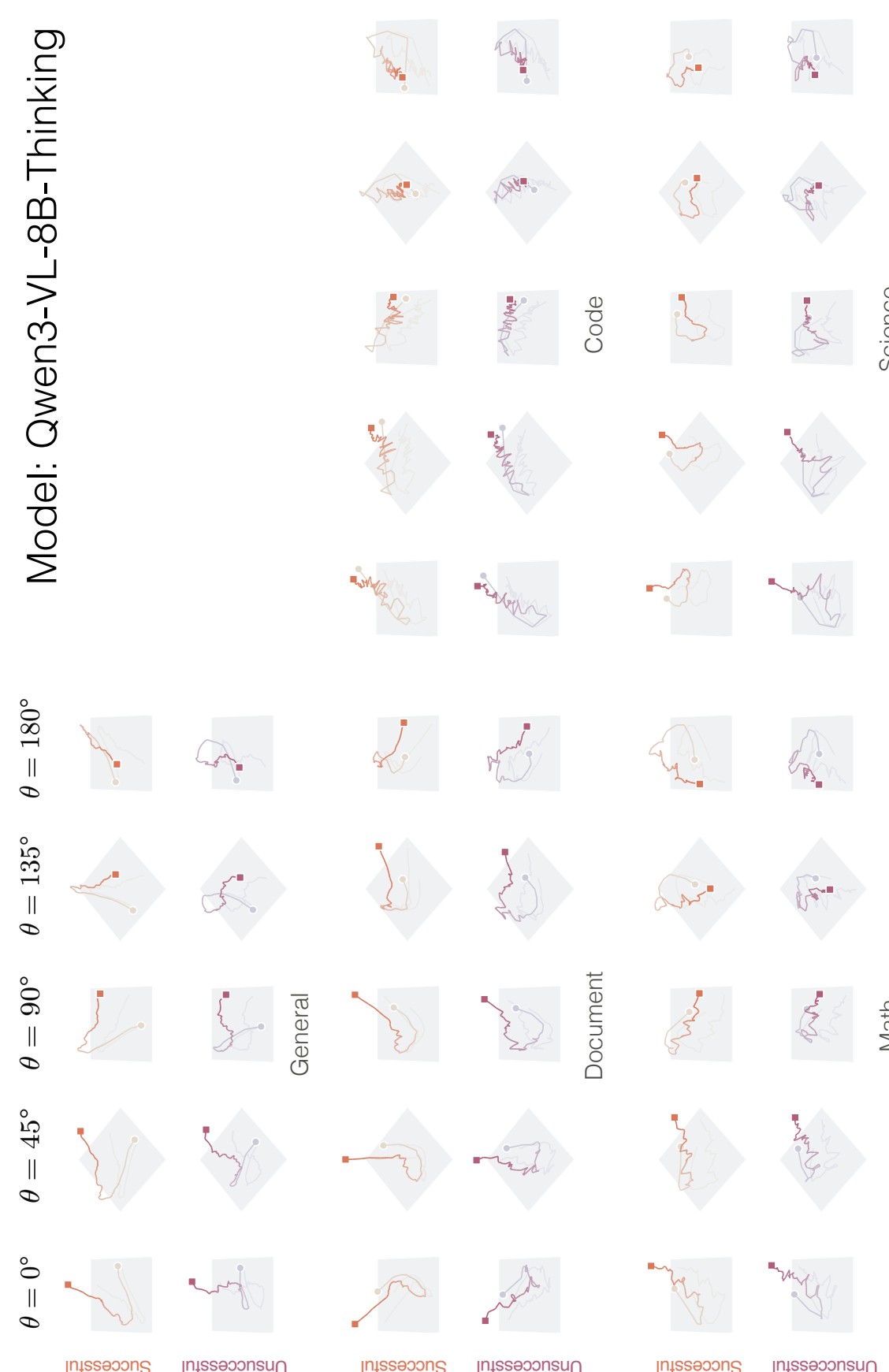

*Figure 18.* **Reasoning trajectory dynamics for Qwen3-VL-8B-Thinking.** Visual attribution trajectories projected into PCA space for successful (*left*) and unsuccessful (*right*) reasoning chains.

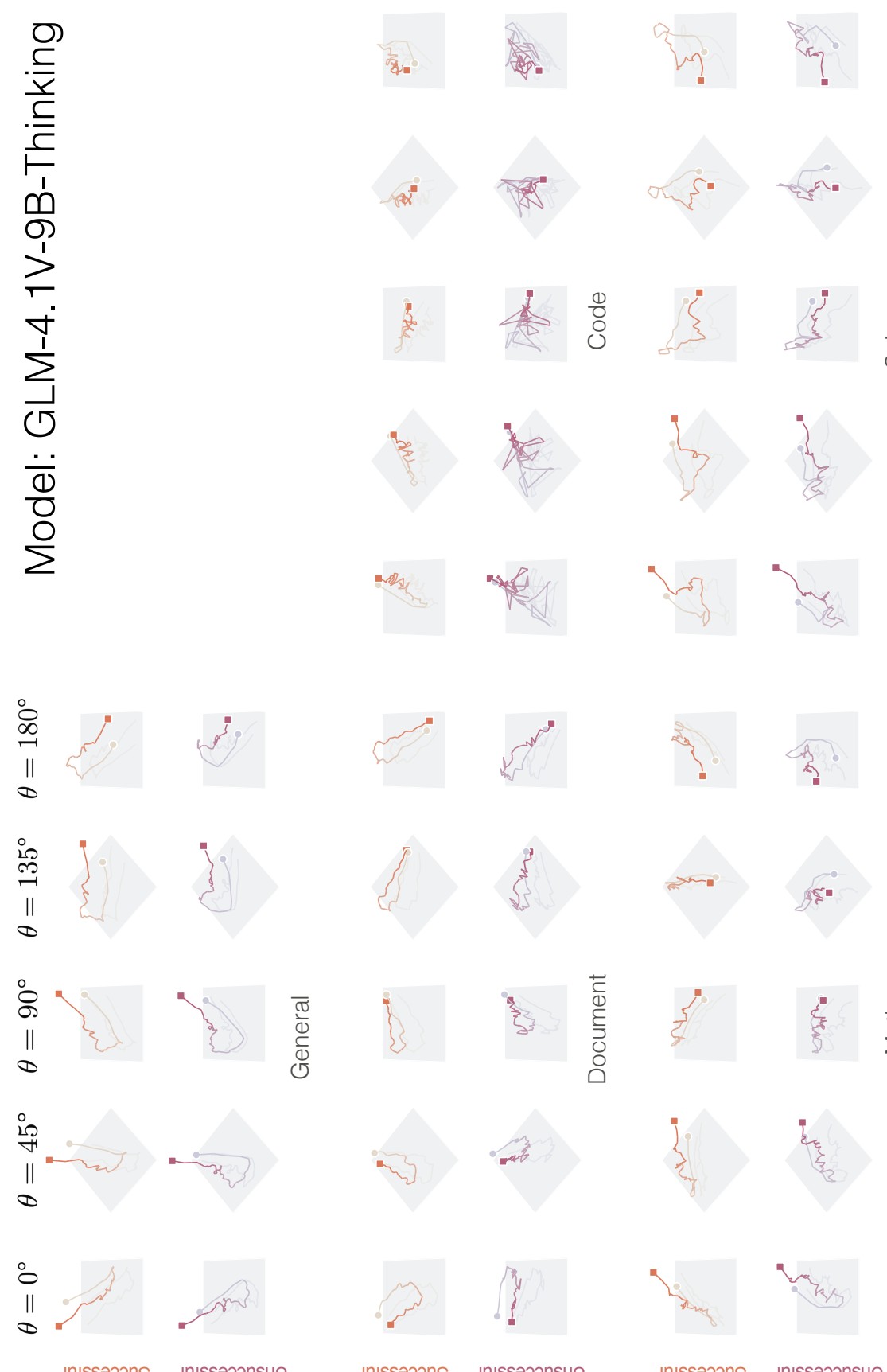

*Figure 19.* **Reasoning trajectory dynamics for GLM-4.1V-9B-Thinking.** Visual attribution trajectories projected into PCA space for successful (*left*) and unsuccessful (*right*) reasoning chains.

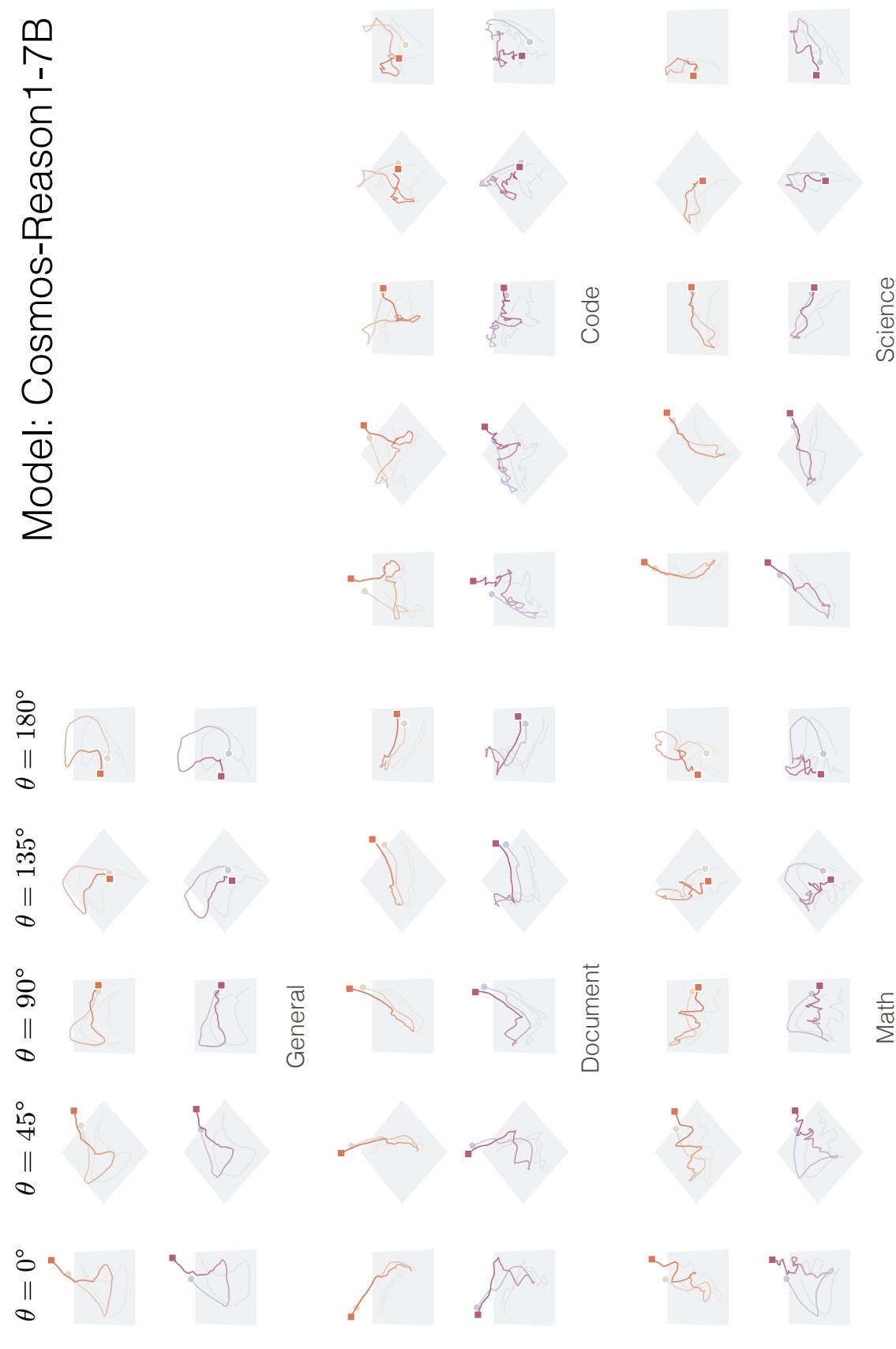

*Figure 20.* **Reasoning trajectory dynamics for Cosmos-R1.** Visual attribution trajectories projected into PCA space for successful (*left*) and unsuccessful (*right*) reasoning chains.

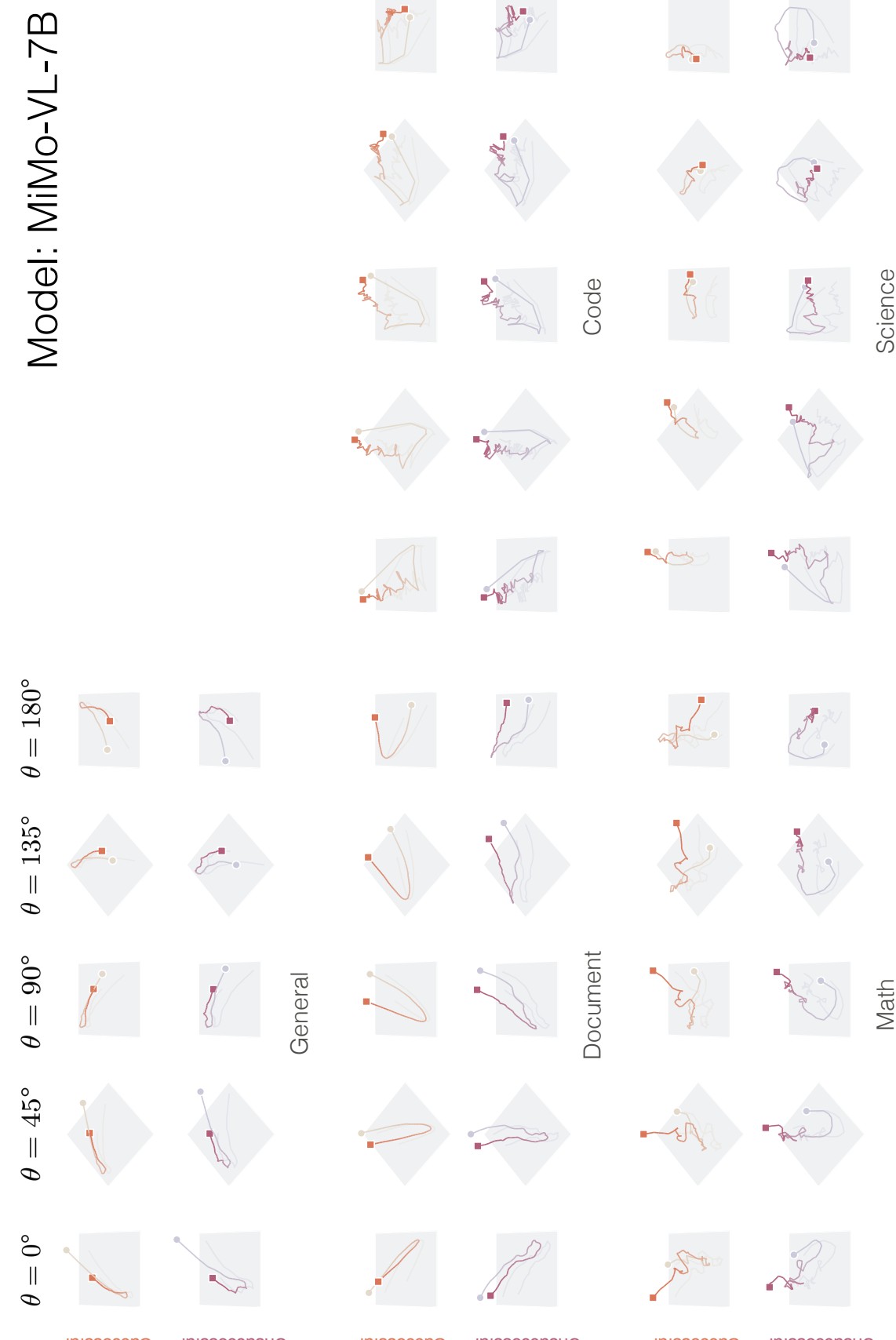

*Figure 21.* **Reasoning trajectory dynamics for MiMo-VL-7B.** Visual attribution trajectories projected into PCA space for successful (*left*) and unsuccessful (*right*) reasoning chains.

# Model: Qwen3VL-8B-Thinking

## General

| Visual Context | **Ours** | Attention | AttnLRP | Input Gradient | TAM |

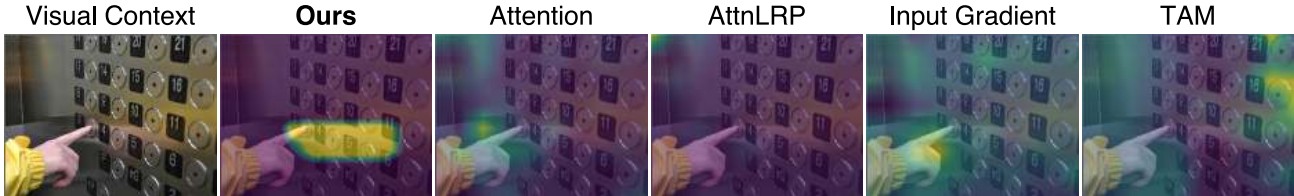

**Query:** Based on the given image, explain which floor the person in the photo is trying to go to.
**Response:** Got it, let's try to figure out this problem. First, let's look at the image. The hand position suggests they are pressing or about to press that floor button. [...] Wait, The person's hand is pointing at the 3rd-floor button, positioned beside the 4th and 5th floor buttons. [...]

## Document

| Visual Context | **Ours** | Attention | AttnLRP | Input Gradient | TAM |

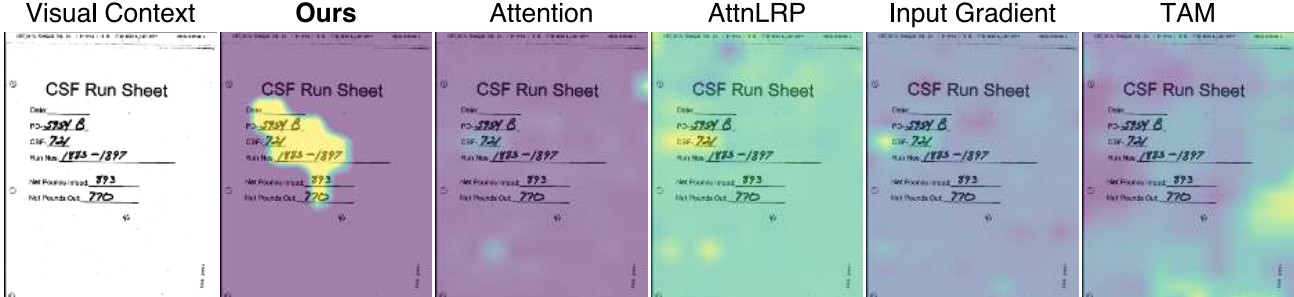

**Query:** What is the CSF? Find the answer in the document image.
**Response:** Okay, let's take a look at the document. First, I noticed some handwritten numbers. [...]

## Code

| Visual Context | **Ours** | Attention | AttnLRP | Input Gradient | TAM |

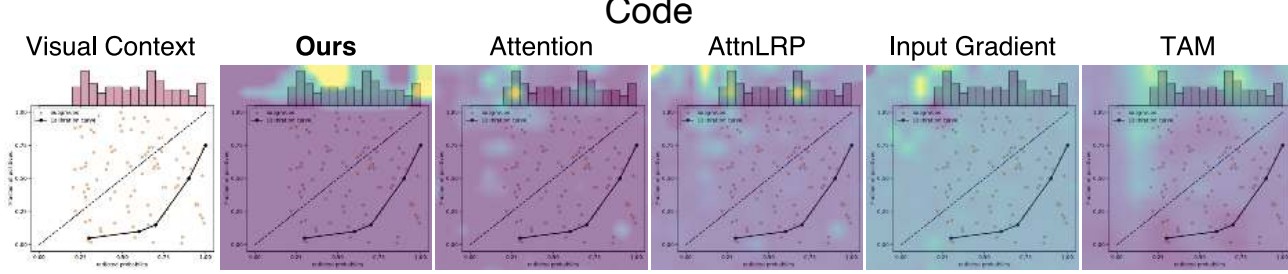

**Query:** You are an expert Python developer who specializes in writing matplotlib code based on a given picture. Now, please give me the matplotlib code that reproduces the picture below.
**Response:** Got it, let's try to figure out how to recreate this plot. First, let's analyze the components. Let's break down the elements: [...] **Top bar chart (subgroups)**: It's a bar plot with pink bars. [...]

*Figure 22.* **Additional qualitative results: Qwen3-VL on general/document/code reasoning.**

# Model: Qwen3VL-8B-Thinking

## Math

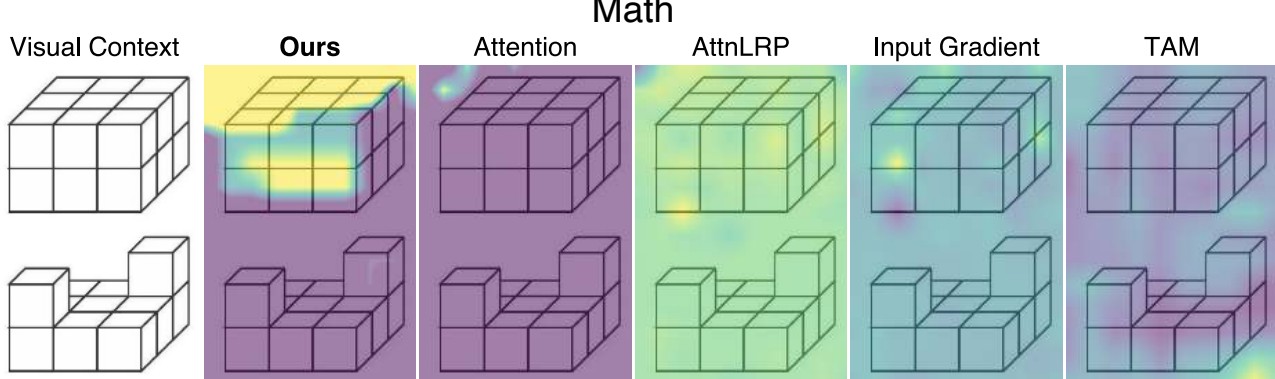

**Query:** Mike has built a construction, shown in the upper picture, from equal cubes. Lily has taken several cubes out of it, thus Mike's construction became such as we see in the lower picture. How many cubes has Lily taken? Provide your answer.

**Response:** Okay, let's try to figure out how many cubes Lily took. First, I need to understand the original construction and the modified one. The upper picture is the original structure. The top layer seems to have 3 columns and 3 rows? Wait, the upper picture is a cube structure. Let me visualize it. If it's a 3x3x2 structure? [...]

## Science

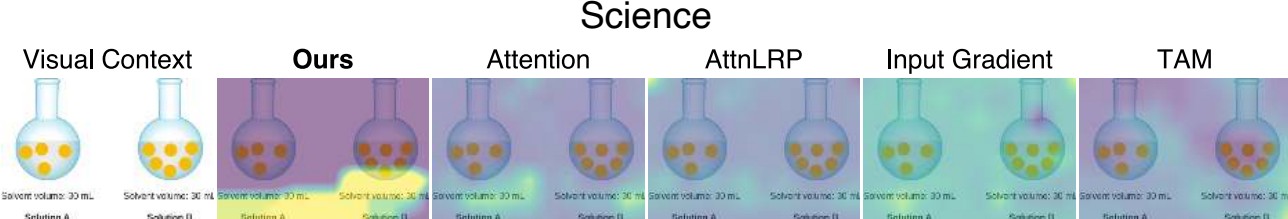

**Query:** Which solution has a higher concentration of yellow particles? Options: (A) Solution A (B) Solution B (C) neither; their concentrations are the same. Select the best answer choice.

**Response:** Got it, let's see. The question is about concentration of yellow particles in two solutions. Concentration is usually amount of solute per volume of solution. Here, the solvent volume is the same for both, 30 mL. [...]

*Figure 23.* **Additional qualitative results: Qwen3-VL on math/science reasoning.**

## Model: Qwen3VL-8B-Thinking

### General

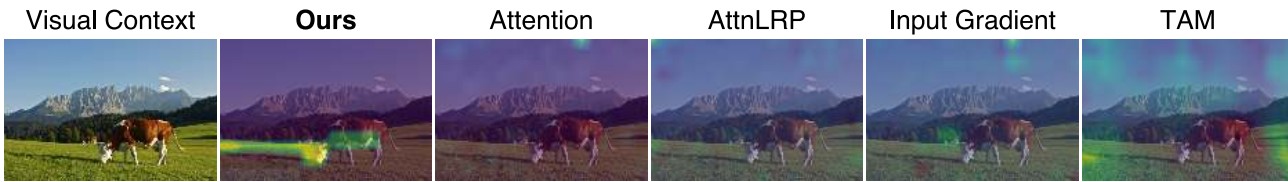

| Visual Context | **Ours** | Attention | AttnLRP | Input Gradient | TAM |

**Query:** What animal is standing against the grass? Answer the question based on what you see in the image.
**Response:** So, let's look at the image. [...] There are animals in the grassy field. The main one in the foreground is a cow, brown and white. There's another one in the background too.

### Document

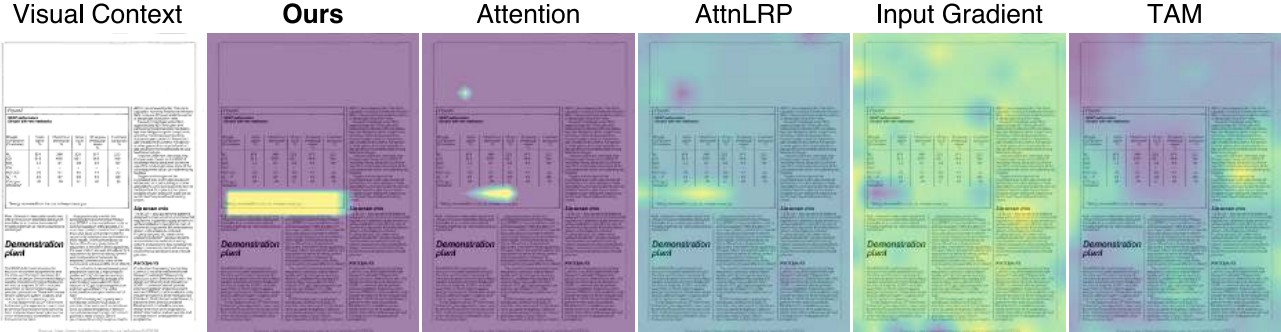

| Visual Context | **Ours** | Attention | AttnLRP | Input Gradient | TAM |

**Query:** What is the footnote of the table in fig. 2? Find the answer in the document image.
**Response:** So, let's look at the image. The table is Figure 2, which is about SCOP performance. The footnote for the table is at the bottom of the table. Let me check. The table has a note: "Energy recovered from the coal in the synthesis gas". [...]

### Code

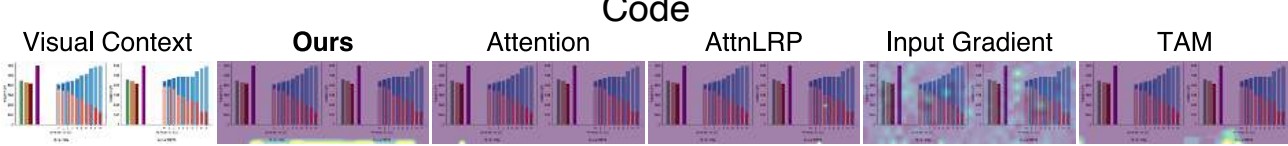

| Visual Context | **Ours** | Attention | AttnLRP | Input Gradient | TAM |

**Query:** You are an expert Python developer who specializes in writing matplotlib code based on a given picture. Now, please give me the matplotlib code that reproduces the picture below.
**Response:** Got it, let's tackle this problem. The user wants Python code using matplotlib to reproduce the given bar chart. First, I need to analyze the image. Looking at the image, there are two subplots: Flickr30k on the left and MS-COCO on the right. [...]

*Figure 24.* **Additional qualitative results: Qwen3-VL on general/document/code reasoning.**

# Model: Qwen3VL-8B-Thinking

## Math

| Visual Context | **Ours** | Attention | AttnLRP | Input Gradient | TAM |

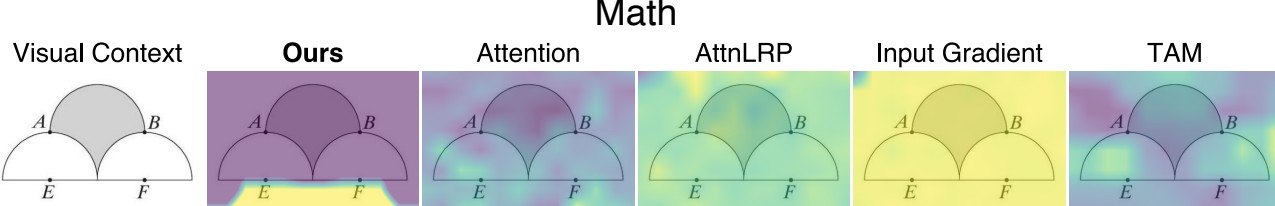

**Query:** We are given three semi-circles as shown. $A B E F$ is a rectangle and the radius of each of the semi-circles is $2 \mathrm{~cm}$. $E$ and $F$ are the centers of the bottom semi-circles. The area of the shaded region (in $\mathrm{cm}^{2}$) is <image> Provide your answer.

**Response:** Okay, let's try to figure out the area of the shaded region. First, let's make sure I understand the problem correctly. There are three semicircles: the bottom two are the ones with centers E and F, and the top one is the shaded region. The problem says ABFE is a rectangle, and each semicircle has a radius of 2 cm. E and F are the centers of the bottom semicircles. [...]

## Science

| Visual Context | **Ours** | Attention | AttnLRP | Input Gradient | TAM |

**Query:** Which property do these three objects have in common? Options: (A) stretchy (B) transparent (C) rough. Select the best answer choice.

**Response:** Got it, let's see. The three objects are labeled by bubble gum, trampoline, and rubber band. I need to figure out what property they have in common. Let's check each option. [...]

*Figure 25.* **Additional qualitative results: Qwen3-VL on math/science reasoning.**

## Model: Qwen3VL-8B-Thinking

### General

| Visual Context | **Ours** | Attention | AttnLRP | Input Gradient | TAM |
|---|---|---|---|---|---|

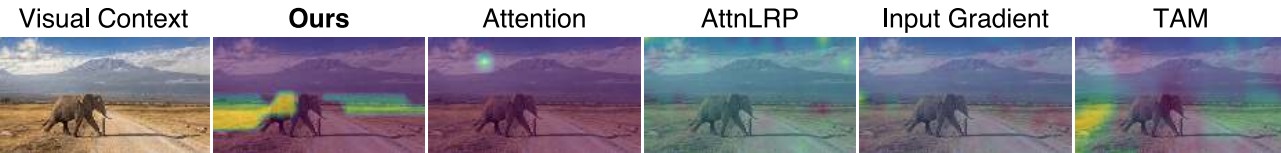

**Query:** Are the trees on the field bare or lush? Answer the question based on what you see in the image.
**Response:** [...] Now, the question is about trees in the field—bare or lush. Let's go through each step logically. Wait, do we see trees in the field? [...]

### Document

| Visual Context | **Ours** | Attention | AttnLRP | Input Gradient | TAM |
|---|---|---|---|---|---|

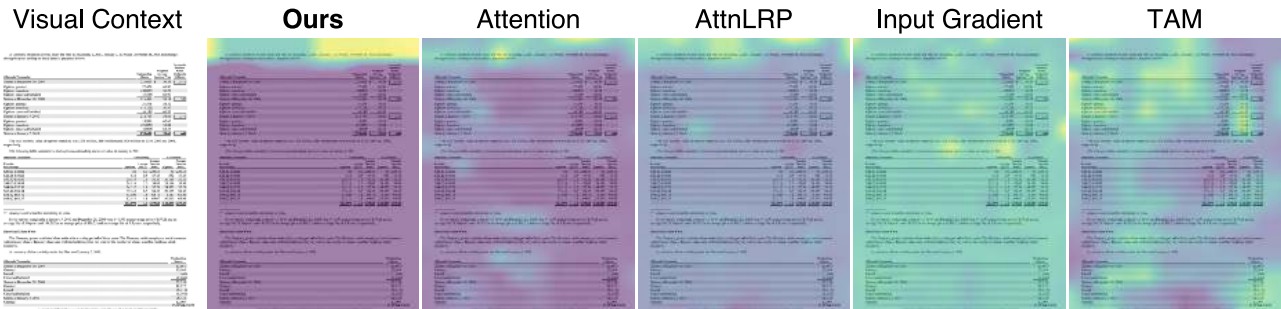

**Query:** What is the total intrinsic value of options exercised in 2010? Find the answer in the document image.
**Response:** [...] Wait, no, let's parse the first table, Wait, the first part says: "A summary of option activity under the Plan as of January 2, 2011, January 3, 2010 and December 28, 2008 and changes during the years ending on those dates is presented below. [...]

### Code

| Visual Context | **Ours** | Attention | AttnLRP | Input Gradient | TAM |
|---|---|---|---|---|---|

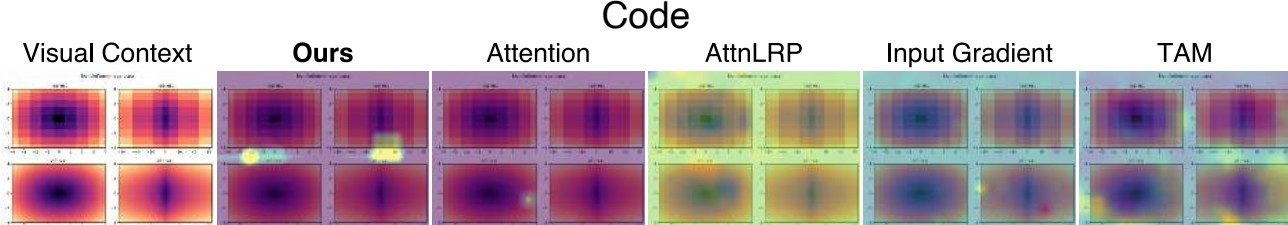

**Query:** You are an expert Python developer who specializes in writing matplotlib code based on a given picture. Now, please give me the matplotlib code that reproduces the picture below.
**Response:** [...] The title is "NonUniformImage class". First, I need to understand what each subplot represents. [...] The bottom row labeled "bilinear" on left and "bilinear" on right. Looking at the color maps: they seem to be using a colormap that goes from red to purple to black. [...]

*Figure 26.* **Additional qualitative results: Qwen3-VL on general/document/code reasoning.**

# Model: Qwen3VL-8B-Thinking

## Math

| Visual Context | **Ours** | Attention | AttnLRP | Input Gradient | TAM |

**Query:** The picture shows 2 mushrooms. What is the difference between their heights? Provide your answer.
**Response:** The problem is about finding the difference in height between two mushrooms. [...] Let's check the scale. The right side has marks from 0 to 12. So, the tops of the two mushrooms— and the taller one's top is at 12? [...]

## Science

| Visual Context | **Ours** | Attention | AttnLRP | Input Gradient | TAM |

**Query:** Based on the latitude and geography of this map, which region is most likely to have a tropical monsoon climate suitable for wet-rice cultivation? Please answer based on given image. Options: A) Northern Europe B) Southeast Asia C) North Africa D) Central America
**Response:** The user is asking for a 'tropical monsoon climate' and 'wet-rice cultivation'. This requires a region with high humidity, heavy rainfall seasons, and specific latitudinal conditions. Wait, I am scanning the map. I am locking my focus on the Indochina Peninsula, Southern China, and the darker archipelagos of the Pacific (Thailand, Vietnam, Philippines). These coordinates correspond to the tropical belt known for monsoon cycles. [...]

*Figure 27.* **Additional qualitative results: Qwen3-VL on math/science reasoning.**

## Model: GLM-4.1V-9B-Thinking

### General

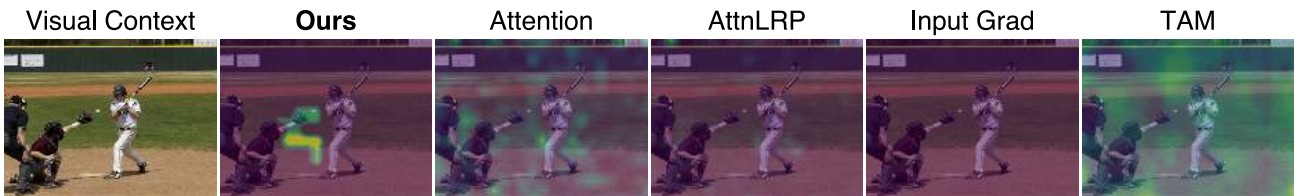

**Query:** On which side of the photo is the catcher? Answer the question based on what you see in the image.
**Response:** Got it, let's look at the image. The catcher is the one in the maroon uniform, crouched down near home plate, with the catcher's mitt extended. He's glove is on the left side of the batter, so his position relative to the photo is the left side. Wait, let me check again. The batter is in the center-right, the catcher is to the left of the batter, so the catcher is on the left side of the photo.

### Document

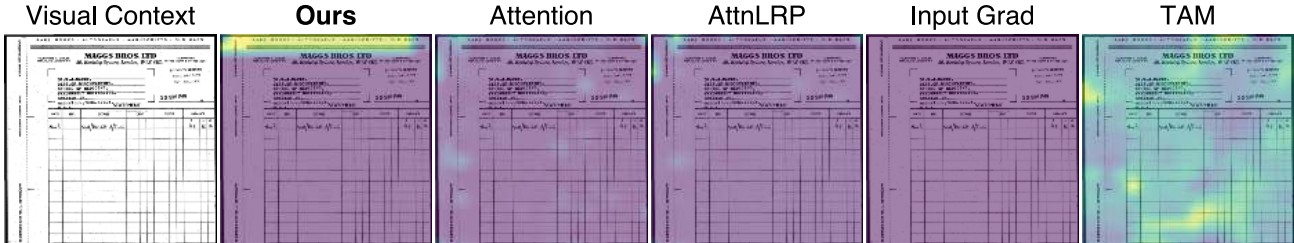

**Query:** What type of maps do they specialize in? Find the answer in the document image.
**Response:** So, let's look at the top of the document. The header has "RARE BOOKS • AUTOGRAPHS • MANUSCRIPTS • OLD MAPS". So the type of maps they specialize in is "OLD MAPS". I need to confirm that. The first line under the company name has those categories, so "OLD MAPS" is one of them.

### Code

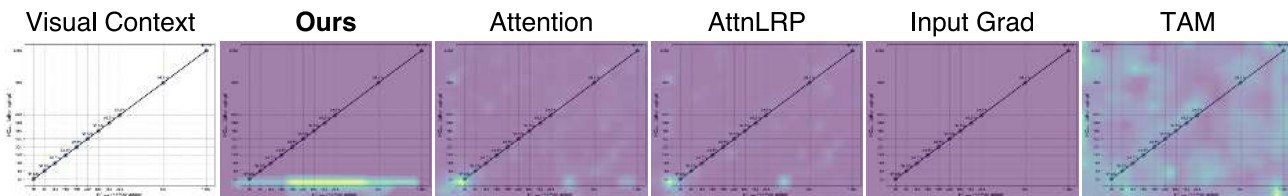

**Query:** You are an expert Python developer who specializes in writing matplotlib code based on a given picture. Now, please give me the matplotlib code that reproduces the picture below.
**Response:** [...] The plot is a scatter plot with a line of best fit, probably a linear regression, since the points seem to follow a straight line. First, I need to identify the data points. Let's look at the x and y coordinates. The x-axis has ticks at 0K, 8K, 16K, 24K, 32K, 40K, 48K, 56K, 64K, 96K, 128K. [...]

*Figure 28.* **Additional qualitative results: GLM-4.1V on general/document/code reasoning.**

# Model: GLM-4.1V-9B-Thinking

## Math

| Visual Context | **Ours** | Attention | AttnLRP | Input Grad | TAM |
|---|---|---|---|---|---|

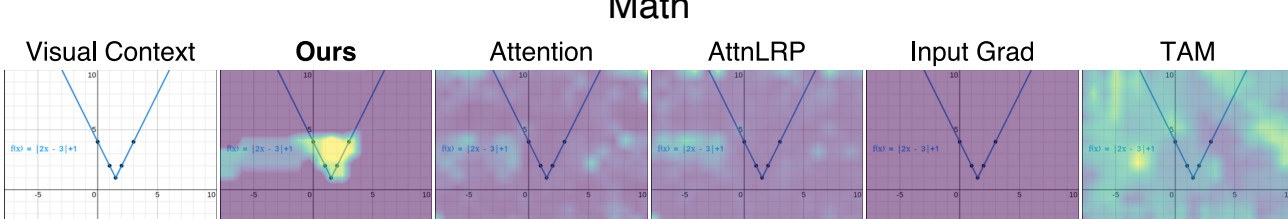

**Query:** Question:
The derivative of f(x) at x=2 is ____ that at x=5 Options: (A) larger than (B) equal to (C) smaller than Select the best answer choice.
**Response:** Got it, let's tackle this problem step by step. First, we need to find the derivative of the function f(x) = |2x - 3| + 1. The absolute value function has a corner (a point where the derivative doesn't exist or changes) at the point where the inside is zero, so 2x - 3 = 0 → x = 3/2. So the function is a V-shape with the vertex at x = 3/2. For x > 3/2, the expression inside the absolute value is positive, so |2x - 3| = 2x - 3. [...]

## Science

| Visual Context | **Ours** | Attention | AttnLRP | Input Grad | TAM |
|---|---|---|---|---|---|

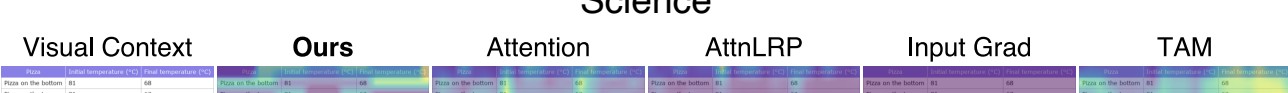

**Query:** During this time, thermal energy was transferred from () to (). Options: (A) the surroundings . . . each pizza (B) each pizza . . . the surroundings  Select the best answer choice.
**Response:** Got it, let's analyze this. First, we need to understand heat transfer. The initial temperatures of both pizzas are higher than the final ones, which means they lost thermal energy. Got it, let's analyze this. [...]

*Figure 29.* **Additional qualitative results: GLM-4.1V on math/science reasoning.**

# Model: GLM-4.1V-9B-Thinking

## General

| Visual Context | **Ours** | Attention | AttnLRP | Input Grad | TAM |
|---|---|---|---|---|---|

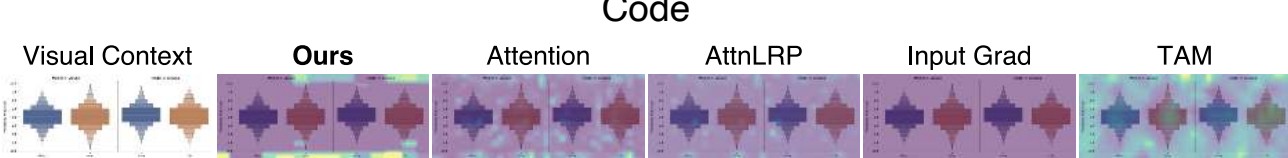

**Query:** Is the hat wet? Answer the question based on what you see in the image.
**Response:** Got it, let's look at the image. The boy's hat is a blue cap with a yellow logo. There's no visible wetness like water droplets or a shiny, damp appearance. The hat looks dry, so the answer would be no.

## Document

| Visual Context | **Ours** | Attention | AttnLRP | Input Grad | TAM |
|---|---|---|---|---|---|

**Query:** What is the year of review of pinnacle ultamet experience? Find the answer in the document image.
**Response:** Got it, let's look at the image. The title says "2011 Review of Pinnacle Ultamet Experience for Drs. Griffin and Mason". So the year of review is 2011. I need to confirm that. The first line has "2011 Review...", so the year is 2011.

## Code

| Visual Context | **Ours** | Attention | AttnLRP | Input Grad | TAM |
|---|---|---|---|---|---|

**Query:** You are an expert Python developer who specializes in writing matplotlib code based on a given picture. Now, please give me the matplotlib code that reproduces the picture below.
**Response:** Got it, let's tackle this problem step by step. First, I need to understand the structure of the plot. It's a pair of box plots, one for "story" and one for "rule" under two models: Vicuna and Alpaca. Wait, no, actually, looking closer, each model has two box plots (story and rule), so there are two groups per model. [...]

*Figure 30.* **Additional qualitative results: GLM-4.1V on general/document/code reasoning.**

# Model: GLM-4.1V-9B-Thinking

## Math

| Visual Context | **Ours** | Attention | AttnLRP | Input Grad | TAM |

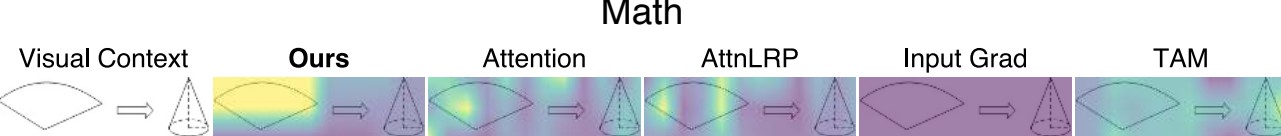

**Query:** Use a sector paper sheet with a central angle of 120.0 and a radius of 6.0 to roll into a conical bottomless paper cap (as shown in the picture), then the bottom perimeter of the paper cap is () Options: (A) 2πcm (B) 3πcm (C) 4πcm (D) 5πcm Select the best answer choice.

**Response:** Got it, let's see. The problem is about rolling a sector into a conical cap, and we need to find the bottom perimeter, which should be the circumference of the base of the cone. [...] The sector has a central angle of 120 degrees and a radius of 6 cm. First, let's find the arc length of the sector.

## Science

| Visual Context | **Ours** | Attention | AttnLRP | Input Grad | TAM |

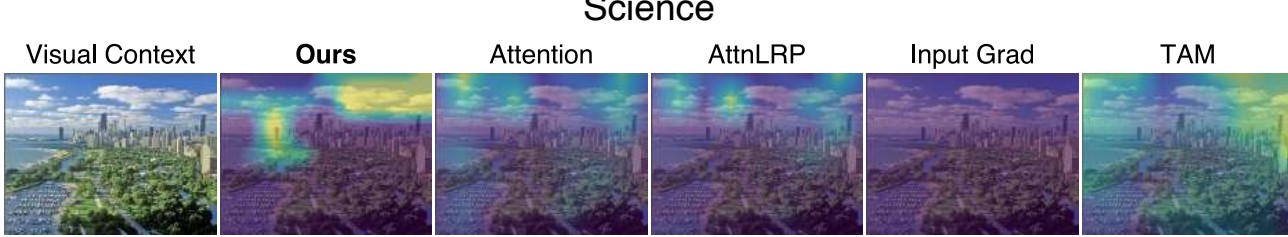

**Query:** Does this passage describe the weather or the climate?
Options:
(A) weather
(B) climate
Select the best answer choice.

**Response:** Got it, let's figure out whether the question is about weather or climate. First, I need to recall the difference: weather is short-term conditions (like a sunny day), climate is long-term average (like a region's usual weather). [...] But wait, the image here—does the image show weather? Since the image is a cityscape with clear sky, maybe the passage describes the weather

*Figure 31.* **Additional qualitative results: GLM-4.1V on math/science reasoning.**

## Model: Cosmos-R1

### General

| Visual Context | **Ours** | Attention | AttnLRP | Input Grad | TAM |
| --- | --- | --- | --- | --- | --- |

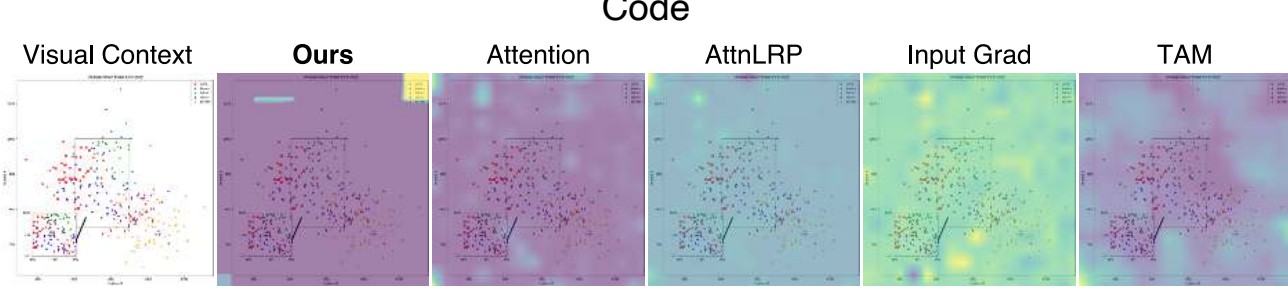

**Query:** Describe the scene in this picture.
**Response:** Okay, Let's break down the picture. The user is asking to describe the photo.. First, I need to recall the details provided in the original image. The image shows a scene with a milk carton, and some donuts the left side.

### Document

| Visual Context | **Ours** | Attention | AttnLRP | Input Grad | TAM |
| --- | --- | --- | --- | --- | --- |

**Query:** What is the page number written in the bottom right corner? Find the answer in the document image.
**Response:** Okay, let to the bottom right corner of the image. The user shows there the page number is "20". I need to confirm that by checking the original document. Looking back again at the provided information, it says: "Page 20" in the bottom right corner. So So, the correct answer should be 20.

### Code

| Visual Context | **Ours** | Attention | AttnLRP | Input Grad | TAM |
| --- | --- | --- | --- | --- | --- |

**Query:** You are an expert Python developer who specializes in writing matplotlib code based on a given picture. Now, please give me the matplotlib code that reproduces the picture below.
**Response:** Okay, let's try to figure out how to recreate this plot using Matplotlib. First, the plot is called "Transportation Mode Efficiency" with two features: X axis, Feature X Looking at the data points, each transportation mode has [...] Let's check the legend. This plot's legend includes cars, buses, bikes, trains, and planes.

*Figure 32.* **Additional qualitative results: Cosmos-R1 on general/document/code reasoning.**

# Model: Cosmos-R1

## Math

| Visual Context | **Ours** | Attention | AttnLRP | Input Grad | TAM |

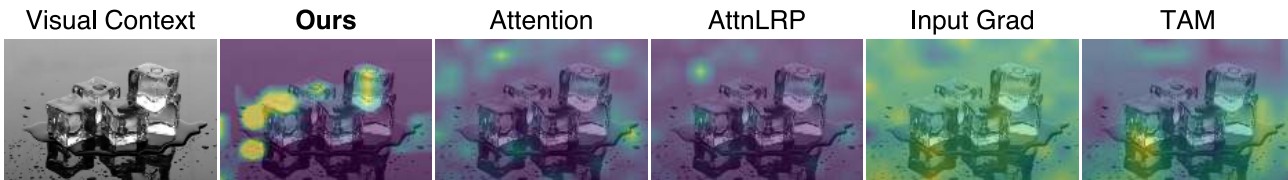

**Query:** Four people can be seated at a square table. How many people at most could be seated if we pushed four tables of this kind together in one row? <image> Provide your answer.

**Response:** Okay, let's try to figure this out. The question is asking how many people could be seated if four square tables like the one shown are pushed together each other in a row. Each table can seat four people. First, I need to visualize the setup. A square table with four sides, each side having a chair. [...]

## Science

| Visual Context | **Ours** | Attention | AttnLRP | Input Grad | TAM |

**Query:** Which term matches the picture?
Options:
(A) chemical change
(B) physical change
Select the best answer choice.

**Response:** Okay, let's see. The user is asking whether the process shown in the scenario matches either a chemical or physical change. First, I need to recall what each type refers to. A chemical change involves a transformation where new matter's composition changes, like burning something or rusting metal. [...] For example, melting ice into water would be a physical change because it's still water, just liquid instead of solid. Looking at the picture: ice cubes melting on a surface. [...]

*Figure 33.* **Additional qualitative results: Cosmos-R1 on math/science reasoning.**

# Model: Cosmos-R1

## General

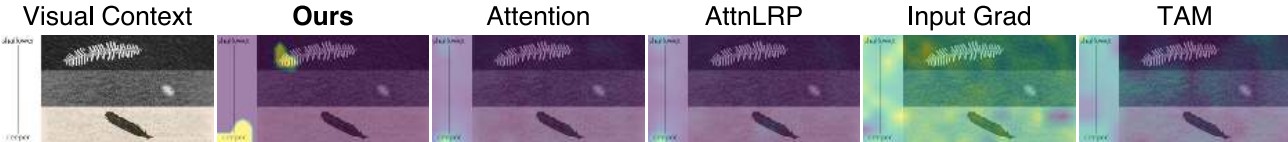

**Query:** Query:
Which of the following fossils is younger? Select the more likely answer. Options: (A) crocodile egg (B) feather
Select the best answer choice.
**Response:** Okay, let to figure out which fossil is younger between the options given. The user mentions that the image shows two feathers and an egg. The top feather is in a shallow area, and the bottom one is in a deeper layer.

## Document

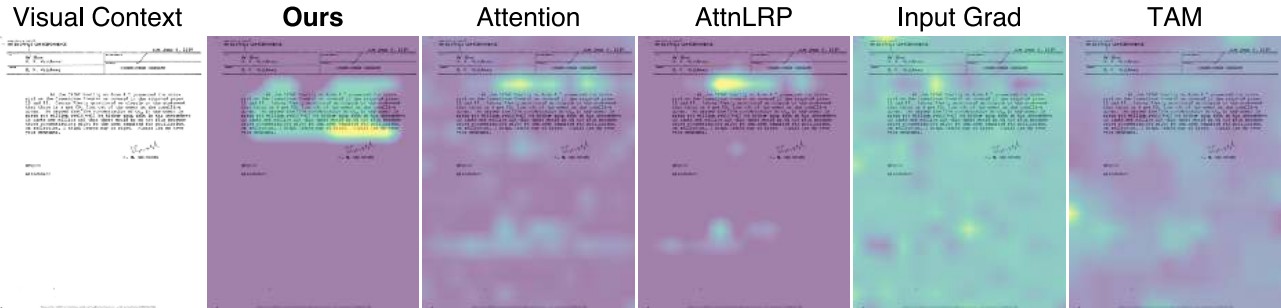

**Query:** What is the subject of this correspondence? Find the answer in the document image.
**Response:** Okay, let me try to figure out the subject of this correspondence. The user shows a scanned document with various sections like "GENERAL", "REFERENCE", "DATE", and "SUBJECT`. [...] I can see that it mentions "GREENHOUSE PROGRAM`. That seems pretty direct. The rest of the document talks about the CRIAC Meeting on June 4 the author presenting material related to the Greenhouse Program.

## Code

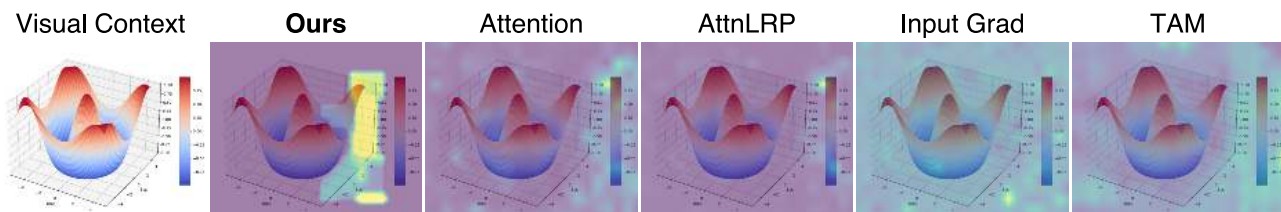

**Query:** You are an expert Python developer who specializes in writing matplotlib code based on a given picture. Now, please give me the matplotlib code that reproduces the picture below.
**Response:** Okay, let's break down the problem. The user wants me to generate a Matplotlib-based plot that matches the one shown in their image. Let me start by understanding the given image and then figure out how to translate that into code. First, looking at the 3D surface plot shown. The x-axis[...] The y-axis [...]. The z-axis is just a numerical value from -1.0 to 1.0.

*Figure 34*. **Additional qualitative results: Cosmos-R1 on general/document/code reasoning.**

# Model: Cosmos-R1

## Math

| Visual Context | **Ours** | Attention | AttnLRP | Input Grad | TAM |

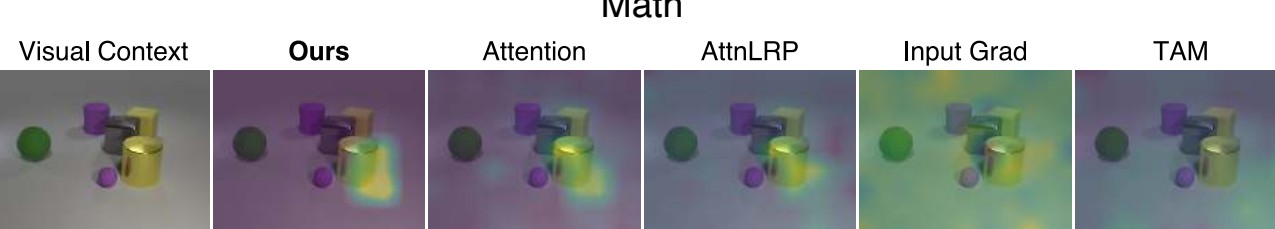

**Query:** Subtract all large rubber spheres. Subtract all big shiny cylinders. How many objects are left? Provide your answer.

**Response:** Okay, let's see. The user is asking to subtract all large rubber spheres and then subtract all big shiny cylinders, and determine how many objects are left. First, I need to recall the video. The scene has has several objects: A large green sphere (rubber) on the the left. A small purple sphere near the center. [...] A big shiny golden cylinder in front of the yellow cube. [...]

## Science

| Visual Context | **Ours** | Attention | AttnLRP | Input Grad | TAM |

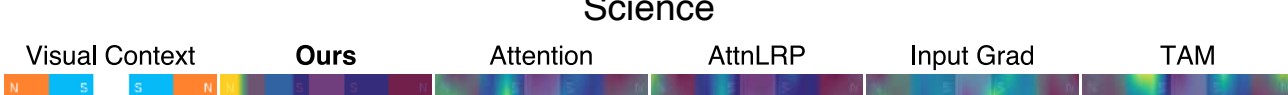

**Query:** Will these magnets attract or repel each other?
Options:
(A) repel
(B) attract
Select the best answer choice.

**Response:** Okay, let's see. The user is asking whether the two magnets shown will attract or repel each other. Let me break this down. First, the setup: there are two magnets. The first one has has as N on the left and S on the right. [...]

*Figure 35.* **Additional qualitative results: Cosmos-R1 on math/science reasoning.**

## Model: MiMo-VL-7B

## General

| Visual Context | **Ours** | Attention | AttnLRP | Input Grad | TAM |
|---|---|---|---|---|---|

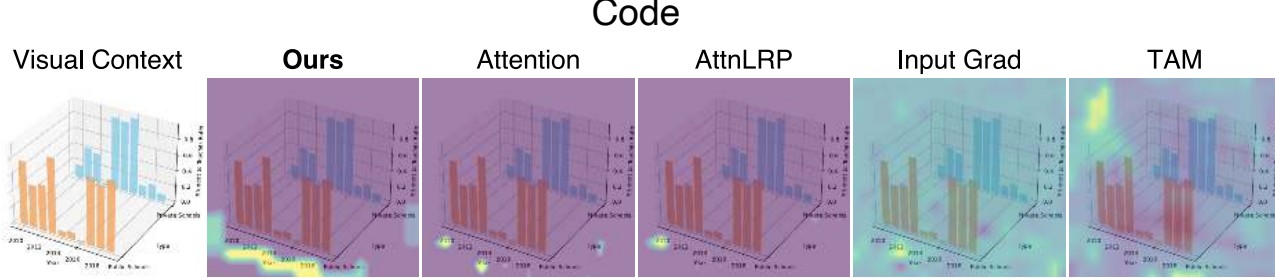

**Query:** Which kind of furniture is in front of the fence? Answer the question based on what you see in the image.
**Response:** So, let's look at the image. There's a fence in the background, and in front of it, there's a table and chairs. The question is asking which kind of furniture. So the furniture in front of the fence is a table and chairs. [...]

## Document

| Visual Context | **Ours** | Attention | AttnLRP | Input Grad | TAM |
|---|---|---|---|---|---|

**Query:** What type of report is this? Find the answer in the document image.
**Response:** So, let's look at the document. In the "Re:" line, it says "ATTENDANCE REPORT FOR MONTH OF: (to be mailed on or before the last working dat of month)" So the type of report is that.

## Code

| Visual Context | **Ours** | Attention | AttnLRP | Input Grad | TAM |
|---|---|---|---|---|---|

**Query:** You are an expert Python developer who specializes in writing matplotlib code based on a given picture. Now, please give me the matplotlib code that reproduces the picture below.
**Response:** Got it, let's try to figure out how to reproduce this 3D bar chart. First, let's analyze the chart. It's a 3D bar plot with two categories: Public Schools and Private Schools, over the years 2010-2018. [...]

*Figure 36.* **Additional qualitative results: MiMo-VL on general/document/code reasoning.**

# Model: MiMo-VL-7B

## Math

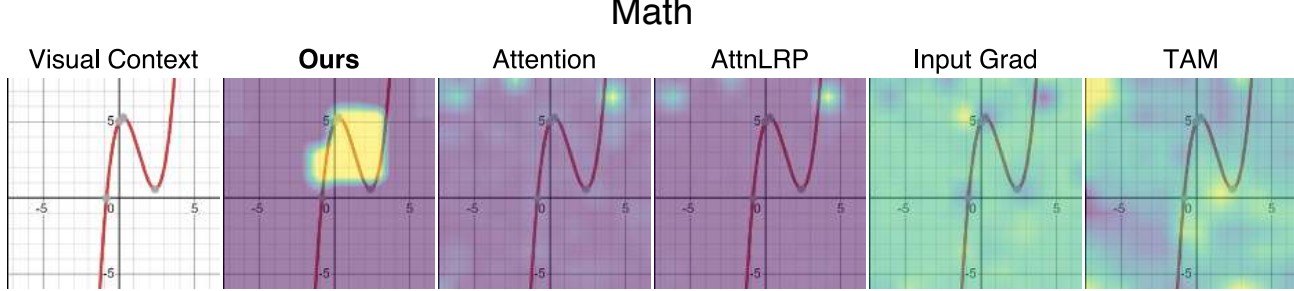

| Visual Context | **Ours** | Attention | AttnLRP | Input Grad | TAM |

**Query:** Question: is f(3) > 0? Options: (A) Yes (B) No Select the best answer choice.
**Response:** Got it, let's try to figure out if f(3) is greater than 0. First, I need to look at the graph. The graph is a red curve on a grid. [...]

## Science

| Visual Context | **Ours** | Attention | AttnLRP | Input Grad | TAM |

**Query:** What would happen to the population of adult spiders if predator ate all the spider eggs?
Options: (A) Adult spider population would remain the same (B) Adult spider population would double. (C) Adults spider population would decrease (D) Adult spider population would increase. Select the best answer choice.
**Response:** Got it, let's analyze this. First, look at the spider life cycle diagram. The cycle is Egg sac → Eggs → Baby spiderlings → Spiderlings → Adult → Egg sac. [...]

*Figure 37*. **Additional qualitative results: MiMo-VL on math/science reasoning.**

## Model: MiMo-VL-7B

### General

| Visual Context | **Ours** | Attention | AttnLRP | Input Grad | TAM |

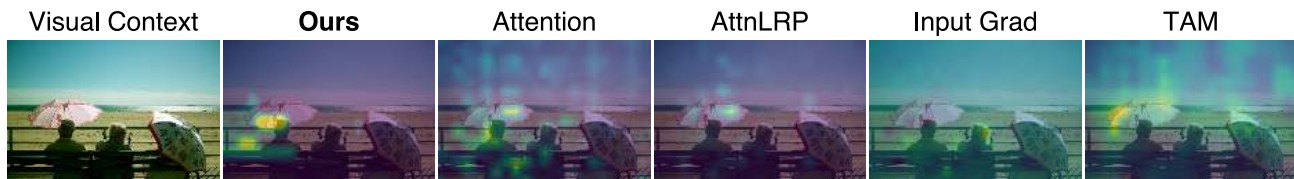

**Query:** What is the man to the left of the glasses doing? Answer the question based on what you see in the image.
**Response:** So, let's look at the image. There's a man on the left, under a pink umbrella. He's sitting on a bench. So what is he doing? He's sitting and probably looking at the beach or the ocean. Let's check the details. The man is seated, so the action is sitting.

### Document

| Visual Context | **Ours** | Attention | AttnLRP | Input Grad | TAM |

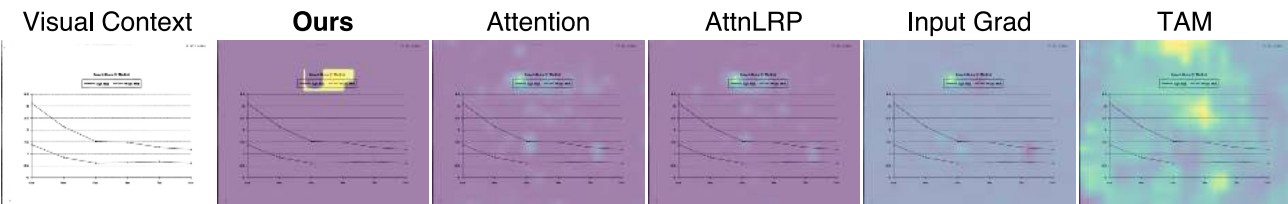

**Query:** What is the share of high market in dec? Find the answer in the document image.
**Response:** So, let's look at the graph. The x-axis is the months, and the y-axis is the share. The "High Mkt" is the line with the plus sign. Find December on the x-axis, then go up to the High Mkt line. The value there is 1.5.

### Code

| Visual Context | **Ours** | Attention | AttnLRP | Input Grad | TAM |

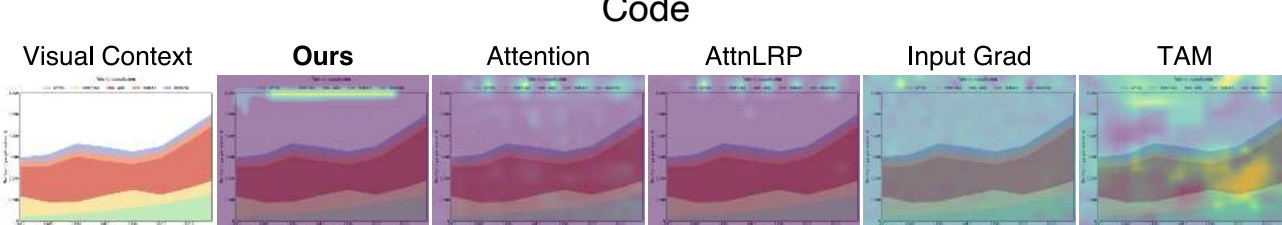

**Query:** You are an expert Python developer who specializes in writing matplotlib code based on a given picture. Now, please give me the matplotlib code that reproduces the picture below.
**Response:** Got it, let's figure out how to reproduce this stacked area chart. First, I need to understand the data. The chart is about world population by region from 1950 to 2010. The regions are africa, americas, asia, europe, oceania. [...]

*Figure 38.* **Additional qualitative results: MiMo-VL on general/document/code reasoning.**

# Model: MiMo-VL-7B

## Math

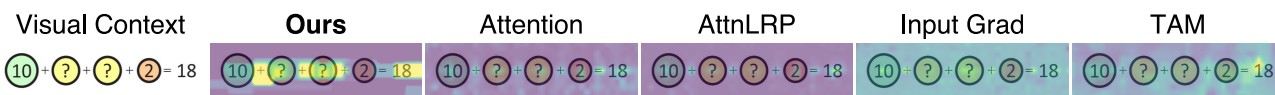

| Visual Context | **Ours** | Attention | AttnLRP | Input Grad | TAM |

**Query:** Question:
The two markers with a question mark have the same number. <image> Which number do you have to put instead of the question mark so that the calculation is correct? Provide your answer.
**Response:** Got it, let's see. And the two question marks are the same number. Let's call that number x. So the equation is 10 + x + x + 2 = 18. [...]

## Science

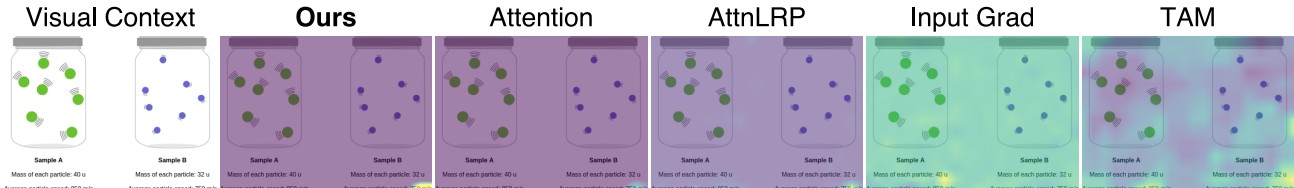

| Visual Context | **Ours** | Attention | AttnLRP | Input Grad | TAM |

**Query:** The diagrams below show two pure samples of gas in identical closed, rigid containers. Each colored ball represents one gas particle. Both samples have the same number of particles. Compare the average kinetic energies of the particles in each sample. Which sample has the higher temperature?
Options: (A) neither; the samples have the same temperature (B) sample A (C) sample B Select the best answer choice.
**Response:** Got it, let's tackle this problem step by step. First, I need to recall what average kinetic energy (KE) of gas particles depends on. The formula for the average kinetic energy of a gas particle is related to temperature, but also to mass and speed. The kinetic energy of a particle is $(1/2)mv^2$, so average KE would be $(1/2)m(v\_avg)^2$. [...] Let's list out the given information for each sample: Sample B: - Average speed (v_B) = 750 m/s

*Figure 39.* **Additional qualitative results: MiMo-VL on math/science reasoning.**

