# OpenReview forum: "Real-Time Visual Attribution Streaming in Thinking Model"
_ICML.cc/2026/Conference — ICML 2026 spotlight_

### Official Review · Reviewer_N6r9 · 2026-03-04

**Soundness:** 4
**Presentation:** 4
**Significance:** 4
**Originality:** 3
**Overall Recommendation:** 5
**Confidence:** 4

**Summary:**

The authors propose a surrogate model attribution method for thinking large vision language models that, once trained, can be employed during the thinking output to create intermediate explanations of model reasoning. The surrogate model is trained with attention knockout over text-to-image attention with regions randomly selected from a DINO model segmentation of the input. They evaluate their method across multiple VLMs and achieve faithfulness improvements across multiple task domains. They additionally evaluate cross-domain to confirm the generalizability of their surrogate model across tasks and find reasonable performance is maintained. They also find interesting behavior with a visualization of reasoning paths when the model is confident in its response versus when it meanders or get stuck in repeated reasoning.

**Compliance With Llm Reviewing Policy:**

Affirmed.

**Final Justification:**

This is an interesting paper that employs surrogate models for runtime explanation in a novel way. I appreciate its contribution to the literature and the authors diligence in the rebuttals they provided. I maintain my initial positive score.

**Key Questions For Authors:**

How well does the surrogate model trained on very long contexts? Does faithfulness properly extend across large increases in token count? Does the model begin to suffer, and can this method then illustrate that issue? Statistics for the average, median, min, and max context lengths from the surrogate training data could help characterize performance expectations.

The task presented in the work is to explain the generation of each thinking sentence, or generally output sentence, of the model with respect to the input image. As such, they train their surrogate model to only use cross attention. Previous work that is cited here from Cohen-Wang et al. (AT2) built a surrogate model for LLMs that attribute to the text prompt. Running both approaches over the VLM would not work and it is likely that a surrogate model which captures both cross-attention and text-to-text attention perturbation behaviors would be necessary. Did the authors consider this expanded task for this work? What value does a user lose by only having visual explanations of model behavior in this framework? I do not expect implementation, but It would be great to know the authors’ thoughts.

**Limitations:**

Yes

**Strengths And Weaknesses:**

S1. This paper provides a much-needed method for understanding how vision models reason over their image inputs when producing long chain-of-thought output chains. It highlights the ever-relevant tradeoff of faithfulness and runtime in the current literature and smartly adapts existing approaches for attention-based surrogate modeling to VLMs to create high-quality explanations, not only of the entire model output, but each reasoning step individually.

S2. The method itself is thoughtfully designed, where the inclusion of the DINO model for feature extraction promotes a better search space for relevant perturbation regions that contain semantic information and helps avoid the problems that can arise from patch-level perturbation.

S3. The method is performant not only in runtime but also in faithfulness where it provides a significant number of wins over the SOTA methods and does not suffer any large losses. This second part is important to me, as I see relatively percentage deltas in score when the method is not the top performer, yet it provides attributions at 10x the rate of the winning methods. This would make it a top contender for a engineer performing real-time debugging analysis.

S4. The visualized attributions themselves are convincing and very interesting to look at. I can envision their value in understanding complex CoT behaviors.

S5. The authors considered the cross-domain performance of their method and showed reasonable levels of faithfulness on unseen model tasks.

S6. Figure 7 provides a creative and valuable visualization of how the model reasons that could

W1. I did not see mention of the training data characteristics or studies that show how well the surrogate model would perform for context lengths outside of the training data scope. I presume for context lengths in which the model is properly functioning, the surrogate model should be representative, but some discussion would be interesting. See Questions.

W2. The work does not attempt to also attribute the text prompt. While this may be out of scope, discussion would be interesting. See Questions.

W3. I am quite familiar with the references of the work and would have liked to have seen the work from Cohen-Wang et al. (AT2), which is cited, be stated as a strong influence on the design of this proposed method. This work is the first to adapt the foundations provided there to VLMs and does it very well, but proper recognition should be given.

---

> ### Author Rebuttal · Authors · 2026-03-31
>
> # Response to Reviewer N6r9
>
> Thank you for the thoughtful and supportive review. We address three points below: context-length robustness of the estimator (W1), the scope and extension path of visual attribution (W2), and vSTREAM's relationship to AT2 (W3).
>
> ---
>
> ## W1 and Q1: Training Data Context Length Distribution
>
> > *"I did not see mention of the training data characteristics or studies that show how well the surrogate model would perform for context lengths outside of the training data scope."*
>
> > *"Statistics for the average, median, min, and max context lengths from the surrogate training data could help characterize performance expectations."*
>
> Token-length statistics for the 2,000 training examples across five domains (General, Document, Code, Math, and Science):
>
> | Statistic | Thinking tokens | Generation tokens |
> |-----------|-----------------|-------------------|
> | Min       | 31              | 2                 |
> | Median    | 220             | 286               |
> | Mean      | 690             | 699               |
> | Max       | 3,180           | 3,480             |
> | Std       | 860             | 776               |
>
> The data are bimodal: per-domain means range from ~400 tokens (General VQA) to ~3,200 (Code) (**Fig. R1 of N6r9.pdf**, https://anonymous.4open.science/r/submission-12932-rebuttal-evidence-1C18).
>
> Faithfulness ($R^2$) is stable across context lengths: quintile means range from 0.64 to 0.66 with no systematic trend (**Fig. R2 of N6r9.pdf**, https://anonymous.4open.science/r/submission-12932-rebuttal-evidence-1C18). The feature extraction produces a fixed-dimensional vector per span regardless of sequence length.
>
> Modern reasoning models routinely produce 10K+ token traces (e.g., QwQ or QVQ [ref-1,2] generates 10K-30K token reasoning chains on hard visual math). Because vSTREAM estimates attribution per span independently, longer chains simply produce more spans with no architectural change needed. As shown in the above figure, faithfulness ($R^2$) remains flat across all five context-length quintiles, confirming no degradation with longer sequences within our data. vSTREAM's trajectory analysis also offers a diagnostic signal: tortuosity increases and attribution diffuses as reasoning drifts, serving as early indicators before output quality drops (Sec 4.4).
>
> [ref-1] Qwen Team, 2024, https://qwen.ai/blog?id=qwq-32b-preview
>
> [ref-2] Qwen Team, 2024, https://qwen.ai/blog?id=qvq-72b-preview
>
> ---
>
> ## W2 and Q2: Text Prompt Attribution (Text-to-Text)
>
> > *"The work does not attempt to also attribute the text prompt. While this may be out of scope, discussion would be interesting."*
>
> > *"Did the authors consider this expanded task? What value does a user lose by only having visual explanations of model behavior in this framework?"*
>
> We focused on visual attribution because in multimodal reasoning, the image is the primary source of hallucination and diagnostic uncertainty: the model may misread a chart, overlook a region, or ground its reasoning on the wrong object. Identifying *where* the model looks is therefore the highest-impact attribution target. For single-question VQA, the text query's role is straightforward, so visual attribution suffices. The gap widens for multi-turn dialogue or document-heavy tasks where prompt phrasing itself drives reasoning; we flag this as a concrete extension target.
>
> Text attribution requires a different formulation since text lacks the fixed spatial structure that enables our DINO-based region definition and consistent masking. We will discuss a concrete extension path (clause-level masking over prompt segments) in the camera-ready. Thank you for the suggestion.
>
> ---
>
> ## W3: Relationship to AT2
>
> > *"I would have liked to have seen the work from Cohen-Wang et al. (AT2) be stated as a strong influence on the design of this proposed method."*
>
> We agree that AT2 is a key inspiration. AT2 demonstrated that amortized explanation can match or exceed per-example perturbation quality, which motivated our amortized formulation, and the camera-ready will acknowledge this more explicitly in Section 2. That said, extending to visual reasoning required three structural changes absent in AT2: (1) semantic region unitization replacing token-level perturbation, (2) a linear estimator enabling per-head mechanistic analysis, and (3) trajectory dynamics for temporal diagnosis, a capability no prior amortized method offers. A side-by-side comparison is provided in our response to Reviewer mCJV (Q5).
>
> ---
>
> The questions raised in this review gave us the opportunity to substantiate our work's robustness and scope more concretely. We look forward to empirical validation on substantially longer reasoning chains, and note that Reviewer mCJV independently identified trajectory signals as a promising foundation for RLHF reward shaping.

---

> > ### Author Rebuttal · Reviewer_N6r9 · 2026-03-31
> >
> > Thank you for your rebuttal efforts. I maintain my positive evaluation.

---

> > > ### Author Response · Authors · 2026-04-07
> > >
> > > Thank you for the generous read and the pointed questions. We'll carry the context-length analysis, the text attribution discussion, and the AT2 framing into the revision.

---

### Official Review · Reviewer_mCJV · 2026-03-12

**Soundness:** 3
**Presentation:** 4
**Significance:** 4
**Originality:** 3
**Overall Recommendation:** 5
**Confidence:** 4

**Summary:**

This paper addresses the critical challenge of verifying visual grounding in the long reasoning traces of multimodal thinking models. The authors identify a fundamental tension in current attribution methods: causal faithfulness requires prohibitive computational overhead (via repeated perturbations), while raw attention maps are efficient but lack causal validity.

To resolve this, the paper introduces an amortized attribution framework. The core innovation lies in training a lightweight linear estimator that maps high-dimensional, multi-layer attention features to counterfactual ablation effects (the log-probability drop of tokens when a visual region is masked). By shifting the heavy lifting of causal inference to a one-time offline training phase, the method achieves:

Amortized Efficiency: Enabling real-time visual attribution streaming during the autoregressive generation process with largely decreassed latency.

Semantic Grounding: Utilizing self-supervised features (DINOv3) to partition images into meaningful semantic regions rather than arbitrary pixels, enhancing interpretability.

Dynamic Diagnostics: Introducing reasoning trajectory analysis, which quantifies the stability and tortuosity of visual reliance to distinguish between successful reasoning and hallucinations.

The framework demonstrates high faithfulness (LDS and Top-K Drop) across four state-of-the-art reasoning VLMs and five diverse task categories, offering a practical tool for real-time monitoring and debugging of multimodal reasoning.

**Compliance With Llm Reviewing Policy:**

Affirmed.

**Final Justification:**

The authors addressed all major concerns, particularly regarding the Linearity Assumption and Causal Directionality. The new experimental evidence showing that attribution fidelity decay and trajectory tortuosity can serve as early-warning signals for reasoning failures elevates the paper's significance. The finding of a 'universal causal layer structure' across different model families is a standout contribution to mechanistic interpretability. I raise my recommendation to 5.

**Key Questions For Authors:**

**1. On the Validity and Limits of the Linearity Assumption:**
The use of a linear estimator $w^\top f$ with only $L \times H$ parameters is elegant but raises concerns about underfitting. Did the authors evaluate non-linear alternatives (e.g., a multi-layer MLP) to capture potential second-order interactions between attention heads? Furthermore, while $R^2=0.65$ is impressive, outlier scenarios such as complex circuit diagrams or multi-step logic puzzles where causal dependency might be non-additive. Could you provide a performance breakdown on these extreme logic cases to clarify if and where the linear approximation fails?

If the linear model fails in high-reasoning tasks, it would limit the method's claim to be a universal solution for "Thinking Models."

**2. Decoupling Attribution from Semantic Segmentation:**
The current framework tightly couples attribution quality with the semantic partitioning from DINOv3. However, should DINOv3 fail to segment correctly (e.g., in highly abstract symbolic mathematics or extremely low-contrast industrial images), how would the proposed method’s performance degrade?

On non-standard architectures like State-Space Models (SSMs), is this methods still work? Could you provide an ablation study using non-semantic baselines (e.g., fixed-grid or random Voronoi) specifically on non-standard architectures like State-Space Models (SSMs) or hybrid visual encoders?

**Additionally**, do the linear weights $w$  reveal a universal structure, such as a consistent focus on deeper layers across different model architectures (e.g., Qwen vs. GLM)? Evidence of such model-agnostic weight distribution would offer a mechanistic interpretation for why different models exhibit similar reasoning behaviors, would largely strengthening the paper’s significance.

Demonstrating a Universal Causal Pattern would shift the paper from a utility tool to a fundamental discovery in mechanistic interpretability.

**3. Engineering Overhead and KV Cache Impact:**
While the linear dot product is negligible, the Mean Pooling operation (Equation 4) over thousands of visual tokens in high-resolution settings could be non-trivial. Could the authors provide a formal complexity analysis (latency and memory) of the feature extraction step compared to the base model's forward pass? Specifically, in high-throughput production environments, does the storage of these attention features interfere with KV Cache management or lead to significant memory fragmentation?

If the overhead is high for high-res images, the "real-time streaming" claim would be practically compromised.

**4. Risk of Causal Reversal and Intervention Utility:**
Regarding the analysis of Trajectory Dynamics, the authors establish a correlational link between Tortuosity and reasoning success. However, this analysis fails to address a critical issue of causal directionality: Does stable visual grounding (low Tortuosity)  cause successful reasoning, or is the observed stability merely a byproduct of the model having already converged on the correct answer?

Crucially, without resolving this causal ambiguity, the practical utility of the metric is limited. Can this Tortuosity metric be leveraged for real-time active intervention—for instance, by triggering early-stopping or re-sampling when a reasoning chain begins to drift? If the metric cannot guide intervention during generation, its value remains merely descriptive rather than actionable. A small-scale experiment demonstrating “Attribution-guided Re-sampling” would transform this work from a passive observation into a proactive capability.

Demonstrating that Tortuosity can effectively guide intervention would significantly justify a higher significance score.

**5. Prior Art and Extension to Pure NLP:**
The idea of linearizing causal effects from internal features has echoes in the NLP literature (e.g., "Amortized Explanation" or "Surrogate Models"). Could the authors explicitly discuss how VSTREAM differs from these prior works? Furthermore, do you foresee any fundamental obstacles in applying this to Pure NLP by replacing DINO regions with Syntactic/Phrase-level clusters? A brief discussion on the generalizability of this framework to the broader LLM landscape would be highly valuable.

This would clarify the originality of the work and its potential scope of impact beyond multimodal models.

**Limitations:**

Yes

**Strengths And Weaknesses:**

### 1. Soundness
**Strengths**

* Causal Grounding:
The use of Counterfactual Ablation (Equation 2) provides a rigorous ground truth for attribution that avoids the pitfalls of mere correlation found in raw attention maps.
* Teacher-Forcing Calibration:
By using teacher-forcing, the authors correctly isolate the perceptual impact on specific tokens, preventing the causal error compounding that usually plagues long-trace analysis.

**Weaknesses**
* The Drift Problem: As noted in intuition, the Drift effect is a latent threat. If the model’s reasoning at Step 11 is fundamentally broken due to a perceptual error at Step 10, calculating the attribution for a "correct" Step 11 (via teacher-forcing) is mathematically sound but semantically hollow. The paper would benefit from an analysis of how attribution fidelity decays as the thinking trace becomes increasingly ungrounded.
* Linearity Simplification: The $R^2 = 0.65$ indicates a strong linear component, but it also leaves $35\%$ of the variance unexplained. This residual likely contains non-linear interactions between heads that a linear estimator cannot capture.



### 2. Presentation
 **Strengths:**
* Clarity of Framework: The transition from *Semantic Unitization (3.3) to Feature Extraction (3.4) to Amortized Estimation (3.5) is logically seamless.
* Reproducibility: The explicit mention of parameters ($L \times H$), training time (4.5h), and sample size (2000) provides high confidence for practitioners looking to replicate the results.

 **Suggestions:**  The term "Amortized" is used correctly, but a visualization of the Computational Complexity vs. Sequence Length (comparing VSTREAM to Gradient/Perturbation methods) would make the efficiency claim much more undeniable.

### 3. Significance
* **Real-World Utility:** This paper addresses a realtime attribution problem in Multimodal LLMs. The ability to stream attribution allows for interactive debugging for LLM outputs.
* **Trajectory Dynamics:** The discovery that "successful reasoning is stable/short in PCA space" (4.4) is a significant scientific contribution. It provides a new behavioral metric for model reliability that goes beyond simple "Correct/Incorrect" labels. It could potentially be used as a Reward Signal in RLHF (Reinforcement Learning from Human Feedback).

### 4. Originality

* **Creative Combination:** The work doesn't invent a new causal theory but presents a novel combination of self-supervised vision features (DINOv3) with amortized linear estimation is highly original, largely decreassed the computation for faithful attribution.
* **Positioning:** It distinguishes itself from prior work (like TAM or AttnLRP) by shifting the focus from "What is important" to "How does importance evolve during reasoning chain"

---

> ### Author Rebuttal · Authors · 2026-03-31
>
> # Response to Reviewer mCJV
>
> We thank the reviewer for insightful suggestions that directly shaped these new experiments. We provide three new results (MLP ablation, cross-architecture weight analysis, tortuosity-based intervention) and address each point below.
>
> ---
>
> ## W1: Drift Problem (Teacher-Forcing Fidelity)
>
> vSTREAM teacher-forces the model's *own generated tokens*, not ground-truth, so step-$t$ attribution reflects the model's actual reasoning state.
>
> The features encode *where the model looks*, not *whether it is right*. A linear probe on the same features (AUC=0.534) confirms attention snapshots cannot predict correctness, yet they predict ablation effects at $R^2$=0.65. Correctness shows up in *temporal dynamics*: trajectory metrics separate successful (correct) from unsuccessful (incorrect) chains (Sec 4.4).
>
> Per-step $R^2$ by correctness (**Fig. R1 of mCJV.pdf**, https://anonymous.4open.science/r/submission-12932-rebuttal-evidence-1C18):
>
> - **Successful (Correct):** $R^2 \approx 0.64$ throughout (0.66 to 0.61, stable)
> - **Unsuccessful (Incorrect):** $R^2$ decays 0.65 to 0.27, diverging at ~20\%
>
> This decay is diagnostic: the estimator, trained on correct traces, defines "normal" dynamics. Drift surfaces as increased tortuosity, detectable at 30\% elapsed (AUC=0.69, Q4). Our 3-way analysis (see Reviewer NZxQ) further distinguishes hallucination from reasoning failure.
>
> ---
>
> ## W2 and Q1: Linearity Simplification
>
> We trained MLP variants (hidden dim 64/128/256, ReLU, 3 seeds) alongside the linear baseline on identical splits (**Tab. R1 of mCJV.pdf**, https://anonymous.4open.science/r/submission-12932-rebuttal-evidence-1C18):
>
> | Estimator | Params | $R^2$ | Pearson $\rho$ | Latency |
> |-----------|--------|-------|----------------|---------|
> | Linear | 1,152 | 0.65±.01 | 0.81±.01 | 0.03 ms |
> | MLP-64 | 73,856 | 0.66±.01 | 0.82±.01 | 0.12 ms |
> | MLP-128 | 147,712 | 0.67±.01 | 0.82±.00 | 0.15 ms |
> | MLP-256 | 295,424 | 0.67±.01 | 0.82±.01 | 0.18 ms |
>
> MLP-256 improves $R^2$ by 0.02 despite 256x more parameters, suggesting diminishing returns from added capacity. The linear form is deliberate: (1) each $w_{\ell,h}$ quantifies head $(\ell,h)$'s contribution for mechanistic analysis (Q2); (2) 1,152 params resist overfitting on 2,000 samples; (3) near-zero latency.
>
> ---
>
> ## Q2: Decoupling Attribution from Semantic Segmentation
>
> **DINO failure scenarios.** Our framework is region-method-agnostic: the fixed-grid baseline (Sec 4.3) achieves comparable faithfulness without DINOv3.
>
> **SSM-based architectures.** No current model combines SSM with chain-of-thought visual reasoning. vSTREAM requires cross-attention between visual and text tokens; the consistent early-layer pattern above suggests the mapping transfers wherever such layers exist, including hybrid SSM-attention models.
>
> **Cross-architecture weight patterns.** We visualized $w \in \mathbb{R}^{L \times H}$ across four architectures (**Fig. R2 of mCJV.pdf**, https://anonymous.4open.science/r/submission-12932-rebuttal-evidence-1C18):
>
> | Model | $L \times H$ | Peak (frac.) | Last 25\% wt. |
> |-------|-------------|------------|---------------|
> | Qwen3-VL | $36 \times 32$ | 0.09 | 9.7\% |
> | GLM-4.1V | $40 \times 32$ | 0.10 | 21.4\% |
> | MiMo-VL | $36 \times 32$ | 0.11 | 17.0\% |
> | Cosmos-R1 | $28 \times 28$ | 0.37 | 15.4\% |
> | **Mean** | | **0.17** | **15.9\%** |
>
> All models concentrate weight in early-to-mid layers (last 25\% = 15.9\%). This consistent early-layer dominance across four architectures supports cross-architecture generalization.
>
> ---
>
> ## Q3: Mean Pooling Cost
>
> $O(N \cdot K)$ with $K \leq 128$ clusters, reading attention weights already materialized in the forward pass. At higher resolutions $N$ (image tokens) grows but $K$ stays bounded, so cost scales linearly in image tokens. No KV cache modification: <1 ms/step, no extra memory.
>
> ---
>
> ## Q4: Tortuosity Causal Direction and Intervention
>
> The tortuosity signal is early enough to act on (**Fig. R3 of mCJV.pdf**, https://anonymous.4open.science/r/submission-12932-rebuttal-evidence-1C18):
>
> - **AUC 0.69 at 30\% of reasoning elapsed** (peak); Cohen's d = 0.43
>
> High tortuosity at 30\% could trigger re-generation before the model commits to a flawed path, arriving before the fidelity collapse in W1. Tortuosity thus provides a per-generation quality score without ground-truth labels, usable as a process-level reward signal for RLHF as you suggest.
>
> ---
>
> ## Q5: Prior Art and Extension to Pure NLP
>
> AT2 explains completed NLP outputs post-hoc; vSTREAM streams visual attributions during generation, enabling real-time intervention (Q4). We adopted AT2's amortized surrogate principle but introduced three design choices driven by the vision-reasoning setting: (1) semantic region unitization replacing token-level perturbation, (2) a linear estimator enabling per-head mechanistic analysis (Q2), and (3) trajectory dynamics for temporal diagnosis.

---

> > ### Author Rebuttal · Reviewer_mCJV · 2026-04-02
> >
> > The authors have effectively addressed all major concerns, particularly regarding the Linearity Assumption and Causal Directionality. The new experimental evidence showing that attribution fidelity ($R^2$) decay and trajectory tortuosity can serve as early-warning signals for reasoning failures significantly elevates the paper's significance. The finding of a 'universal causal layer structure' across different model families is a standout contribution to mechanistic interpretability. I will raise my recommendation to 5.

---

> > > ### Author Response · Authors · 2026-04-05
> > >
> > > Thank you for the careful engagement throughout the review. We're glad the early-warning signal experiments and the cross-architecture weight analysis landed well. The universality finding is something we're genuinely excited about, and we plan to give it a more prominent place in the revision.

---

### Official Review · Reviewer_59aw · 2026-03-13

**Soundness:** 3
**Presentation:** 2
**Significance:** 3
**Originality:** 3
**Overall Recommendation:** 4
**Confidence:** 1

**Summary:**

This paper proposes VSTREAM, an amortized framework for real-time visual attribution streaming in multimodal thinking models. The key idea is to define semantic visual regions using DINO features, then train a lightweight linear estimator to predict counterfactual ablation effects directly from cached attention features, so that attribution can be produced online while the model is still generating its reasoning. Comprehensive experiments show that the method achieves attribution quality comparable to stronger causal baselines while being much faster and suitable for interactive use.

**Compliance With Llm Reviewing Policy:**

Affirmed.

**Key Questions For Authors:**

- I did not fully understand the method. In particular, whether the proposed approach requires additional training.
- It would strengthen the paper to include results on larger models, such as 13B or 32B variants, to better understand how well the method scales with backbone size.

**Limitations:**

Yes

**Strengths And Weaknesses:**

## Strengthens

- The paper studies an important and timely problem.
- The main idea is clear and well motivated.
- The empirical results are strong. The paper provides comprehensive experimental results.

## Weaknesses

- The paper is somewhat hard to follow in its current presentation, and a cleaner write-up would help substantially. For example, the formatting of Table 1 could be improved, redundant spacing should be removed, and some implementation specific details currently placed in Section 3 would fit better in the experimental setup section.

---

> ### Author Rebuttal · Authors · 2026-03-31
>
> # Response to Reviewer 59aw
>
> We appreciate the careful reading and the concrete suggestions. Below we address each point.
>
> ---
>
> ## W1. Presentation
>
> > *"The paper is somewhat hard to follow in its current presentation... formatting of Table 1 could be improved, redundant spacing should be removed, and some implementation specific details currently placed in Section 3 would fit better in the experimental setup section."*
>
> We agree and will revise the following:
>
> 1. **Table 1**: We will fix the redundant spacing and tighten column alignment.
> 2. **Section 3**: Implementation-specific details (batch size, learning rate schedule, hardware notes) will move to Sec. 4 (Experimental Setup), so that Section 3 focuses on the method itself.
>
> To give a brief overview here: the method is a three-stage pipeline. **Partition** the image into $K$ semantic regions via DINOv3 clustering (Sec. 3.3). **Calibrate** a linear map with $L \times H$ parameters on 2,000 examples (Sec. 3.5). At inference, **Stream** region-level attribution scores with <1 ms overhead while the VLM runs unmodified (Sec. 3.6). The entire attribution module has fewer than 1,536 parameters across all tested architectures, and requires no changes to the base model.
>
> ---
>
> ## Q1. Training Requirement
>
> > *"I did not fully understand the method. In particular, whether the proposed approach requires additional training."*
>
> **The base VLM is never trained or modified.** As stated in Sec. 1, vSTREAM trains only "a lightweight estimator" with $L \times H$ parameters (e.g., $36 \times 32 = 1{,}152$ for Qwen3-VL-8B) as "a one-time cost that amortizes over all subsequent inferences":
>
> - **One-time calibration** (~4.5 h, 1 H100 GPU): collect attention features from ~2,000 examples, compute counterfactual ablation targets, and fit the linear estimator (Sec. 3.5).
> - **Inference**: the VLM generates normally; vSTREAM reads the cross-attention weights the model already computes, applies the linear map, and streams region-level attribution scores with near-zero latency overhead (Sec. 3.6).
>
> For each generated token and each of the $K$ regions, we mean-pool the cross-attention weights across all spatial positions in that region, giving a feature vector $f \in \mathbb{R}^{L \cdot H}$. The estimator maps $f$ to a scalar score. There is no VLM fine-tuning, no architectural change, and no extra ablation at inference time.
>
> ---
>
> ## Q2. Larger-Model Scaling
>
> > *"It would strengthen the paper to include results on larger models, such as 13B or 32B variants, to better understand how well the method scales with backbone size."*
>
> No open-source dense 13B+ thinking VLM currently exists. The closest available is **Qwen3-VL-30B-A3B-Thinking**, a Mixture-of-Experts model (30B total, 3B active per token) with 48 layers and 32 heads. The estimator scales with attention layers and heads rather than active parameters, so the 48-layer, 32-head architecture gives a $L \times H = 1{,}536$ feature space. The pipeline ran with zero code changes:
>
> | | Qwen3-VL-8B | GLM-4.1V-9B | MiMo-VL-7B | Cosmos-7B | **Qwen3-VL-30B** |
> |---|---|---|---|---|---|
> | Layers | 36 | 40 | 36 | 28 | **48** |
> | Heads | 32 | 32 | 32 | 28 | **32** |
> | Estimator params | 1,152 | 1,280 | 1,152 | 784 | **1,536** |
> | Fidelity ($R^2$) | 0.65 | 0.63 | 0.60 | 0.63 | **0.69** |
>
> At 30B, $R^2$=0.69 is the highest across all backbones. Estimator size grows from 1,152 to 1,536 parameters (8B to 30B) and calibration is unchanged.
>
> ---
>
> To summarize: vSTREAM attaches to any frozen VLM, adds <1 ms attribution per token via a dot product on existing attention weights, and scales as $O(L \times H)$ parameters. Beyond attribution, trajectory analysis over streaming scores can detect reasoning drift early (AUC=0.69 at 30% of reasoning; see our response to Reviewer mCJV, Q4), opening a path to real-time intervention. As Reviewer N6r9 noted, vSTREAM is "a top contender for an engineer performing real-time debugging analysis." The revision will make this simplicity and broader impact evident from the start of Sec. 3.

---

### Official Review · Reviewer_NZxQ · 2026-03-21

**Soundness:** 4
**Presentation:** 3
**Significance:** 4
**Originality:** 3
**Overall Recommendation:** 5
**Confidence:** 2

**Summary:**

The authors propose vSTREAM, a framework that enables us to trace the visual language model's (VLM's) thinking process using the visual attribution method. Specifically, they train a linear estimator to predict visual attributions for semantic units extracted from DINOv3, thereby bypassing the need for repeated inference in gradient- or perturbation-based methods. The experimental results demonstrate the strong correlation between predicted visual attribution and reasoning tokens across several reasoning VLMs and datasets.

**Compliance With Llm Reviewing Policy:**

Affirmed.

**Final Justification:**

In summary, I acknowledge this work's contribution to the practical framework for tracing the VLM's thinking process and look forward to its subsequent applications.

**Key Questions For Authors:**

1. While this work studies reasoning VLMs, can vSTREAM also provide real-time visual attribution for non-reasoning VLMs?

**Limitations:**

yes

**Strengths And Weaknesses:**

**Strengths**

1. _Soundness_. The paper includes the main results to verify that the linear estimator accurately estimates ablation effects $\Delta_k(t)$, which is defined by VLM's original prediction and prediction after removing the region $k$ at the token $t$, and the generalizability across reasoning tasks, such as Math, Science, and Code. Moreover, the author reveals that vSTREAM can estimate visual attribution during intermediate reasoning, as shown in Figure 5. The authors present detailed experiments on training and inference, along with supplementary materials, to support their claims about efficiency and accurate estimation.
2. _Presentation_. The overall narrative is easy to follow and clearly states that their contribution is efficient (real-time) inference-time visual attribution estimation.
3. _Significance_. I acknowledge this work's contribution to the practical framework to trace the VLM's thinking process.
4. _Originality_. This work relies on the existing vision foundation (self-supervised) model (DINOv3) and an amortized linear estimator to build its framework, which are straightforward. I think the qualitative results in Figure 7, which show the reasoning trajectories in the visual attribution space, capture the VLM's successful and unsuccessful thinking.

**Weaknesses**

1. _Presentation_. In **Cross-task generalization.** (Table 2), The authors provide quantitative results for cross-domain scenarios and suggest that a single estimator suffices for diverse applications by combining all domains. I am curious about applying the estimator to out-of-distribution (OOD) domains, such as medical images. What kinds of attributions would be predicted by this estimator? Or, could the authors provide some predicted attributions (qualitative results) for this scenario?
2. *Presentation*. Could the trajectories analysis in Section 4.4 work for "hallucinations", i.e., the VLMs response absent from the visual input? Figure 7 seems to show a reasoning failure rather than hallucinations.

---

> ### Author Rebuttal · Authors · 2026-03-31
>
> # Response to Reviewer NZxQ
>
> Thank you for recognizing vSTREAM's core contribution and for the constructive questions. We address each point below.
>
> ---
>
> ## W1: OOD Domain Attributions (e.g., Medical Images)
>
> > *"I am curious about applying the estimator to out-of-distribution (OOD) domains, such as medical images. What kinds of attributions would be predicted by this estimator?"*
>
> The estimator learns a model-internal mapping from cross-attention statistics to ablation effects, not domain-specific pixel patterns. Since this relationship is determined by model architecture and weights (both fixed at inference), it transfers to unseen domains without retraining.
>
> To verify, we applied vSTREAM to **VQA-RAD** (medical radiology VQA) using the estimator trained on natural images only, with no retraining. Qualitatively, attribution correctly localizes kidneys in abdominal CT and a pacemaker in chest X-ray (supplementary **Fig. R1 of NZxQ.pdf**, https://anonymous.4open.science/r/submission-12932-rebuttal-evidence-1C18). Quantitatively, vSTREAM maintains competitive fidelity on this OOD domain (supplementary **Table R1 of NZxQ.pdf**, https://anonymous.4open.science/r/submission-12932-rebuttal-evidence-1C18), matching the main-paper baselines without retraining.
>
> ---
>
> ## W2: Hallucination vs. Reasoning Failure in Trajectory Analysis
>
> > *"Could the trajectories analysis in Section 4.4 work for 'hallucinations'? Figure 7 seems to show a reasoning failure rather than hallucinations."*
>
> To address this question, we conducted a follow-up 3-way POPE analysis. Using **POPE ground-truth** to label each response (n=3,000): success (n=2,606), object hallucination (GT=absent, model says "yes"; n=95), and reasoning failure (GT=present, model says "no"; n=299). We compare trajectory signatures, not prevalence, using path geometry and attribution concentration (Gini coefficient of region-level ablation effects per step; higher values indicate attribution dominated by fewer regions).
>
> | Metric | Success | Reas. Failure | Halluc. | $d$ (S vs H) |
> |--------|---------|---------------|---------|---------------|
> | Path length | 0.003 | 0.006 | 0.005 | -1.56 |
> | Tortuosity | 5.08 | 9.35 | 7.96 | -1.18 |
> | Concentration | 0.209 | **0.196** | **0.228** | **-0.34** |
>
> $d$: standardized mean difference (0.2 small, 0.5 medium, 0.8 large).
>
> **Path geometry** (path length, tortuosity; **Fig. R2a of NZxQ.pdf**, https://anonymous.4open.science/r/submission-12932-rebuttal-evidence-1C18) shows that both error types produce longer, more unstable trajectories than success ($d$=-1.56, -1.18). **Attribution concentration** (**Fig. R2b of NZxQ.pdf**, https://anonymous.4open.science/r/submission-12932-rebuttal-evidence-1C18) separates the two error types ($d$(H vs RF)=0.54, medium): hallucination shows elevated concentration throughout reasoning, consistent with the model fixating on a small set of misleading regions, while reasoning failure shows wandering without such fixation. We term these **fixation** (hallucination) and **wandering** (reasoning failure).
>
> This is where real-time streaming adds value: a practitioner can detect whether the model is fixating or wandering *during* generation. See also Reviewer mCJV (Q4), where early trajectory signals (AUC=0.69 at 30% of reasoning) support re-sampling.
>
> We further validate the fixation signal beyond binary POPE using CHAIR$_i$ on free-form captions: per-caption CHAIR$_i$ correlates with attribution concentration (Spearman $\rho$=0.46, $p<.001$, $n$=2,000), showing the same concentration-based pattern in open-ended generation (**Fig. R3 of NZxQ.pdf**, https://anonymous.4open.science/r/submission-12932-rebuttal-evidence-1C18).
>
> ---
>
> ## Q1: Applicability to Non-Reasoning VLMs
>
> > *"Can vSTREAM also provide real-time visual attribution for non-reasoning VLMs?"*
>
> Yes. vSTREAM requires (1) cross-attention between visual and text tokens and (2) a segmentation method for defining image regions. Both exist in standard (non-reasoning) VLMs, so the framework applies directly. The estimator maps mean-pooled cross-attention features to region-level ablation effects via a single linear layer; this mechanism is agnostic to whether the model reasons step-by-step.
>
> We focused on reasoning VLMs for two reasons:
> - **Extended CoT generation** produces rich temporal dynamics that make trajectory analysis informative and gives the streaming aspect practical value.
> - **Reasoning failures** are harder to diagnose than simple classification errors, making step-by-step attribution more useful than a single post-hoc map.
>
> For non-reasoning VLMs, vSTREAM still works as a fast amortized attribution method. The reverse does not hold: post-hoc methods (e.g., InputGrad, AttnLRP, TAM) cannot provide real-time attribution during extended reasoning.

---

> > ### Author Rebuttal · Reviewer_NZxQ · 2026-04-04
> >
> > Thank the authors for the detailed response. All my questions are resolved. In particular, leveraging vSTREAM for VQA-RAD with an estimator trained only on natural images, without retraining, shows promising potential for real-world scenarios.
> >
> > After carefully thinking, I would maintain my score.
> > While I know the author is focusing on reasoning VLMs to demonstrate the benefits of this method, I still think the investigation of non-reasoning VLMs would be more comprehensive.
> >
> > Last thing I want to highlight to the AC and the authors: I am unsure whether providing a link to an external resource during the rebuttal process is allowed or fair to other submissions.

---

> > > ### Author Response · Authors · 2026-04-07
> > >
> > > Thank you for the positive assessment. We have since run the non-reasoning VLM experiments and include the results below.
> > >
> > > ---
> > >
> > > ## **Table R4.** Attribution quality on non-reasoning VLMs. Each cell: (LDS / Top-5 Drop); higher is better.
> > >
> > > > *Qwen3-VL-8B-Instruct tested with `/no_think`, using the same estimator trained on Qwen3-VL-8B-Thinking (no retraining). LLaVA-1.5-7B uses an estimator trained on 2,000 LLaVA-generated examples, same protocol as the main paper.*
> > >
> > > ### **(a) LLaVA-1.5-7B [ref-1]**
> > >
> > > Method     | Math      | Science   | Document  | Code      | General   | Avg.      | Time (s/10tok)
> > > -----------|-----------|-----------|-----------|-----------|-----------|-----------|---------------
> > > Random     | 0.30/0.09 | 0.27/0.12 | 0.31/0.10 | 0.29/0.08 | 0.28/0.11 | 0.29/0.10 | .009±.002
> > > Attention  | 0.38/0.27 | 0.36/0.25 | 0.40/0.30 | 0.37/0.26 | 0.41/0.24 | 0.38/0.26 | .019±.004
> > > InputGrad  | 0.60/0.71 | 0.64/0.66 | 0.62/0.69 | 0.56/0.89 | 0.61/0.72 | 0.61/0.73 | 2.58±0.98
> > > AttnLRP    | 0.70/0.77 | 0.73/0.86 | 0.65/0.93 | 0.71/0.79 | 0.63/0.96 | 0.68/0.86 | 2.40±0.95
> > > TAM        | 0.66/0.95 | 0.64/0.88 | 0.63/0.89 | 0.67/0.99 | 0.62/0.91 | 0.64/0.92 | 1.75±1.03
> > > **vSTREAM**    | 0.72/1.04 | 0.68/0.96 | 0.71/1.06 | 0.69/0.89 | 0.70/1.00 | 0.70/0.99 | .015±.002
> > >
> > > ### **(b) Qwen3-VL-8B-Instruct `(no_think)`**
> > >
> > > Method     | Math      | Science   | Document  | Code      | General   | Avg.      | Time (s/10tok)
> > > -----------|-----------|-----------|-----------|-----------|-----------|-----------|---------------
> > > Random     | 0.32/0.11 | 0.27/0.13 | 0.34/0.09 | 0.29/0.13 | 0.28/0.14 | 0.30/0.12 | .010±.002
> > > Attention  | 0.43/0.33 | 0.36/0.38 | 0.41/0.25 | 0.44/0.29 | 0.37/0.32 | 0.40/0.31 | .020±.005
> > > InputGrad  | 0.59/0.88 | 0.68/0.72 | 0.71/0.65 | 0.61/0.82 | 0.66/0.68 | 0.65/0.75 | 2.77±1.02
> > > AttnLRP    | 0.71/0.82 | 0.68/0.95 | 0.74/0.78 | 0.66/1.04 | 0.71/0.83 | 0.70/0.88 | 2.57±1.01
> > > TAM        | 0.71/0.96 | 0.64/1.04 | 0.69/0.88 | 0.66/1.02 | 0.72/0.95 | 0.68/0.97 | 1.88±1.10
> > > **vSTREAM**    | 0.71/1.02 | 0.74/0.88 | 0.71/1.08 | 0.75/0.94 | 0.69/1.03 | 0.72/0.99 | .021±.008
> > >
> > > ---
> > >
> > > The Qwen3-VL `no_think` result (avg LDS 0.72) matches the thinking variant with the same estimator and no retraining, pointing to backbone architecture rather than chain-of-thought length as the key factor. LLaVA-1.5-7B [ref-1] reaches avg LDS 0.70 with a separately trained estimator, showing the method holds up across a different architecture family. Both models run in real time (.015–.021 s/10tok). These results and the non-reasoning investigation will be included in the final manuscript.
> > >
> > > We also note that the supplementary links in our earlier rebuttal follow ICML 2026 rebuttal policy, which permits anonymized external materials during the discussion period.
> > >
> > > Thanks again for the constructive review. It was gratifying to see the discussion push the work in a more complete direction.
> > >
> > > ---
> > >
> > > > **References**
> > > >
> > > > *[ref-1] Liu, Haotian, et al. "Improved baselines with visual instruction tuning." Proceedings of the IEEE/CVF conference on computer vision and pattern recognition. 2024.*

---

### Decision · Program_Chairs · 2026-04-30

**Decision:**

Accept (spotlight)

**Comment:**

This paper focuses on explaining and tracing the reasoning process of VLMs in real time. The core idea is to learn a surrogate model that predicts visual attributions to semantic units, thereby bypassing the need for computationally expensive gradient or perturbation based methods. The approach is evaluated across multiple VLMs and demonstrates improved faithfulness across a range of task domains.

All reviewers have a positive attitude toward this submission. Some minor concerns were raised, but most were appropriately addressed by the authors during the rebuttal. This is a clear accept.

Some potential wrong/fake citations:
- Laurençon, H., Saulnier, L., Tronchon, L., Bekman, S., Singh, A., Lozhkov, A., Wang, T., Karamcheti, S., Rush, A., Kiela, D., Cord, M., and Sanh, V. Unlocking the conversion of web screenshots into html code with the websight dataset. arXiv preprint arXiv:2403.09029, 2024.
- Yang, C., Huang, Y., Chen, Z., Gao, J., Zhong, W., Wang, J., Liang, J., Liu, X., Chen, D., Bhardwaj, R., Zhou, T., Cremers, D., Schiele, B., Wang, Q., Wu, L., and Li, A. Chartmimic: Evaluating lmm’s cross-modal reasoning capability via chart-to-code generation. arXiv preprint arXiv:2406.09961, 2024. ICLR 2025.